# Inductive Moment Matching

Linqi Zhou [1]  Stefano Ermon [2]  Jiaming Song [1]

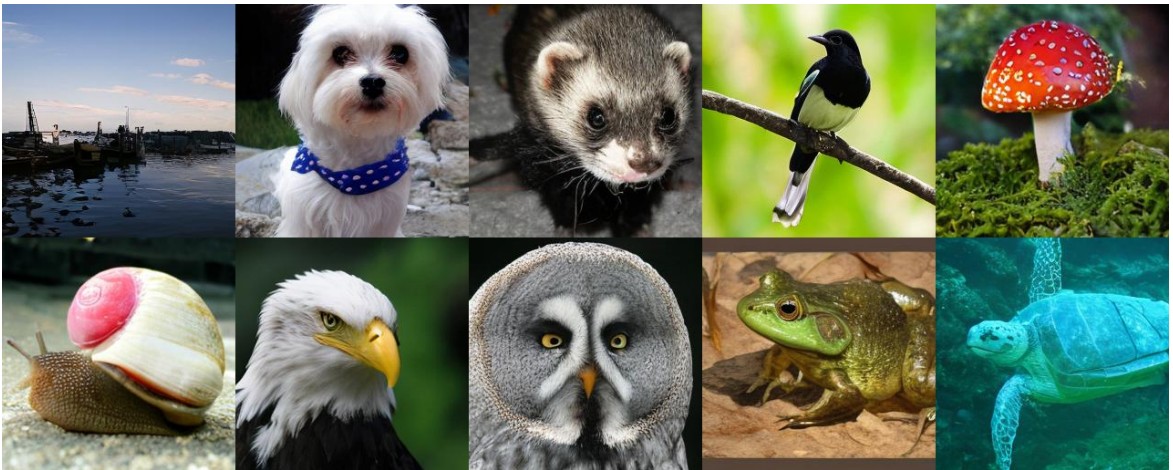

*Figure 1.* Generated samples on ImageNet-256×256 using 8 steps.

## Abstract

Diffusion models and Flow Matching generate high-quality samples but are slow at inference, and distilling them into few-step models often leads to instability and extensive tuning. To resolve these trade-offs, we propose Inductive Moment Matching (IMM), a new class of generative models for one- or few-step sampling with a *single-stage* training procedure. Unlike distillation, IMM does not require pre-training initialization and optimization of two networks; and unlike Consistency Models, IMM guarantees distribution-level convergence and remains stable under various hyperparameters and standard model architectures. IMM surpasses diffusion models on ImageNet-256×256 with 1.99 FID using only 8 inference steps and achieves state-of-the-art 2-step FID of 1.98 on CIFAR-10 for a model trained from scratch.

## 1. Introduction

Generative models for continuous domains have enabled numerous applications in images (Rombach et al., 2022; Saharia et al., 2022; Esser et al., 2024), videos (Ho et al., 2022a; Blattmann et al., 2023; OpenAI, 2024), and audio (Chen et al., 2020; Kong et al., 2020; Liu et al., 2023), yet achieving high-fidelity outputs, efficient inference, and stable training remains a core challenge — a trilemma that continues to motivate research in this domain. Diffusion models (Sohl-Dickstein et al., 2015; Ho et al., 2020; Song et al., 2020b), one of the leading techniques, require many inference steps for high-quality results, while step-reduction methods, such as diffusion distillation (Yin et al., 2024; Sauer et al., 2025; Zhou et al., 2024; Luo et al., 2024a) and Consistency Models (Song et al., 2023; Geng et al., 2024; Lu & Song, 2024; Kim et al., 2023), often risk training collapse without careful tuning and regularization (such as pre-generating data-noise pair and early stopping).

To address the aforementioned trilemma, we introduce Inductive Moment Matching (IMM), a stable, *single-stage* training procedure that learns generative models from scratch for single- or multi-step inference. IMM operates on the time-dependent marginal distributions of stochastic interpolants (Albergo et al., 2023) — continuous-time stochastic processes that connect two arbitrary probability

[1]Luma AI [2]Stanford University. Correspondence to: Linqi Zhou <alexzhou@lumalabs.ai>.

*Proceedings of the $42^{nd}$ International Conference on Machine Learning*, Vancouver, Canada. PMLR 267, 2025. Copyright 2025 by the author(s).

density functions (data at $t = 0$ and prior at $t = 1$). By learning a (stochastic or deterministic) mapping from any marginal at time $t$ to any marginal at time $s < t$, it can naturally support one- or multi-step generation (Figure 2).

IMM models can be trained efficiently from mathematical induction. For time $s < r < t$, we form two distributions at $s$ by running a one-step IMM from samples at $r$ and $t$. We then minimize their divergence, enforcing that the distributions at $s$ are independent of the starting time-steps. This construction by induction guarantees convergence to the data distribution. To help with training stability, we model IMM based on certain stochastic interpolants and optimize the objective with stable sample-based divergence estimators such as moment matching (Gretton et al., 2012). Notably, we prove that Consistency Models (CMs) are a single-particle, first-moment matching special case of IMM, which partially explains the training instability of CMs.

On ImageNet-256×256, IMM surpasses diffusion models and achieves 1.99 FID with only 8 inference steps using standard transformer architectures. On CIFAR-10, IMM similarly achieves state-of-the-art of 1.98 FID with 2-step generation for a model trained from scratch.

## 2. Preliminaries

### 2.1. Diffusion, Flow Matching, and Interpolants

For a data distribution $q(\mathbf{x})$, Variance-Preserving (VP) diffusion models (Ho et al., 2020; Song et al., 2020b) and Flow Matching (FM) (Lipman et al., 2022; Liu et al., 2022) construct time-augmented variables $\mathbf{x}_t$ as an interpolation between data $\mathbf{x} \sim q(\mathbf{x})$ and prior $\boldsymbol{\epsilon} \sim \mathcal{N}(0, I)$ such that $\mathbf{x}_t = \alpha_t \mathbf{x} + \sigma_t \boldsymbol{\epsilon}$ where $\alpha_0 = \sigma_1 = 1, \alpha_1 = \sigma_0 = 0$. VP diffusion commonly chooses $\alpha_t = \cos\left(\frac{\pi}{2}t\right), \sigma_t = \sin\left(\frac{\pi}{2}t\right)$ and FM chooses $\alpha_t = 1 - t, \sigma_t = t$. Both $v$-prediction diffusion (Salimans & Ho, 2022) and FM are trained by matching the conditional velocity $\mathbf{v}_t = \alpha_t' \mathbf{x} + \sigma_t' \boldsymbol{\epsilon}$ such that a neural network $\boldsymbol{G}_\theta(\mathbf{x}_t, t)$ approximates $\mathbb{E}_{\mathbf{x}, \boldsymbol{\epsilon}}[\mathbf{v}_t | \mathbf{x}_t]$. Samples can then be generated via probability-flow ODE (PF-ODE) $\frac{d\mathbf{x}_t}{dt} = \boldsymbol{G}_\theta(\mathbf{x}_t, t)$ starting from $\boldsymbol{\epsilon} \sim \mathcal{N}(0, I)$.

**Stochastic interpolants.** Unifying diffusion models and FM, stochastic interpolants (Albergo et al., 2023; Albergo & Vanden-Eijnden, 2022) construct a conditional interpolation $q_t(\mathbf{x}_t | \mathbf{x}, \boldsymbol{\epsilon}) = \mathcal{N}(\boldsymbol{I}_t(\mathbf{x}, \boldsymbol{\epsilon}), \gamma_t^2 I)$ between any data $\mathbf{x} \sim q(\mathbf{x})$ and prior $\boldsymbol{\epsilon} \sim p(\boldsymbol{\epsilon})$ and sets constraints $\boldsymbol{I}_1(\mathbf{x}, \boldsymbol{\epsilon}) = \boldsymbol{\epsilon}$, $\boldsymbol{I}_0(\mathbf{x}, \boldsymbol{\epsilon}) = \mathbf{x}$, and $\gamma_1 = \gamma_0 = 0$. Similar to FM, a deterministic sampler can be learned by explicitly matching the conditional interpolant velocity $\mathbf{v}_t = \partial_t \boldsymbol{I}_t(\mathbf{x}, \boldsymbol{\epsilon}) + \dot{\gamma}_t \mathbf{z}$ where $\mathbf{z} \sim \mathcal{N}(0, I)$ such that $\boldsymbol{G}_\theta(\mathbf{x}_t, t) \approx \mathbb{E}_{\mathbf{x}, \boldsymbol{\epsilon}, \mathbf{z}}[\mathbf{v}_t | \mathbf{x}_t]$. Sampling is performed following the PF-ODE $\frac{d\mathbf{x}_t}{dt} = \boldsymbol{G}_\theta(\mathbf{x}_t, t)$ similarly starting from prior $\boldsymbol{\epsilon} \sim p(\boldsymbol{\epsilon})$.

When $\gamma_t \equiv 0$ and $\boldsymbol{I}_t(\mathbf{x}, \boldsymbol{\epsilon}) = \alpha_t \mathbf{x} + \sigma_t \boldsymbol{\epsilon}$ for $\alpha_t, \sigma_t$ defined

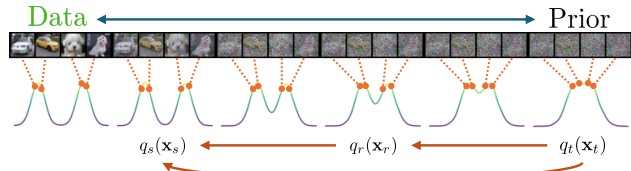

*Figure 2.* Using an interpolation from data to prior, we define a one-step sampler that moves from any $t$ to $s < t$, directly transforming $q_t(\mathbf{x}_t)$ to $q_s(\mathbf{x}_s)$. This can be repeated by jumping to an intermediate $r < t$ before moving to $s < r$.

in FM, the intermediate variable $\mathbf{x}_t = \alpha_t \mathbf{x} + \sigma_t \boldsymbol{\epsilon}$ becomes a deterministic interpolation and its interpolant velocity $\mathbf{v}_t = \alpha_t' \mathbf{x} + \sigma_t' \boldsymbol{\epsilon}$ reduces to FM velocity. Thus, its training and inference both reduce to that of FM. When $\boldsymbol{\epsilon} \sim \mathcal{N}(0, I)$, stochastic interpolants reduce to $v$-prediction diffusion.

### 2.2. Maximum Mean Discrepancy

Maximum Mean Discrepancy (MMD, Gretton et al. (2012)) between distribution $p(\mathbf{x}), q(\mathbf{y})$ for $\mathbf{x}, \mathbf{y} \in \mathbb{R}^D$ is an integral probability metric (Müller, 1997) commonly defined on Reproducing Kernel Hilbert Space (RKHS) $\mathcal{H}$ with a positive definite kernel $k : \mathbb{R}^D \times \mathbb{R}^D \to \mathbb{R}$ as

$$\text{MMD}^2(p(\mathbf{x}), q(\mathbf{y})) = \|\mathbb{E}_{\mathbf{x}}[k(\mathbf{x}, \cdot)] - \mathbb{E}_{\mathbf{y}}[k(\mathbf{y}, \cdot)]\|_{\mathcal{H}}^2 \quad (1)$$

where the norm is in $\mathcal{H}$. Choices such as the RBF kernel imply an inner product of infinite-dimensional feature maps consisting of *all* moments of $p(\mathbf{x})$ and $q(\mathbf{y})$, *i.e.* $\mathbb{E}[\mathbf{x}^j]$ and $\mathbb{E}[\mathbf{y}^j]$ for integer $j \geq 1$ (Steinwart & Christmann, 2008).

## 3. Inductive Moment Matching

We introduce Inductive Moment Matching (IMM), a method that trains a model of both high quality and sampling efficiency in a single stage. To do so, we assume a time-augmented interpolation between data (distribution at $t = 0$) and prior (distribution at $t = 1$) and propose learning an implicit one-step model (*i.e.* a one-step sampler) that transforms the distribution at time $t$ to the distribution at time $s$ for any $s < t$ (Section 3.1). The model enables direct one-step sampling from $t = 1$ to $s = 0$ and few-step sampling via recursive application from any $t$ to any $r < t$ and then to any $s < r$ until $s = 0$; this allows us to learn the model from its own samples via bootstrapping (Section 3.2).

### 3.1. Model Construction via Interpolants

Given data $\mathbf{x} \sim q(\mathbf{x})$ and prior $\boldsymbol{\epsilon} \sim p(\boldsymbol{\epsilon})$, the time-augmented interpolation $\mathbf{x}_t$ defined in Albergo et al. (2023) follows $\mathbf{x}_t \sim q_t(\mathbf{x}_t | \mathbf{x}, \boldsymbol{\epsilon})$. This implies a marginal interpolating distribution

$$q_t(\mathbf{x}_t) = \iint q_t(\mathbf{x}_t | \mathbf{x}, \boldsymbol{\epsilon}) q(\mathbf{x}) p(\boldsymbol{\epsilon}) d\mathbf{x} d\boldsymbol{\epsilon}. \quad (2)$$

We learn a model distribution implicitly defined by a one-step sampler that transforms $q_t(\mathbf{x}_t)$ into $q_s(\mathbf{x}_s)$ for some $s \leq t$. This can be done via a special class of interpolants, which preserves the marginal distribution $q_s(\mathbf{x}_s)$ while interpolating between $\mathbf{x}$ and $\mathbf{x}_t$. We term these *marginal-preserving* interpolants among a class of generalized interpolants. Formally, we define $\mathbf{x}_s$ as a generalized interpolant between $\mathbf{x}$ and $\mathbf{x}_t$ if, for all $s \in [0, t]$, its distribution follows

$$q_{s|t}(\mathbf{x}_s|\mathbf{x}, \mathbf{x}_t) = \mathcal{N}(\boldsymbol{I}_{s|t}(\mathbf{x}, \mathbf{x}_t), \gamma_{s|t}^2 I) \quad (3)$$

and satisfies constraints $\boldsymbol{I}_{t|t}(\mathbf{x}, \mathbf{x}_t) = \mathbf{x}_t$, $\boldsymbol{I}_{0|t}(\mathbf{x}, \mathbf{x}_t) = \mathbf{x}$, $\gamma_{t|t} = \gamma_{0|t} = 0$, and $q_{t|1}(\mathbf{x}_t|\mathbf{x}, \boldsymbol{\epsilon}) \equiv q_t(\mathbf{x}_t|\mathbf{x}, \boldsymbol{\epsilon})$. When $t = 1$, it reduces to regular stochastic interpolants. Next, we define marginal-preserving interpolants.

**Definition 1** (Marginal-Preserving Interpolants). A generalized interpolant $\mathbf{x}_s$ is *marginal-preserving* if for all $t \in [0, 1]$ and for all $s \in [0, t]$, the following equality holds:

$$q_s(\mathbf{x}_s) = \iint q_{s|t}(\mathbf{x}_s|\mathbf{x}, \mathbf{x}_t) q_t(\mathbf{x}|\mathbf{x}_t) q_t(\mathbf{x}_t) \mathrm{d}\mathbf{x}_t \mathrm{d}\mathbf{x}, \quad (4)$$

where

$$q_t(\mathbf{x}|\mathbf{x}_t) = \int \frac{q_t(\mathbf{x}_t|\mathbf{x}, \boldsymbol{\epsilon}) q(\mathbf{x}) p(\boldsymbol{\epsilon})}{q_t(\mathbf{x}_t)} \mathrm{d}\boldsymbol{\epsilon}. \quad (5)$$

That is, this class of interpolants has the same marginal at $s$ regardless of $t$. For all $t \in [0, 1]$, we define our *noisy* model distribution at $s \in [0, t]$ as

$$p_{s|t}^\theta(\mathbf{x}_s) = \iint q_{s|t}(\mathbf{x}_s|\mathbf{x}, \mathbf{x}_t) p_{s|t}^\theta(\mathbf{x}|\mathbf{x}_t) q_t(\mathbf{x}_t) \mathrm{d}\mathbf{x}_t \mathrm{d}\mathbf{x} \quad (6)$$

where the interpolant is marginal preserving and $p_{s|t}^\theta(\mathbf{x}|\mathbf{x}_t)$ is our *clean* model distribution implicitly parameterized as a one-step sampler. This definition also enables multi-step sampling. To produce a clean sample $\mathbf{x}$ given $\mathbf{x}_t \sim q_t(\mathbf{x}_t)$ in two steps via an intermediate $s$: (1) we sample $\hat{\mathbf{x}} \sim p_{s|t}^\theta(\mathbf{x}|\mathbf{x}_t)$ followed by $\hat{\mathbf{x}}_s \sim q_{s|t}(\mathbf{x}_s|\hat{\mathbf{x}}, \mathbf{x}_t)$ and (2) if the marginal of $\hat{\mathbf{x}}_s$ matches $q_s(\mathbf{x}_s)$, we can obtain $\mathbf{x}$ by $\mathbf{x} \sim p_{0|s}^\theta(\mathbf{x}|\hat{\mathbf{x}}_s)$. We are therefore motivated to minimize divergence between Eq. (4) and (6) using the objective below.

**Naïve objective.** As one can easily draw samples from the model, it can be naïvely learned by directly minimizing

$$\mathcal{L}(\theta) = \mathbb{E}_{s,t} \left[ D(q_s(\mathbf{x}_s), p_{s|t}^\theta(\mathbf{x}_s)) \right] \quad (7)$$

with time distribution $p(s, t)$ and a sample-based divergence metric $D(\cdot, \cdot)$ such as MMD or GAN (Goodfellow et al., 2020). If an interpolant $\mathbf{x}_s$ is marginal-preserving, then the minimum loss is 0 (see Lemma 3). One might also notice the similarity between right-hand sides of Eq. (4) and (6). However, $q_s(\mathbf{x}_s) = p_{s|t}^\theta(\mathbf{x}_s)$ does not necessarily imply $p_{s|t}^\theta(\mathbf{x}|\mathbf{x}_t) = q_t(\mathbf{x}|\mathbf{x}_t)$. In fact, the minimizer $p_{s|t}^\theta(\mathbf{x}|\mathbf{x}_t)$ is not unique and, under mild assumptions, a deterministic minimizer exists (see Section 4).

### 3.2. Learning via Inductive Bootstrapping

While sound, the naïve objective in Eq. (7) is difficult to optimize in practice because when $t$ is far from $s$, the input distribution $q_t(\mathbf{x}_t)$ can be far from the target $q_s(\mathbf{x}_s)$. Fortunately, our interpolant construction implies that the model definition in Eq. (6) satisfies boundary condition $q_s(\mathbf{x}_s) = p_{s|s}^\theta(\mathbf{x}_s)$ regardless of $\theta$ (see Lemma 4), which indicates that $p_{s|t}^\theta(\mathbf{x}_s) \approx q_s(\mathbf{x}_s)$ when $t$ is close to $s$. Furthermore, the interpolant enforces $p_{s|t}^\theta(\mathbf{x}_s) \approx p_{s|r}^\theta(\mathbf{x}_s)$ for any $r < t$ close to $t$ as long as the model is continuous around $t$. Therefore, we can construct an inductive learning algorithm for $p_{s|t}^\theta(\mathbf{x}_s)$ by using samples from $p_{s|r}^\theta(\mathbf{x}_s)$.

For better analysis, we define a sequence number $n$ for parameter $\theta_n$ and function $r(s, t)$ where $s \leq r(s, t) < t$ such that $p_{s|t}^{\theta_n}(\mathbf{x}_s)$ learns to match $p_{s|r}^{\theta_{n-1}}(\mathbf{x}_s)$.[1] We omit $r$'s arguments when context is clear and let $r(s, t)$ be a finite decrement from $t$ but truncated at $s \leq t$ (see Appendix B.3 for well-conditioned $r(s, t)$).

**General objective.** With marginal-preserving interpolants and mapping $r(s, t)$, we learn $\theta_n$ in the following objective:

$$\mathcal{L}(\theta_n) = \mathbb{E}_{s,t} \left[ w(s,t) \mathrm{MMD}^2(p_{s|r}^{\theta_{n-1}}(\mathbf{x}_s), p_{s|t}^{\theta_n}(\mathbf{x}_s)) \right] \quad (8)$$

where $w(s, t)$ is a weighting function. We choose MMD as our objective due to its superior optimization stability and show that this objective learns the correct data distribution.

**Theorem 1.** *Assuming $r(s, t)$ is well-conditioned, the interpolant is marginal-preserving, and $\theta_n^*$ is a minimizer of Eq. (8) for each $n$ with infinite data and network capacity, for all $t \in [0, 1]$, $s \in [0, t]$,*

$$\lim_{n \to \infty} \mathrm{MMD}^2(q_s(\mathbf{x}_s), p_{s|t}^{\theta_n^*}(\mathbf{x}_s)) = 0. \quad (9)$$

In other words, $\theta_n$ eventually learns the target distribution $q_s(\mathbf{x}_s)$ by parameterizing a one-step sampler $p_{s|t}^{\theta_n}(\mathbf{x}|\mathbf{x}_t)$.

## 4. Simplified Formulation and Practice

We present algorithmic and practical decisions below.

### 4.1. Algorithmic Considerations

Despite theoretical soundness, it remains unclear how to empirically choose a marginal-preserving interpolant. First, we present a sufficient condition for marginal preservation.

**Definition 2** (Self-Consistent Interpolants). Given $s, t \in [0, 1]$, $s \leq t$, an interpolant $\mathbf{x}_s \sim q_{s|t}(\mathbf{x}_s|\mathbf{x}, \mathbf{x}_t)$ is *self-consistent* if for all $r \in [s, t]$, the following holds:

$$q_{s|t}(\mathbf{x}_s|\mathbf{x}, \mathbf{x}_t) = \int q_{s|r}(\mathbf{x}_s|\mathbf{x}, \mathbf{x}_r) q_{r|t}(\mathbf{x}_r|\mathbf{x}, \mathbf{x}_t) \mathrm{d}\mathbf{x}_r \quad (10)$$

---

[1]Note that $n$ is different from optimization steps. Advancing from $n - 1$ to $n$ can take arbitrary number of optimization steps.

In other words, $\mathbf{x}_s$ has the same distribution if one (1) directly samples it by interpolating $\mathbf{x}$ and $\mathbf{x}_t$ and (2) first samples any $\mathbf{x}_r$ (given $\mathbf{x}$ and $\mathbf{x}_t$) and then samples $\mathbf{x}_s$ (given $\mathbf{x}$ and $\mathbf{x}_r$). Furthermore, self-consistency implies marginal preservation (Lemma 5).

**DDIM interpolant.** Denoising Diffusion Implicit Models (Song et al., 2020a) was introduced as a fast ODE sampler for diffusion models, defined as

$$\mathrm{DDIM}(\mathbf{x}_t, \mathbf{x}, s, t) = \left(\alpha_s - \frac{\sigma_s}{\sigma_t}\alpha_t\right)\mathbf{x} + \frac{\sigma_s}{\sigma_t}\mathbf{x}_t \quad (11)$$

and sample $\mathbf{x}_s = \mathrm{DDIM}(\mathbf{x}_t, \mathbb{E}_{\mathbf{x}}[\mathbf{x}|\mathbf{x}_t], s, t)$ can be drawn when $\mathbb{E}_{\mathbf{x}}[\mathbf{x}|\mathbf{x}_t]$ is approximated by a network. We show in Appendix C.1 that DDIM as an interpolant, *i.e.* $\gamma_{s|t} \equiv 0$ and $\boldsymbol{I}_{s|t}(\mathbf{x}, \mathbf{x}_t) = \mathrm{DDIM}(\mathbf{x}_t, \mathbf{x}, s, t)$, is self-consistent. Moreover, with deterministic interpolants such as DDIM, there exists a deterministic minimizer $p_{s|t}^\theta(\mathbf{x}|\mathbf{x}_t)$ of Eq. (7).

**Proposition 1.** *(Informal) If $\gamma_{s|t} \equiv 0$ and $\boldsymbol{I}_{s|t}(\mathbf{x}, \mathbf{x}_t)$ satisfies mild assumptions, there exists a deterministic $p_{s|t}^\theta(\mathbf{x}|\mathbf{x}_t)$ that attains 0 loss for Eq. (7).*

See Appendix B.6 for formal statement and proof. This allows us to define $p_{s|t}^\theta(\mathbf{x}|\mathbf{x}_t) = \delta(\mathbf{x} - \boldsymbol{g}_\theta(\mathbf{x}_t, s, t))$ for a neural network $\boldsymbol{g}_\theta(\mathbf{x}_t, s, t)$ with parameter $\theta$ by default.

**Eliminating stochasticity.** We use DDIM interpolant, deterministic model, and prior $p(\boldsymbol{\epsilon}) = \mathcal{N}(0, \sigma_d^2 I)$ where $\sigma_d$ is the data standard deviation (Lu & Song, 2024). As a result, one can draw $\mathbf{x}_s$ from model via $\mathbf{x}_s = \boldsymbol{f}_{s,t}^\theta(\mathbf{x}_t) := \mathrm{DDIM}(\mathbf{x}_t, \boldsymbol{g}_\theta(\mathbf{x}_t, s, t), s, t)$ where $\mathbf{x}_t \sim q_t(\mathbf{x}_t)$.

**Re-using $\mathbf{x}_t$ for $\mathbf{x}_r$.** Inspecting Eq. (8) and (6), one requires $\mathbf{x}_r \sim q_r(\mathbf{x}_r)$ to generate samples from the target distribution. Instead of sampling $\mathbf{x}_r$ given a new $(\mathbf{x}, \boldsymbol{\epsilon})$ pair, we can reduce variance by reusing $\mathbf{x}_t$ and $\mathbf{x}$ such that $\mathbf{x}_r = \mathrm{DDIM}(\mathbf{x}_t, \mathbf{x}, r, t)$. This is justified because $\mathbf{x}_r$ derived from $\mathbf{x}_t$ preserves the marginal distribution $q_r(\mathbf{x}_r)$ (see Appendix C.2).

**Stop gradient.** We set $n$ to optimization step number, *i.e.* advancing from $n-1$ to $n$ is a single optimizer step where $\theta_n$ is initialized from $\theta_{n-1}$. Equivalently, we can omit $n$ from $\theta_n$ and write $\theta_{n-1}$ as the stop-gradient parameter $\theta^-$.

**Simplified objective.** Let $\mathbf{x}_t, \mathbf{x}'_t$ be i.i.d. random variables from $q_t(\mathbf{x}_t)$ and $\mathbf{x}_r, \mathbf{x}'_r$ are variables obtained by reusing $\mathbf{x}_t, \mathbf{x}'_t$ respectively, the training objective can be derived from the MMD definition in Eq. (1) (see Appendix C.3) as

$$\mathcal{L}_{\mathrm{IMM}}(\theta) = \mathbb{E}_{\mathbf{x}_t, \mathbf{x}'_t, \mathbf{x}_r, \mathbf{x}'_r, s, t}\Big[w(s, t)\big[k\big(\mathbf{y}_{s,t}, \mathbf{y}'_{s,t}\big) \quad (12)$$

$$+ k\big(\mathbf{y}_{s,r}, \mathbf{y}'_{s,r}\big) - k\big(\mathbf{y}_{s,t}, \mathbf{y}'_{s,r}\big) - k\big(\mathbf{y}'_{s,t}, \mathbf{y}_{s,r}\big)\big]\Big]$$

where $\mathbf{y}_{s,t} = \boldsymbol{f}_{s,t}^\theta(\mathbf{x}_t)$, $\mathbf{y}'_{s,t} = \boldsymbol{f}_{s,t}^\theta(\mathbf{x}'_t)$, $\mathbf{y}_{s,r} = \boldsymbol{f}_{s,r}^{\theta^-}(\mathbf{x}_r)$, $\mathbf{y}'_{s,r} = \boldsymbol{f}_{s,r}^{\theta^-}(\mathbf{x}'_r)$, $k(\cdot, \cdot)$ is a kernel function, and $w(s, t)$ is a prior weighting function.

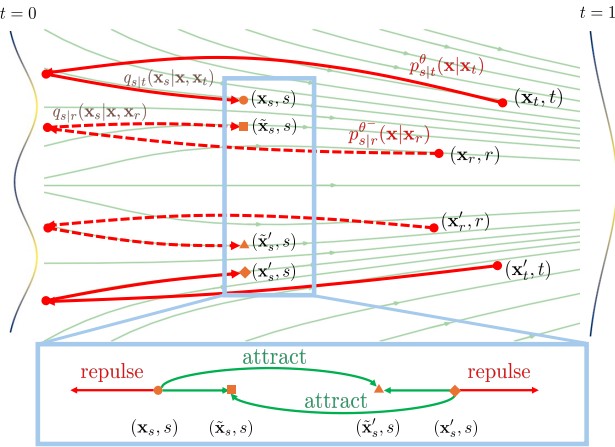

*Figure 3.* With self-consistent interpolants, IMM uses $M$ particle samples ($M = 2$ is shown above) for moment matching. Samples from $p_{s|t}^\theta(\mathbf{x}_s)$ are obtained by drawing from $p_{s|t}^\theta(\mathbf{x}|\mathbf{x}_t)$ followed by $q_{s|t}(\mathbf{x}_s|\mathbf{x}, \mathbf{x}_t)$. Solid and dashed red lines indicate sampling with and without gradient propagation respectively. After $M$ samples are drawn, sample $\mathbf{x}_s$ is repulsed by $\mathbf{x}'_s$ and attracted towards samples of $\tilde{\mathbf{x}}_s$ and $\tilde{\mathbf{x}}'_s$ through kernel function $k(\cdot, \cdot)$.

An empirical estimate of the above objective uses $M$ particle samples to approximate each distribution indexed by $t$. In practice, we divide a batch of model output with size $B$ into $B/M$ groups within which share the same $(s, t)$ sample, and the objective is approximated by instantiating $B/M$ number of $M \times M$ matrices. Note that the number of model passes does not change with respect to $M$ (see Appendix C.4). A $M = 2$ version is visualized in Figure 3 and a simplified training algorithm is shown in Algorithm 1. A full training algorithm is shown in Appendix D.

### 4.2. Other Implementation Choices

We defer detailed analysis of each decision to Appendix C.

**Flow trajectories.** We investigate the two most used flow trajectories (Nichol & Dhariwal, 2021; Lipman et al., 2022),

- **Cosine.** $\alpha_t = \cos\big(\frac{1}{2}\pi t\big)$, $\sigma_t = \sin\big(\frac{1}{2}\pi t\big)$.
- **OT-FM.** $\alpha_t = 1 - t$, $\sigma_t = t$.

**Network $\boldsymbol{g}_\theta(\mathbf{x}_t, s, t)$.** We set $\boldsymbol{g}_\theta(\mathbf{x}_t, s, t) = c_{\mathrm{skip}}(t)\mathbf{x}_t + c_{\mathrm{out}}(t)\boldsymbol{G}_\theta(c_{\mathrm{in}}(t)\mathbf{x}_t, c_{\mathrm{noise}}(s), c_{\mathrm{noise}}(t))$ with a neural network $\boldsymbol{G}_\theta$, following EDM (Karras et al., 2022). For all choices we let $c_{\mathrm{in}}(t) = 1/\sqrt{\alpha_t^2 + \sigma_t^2}/\sigma_d$ (Lu & Song, 2024). Listed below are valid choices for other coefficients.

- **Identity.** $c_{\mathrm{skip}}(t) = 0$, $c_{\mathrm{out}}(t) = 1$.
- **Simple-EDM** (Lu & Song, 2024). $c_{\mathrm{skip}}(t) = \alpha_t/(\alpha_t^2 + \sigma_t^2)$, $c_{\mathrm{out}}(t) = -\sigma_d\sigma_t/\sqrt{\alpha_t^2 + \sigma_t^2}$.
- **Euler-FM.** $c_{\mathrm{skip}}(t) = 1$, $c_{\mathrm{out}}(t) = -t\sigma_d$. This is specific to OT-FM schedule.

We show in Appendix C.5 that $\boldsymbol{f}_{s,t}^\theta(\mathbf{x}_t)$ similarly follows the

EDM parameterization of the form $\boldsymbol{f}_{s,t}^\theta(\mathbf{x}_t) = c_{\text{skip}}(s,t)\mathbf{x}_t + c_{\text{out}}(s,t)\boldsymbol{G}_\theta(c_{\text{in}}(t)\mathbf{x}_t, c_{\text{noise}}(s), c_{\text{noise}}(t))$.

**Noise conditioning $c_{\text{noise}}(\cdot)$.** We choose $c_{\text{noise}}(t) = ct$ for some constant $c \geq 1$. We find our model convergence relatively insensitive to $c$ but recommend using larger $c$, *e.g.* 1000 (Song et al., 2020b; Peebles & Xie, 2023), because it enables sufficient distinction between nearby $r$ and $t$.

**Mapping function $r(s,t)$.** We find that $r(s,t)$ via constant decrement in $\eta_t = \sigma_t/\alpha_t$ works well where the decrement is chosen in the form of $(\eta_{\max} - \eta_{\min})/2^k$ for some appropriate $k$ (details in Appendix C.7).

**Kernel function.** We use time-dependent Laplace kernels of the form $k_{s,t}(x,y) = \exp(-\tilde{w}(s,t)\max(\|x-y\|_2, \epsilon)/D)$ for $x, y \in \mathbb{R}^D$, some $\epsilon > 0$ to avoid undefined gradients, and $\tilde{w}(s,t) = 1/|c_{\text{out}}(s,t)|$. We find Laplace kernels provide better gradient signals than RBF kernels. (see Appendix C.8).

**Weighting $w(s,t)$ and distribution $p(s,t)$.** We follow VDM (Kingma et al., 2021; Kingma & Gao, 2024) and define $p(t) = \mathcal{U}(\epsilon, T)$ and $p(s|t) = \mathcal{U}(\epsilon, t)$ for constants $\epsilon, T \in [0,1]$. Similarly, weighting is defined as

$$w(s,t) = \frac{1}{2}\sigma(b - \lambda_t)\left(-\frac{\mathrm{d}}{\mathrm{d}t}\lambda_t\right)\frac{\alpha_t^a}{\alpha_t^2 + \sigma_t^2} \quad (13)$$

where $\sigma(\cdot)$ is sigmoid function, $\lambda_t$ denotes log-SNR$_t$ and $a \in \{1,2\}$, $\sigma(\cdot), b \in \mathbb{R}$ are constants (see Appendix C.9).

### 4.3. Sampling

**Pushforward sampling.** A sample $\mathbf{x}_s$ can be obtained by directly pushing $\mathbf{x}_t \sim q_t(\mathbf{x}_t)$ through $\boldsymbol{f}_{s,t}^\theta(\mathbf{x}_t)$. This can be iterated for an arbitrary number of steps starting from $\boldsymbol{\epsilon} \sim \mathcal{N}(0, \sigma_d^2 I)$ until $s = 0$. We note that, by definition, one application of $\boldsymbol{f}_{s,t}^\theta(\mathbf{x}_t)$ is equivalent to one DDIM step using the learned network $\boldsymbol{g}_\theta(\mathbf{x}_t, s, t)$ as the $\mathbf{x}$ prediction. This sampler can then be viewed as a few-step sampler using DDIM where $\boldsymbol{g}_\theta(\mathbf{x}_t, s, t)$ outputs a realistic sample $\mathbf{x}$ instead of its expectation $\mathbb{E}_{\mathbf{x}}[\mathbf{x}|\mathbf{x}_t]$ as in diffusion models.

**Restart sampling.** Similar to Xu et al. (2023); Song et al. (2023), one can introduce stochasticity during sampling by re-noising a sample to a higher noise-level before sampling again. For example, a two-step restart sampler from $\mathbf{x}_t$ requires $s \in (0, t)$ for drawing sample $\hat{\mathbf{x}} = \boldsymbol{f}_{0,s}^\theta(\mathbf{x}_s)$ where $\mathbf{x}_s \sim q_s(\mathbf{x}_s|\boldsymbol{f}_{0,t}^\theta(\mathbf{x}_t))$.

**Classifier-free guidance.** Given a data-label pair $(\mathbf{x}, \mathbf{c})$, during inference time, classifier-free guidance (Ho & Salimans, 2022) with weight $w$ replaces conditional model output $\boldsymbol{G}_\theta(\mathbf{x}_t, s, t, \mathbf{c})$ by a reweighted model output via

$$w\boldsymbol{G}_\theta(\mathbf{x}_t, s, t, \mathbf{c}) + (1-w)\boldsymbol{G}_\theta(\mathbf{x}_t, s, t, \varnothing) \quad (14)$$

where $\varnothing$ denotes the null-token indicating unconditional out-

---

**Algorithm 1** Training (see Appendix D for full version)

**Input:** parameter $\theta$, DDIM$(\mathbf{x}_t, \mathbf{x}, s, t)$, $B$, $M$, $p$
**Output:** learned $\theta$
**while** model not converged **do**
    Sample data $x$, label $c$, and prior $\epsilon$ with batch size $B$ and split into $B/M$ groups. Each group shares a $(s, r, t)$ sample.
    For each group, $x_t \leftarrow$ DDIM$(\epsilon, x, t, 1)$.
    For each group, $x_r \leftarrow$ DDIM$(x_t, x, r, t)$.
    For each instance, set $c = \varnothing$ with prob. $p$.
    Minimize the empirical loss $\hat{\mathcal{L}}_{\text{IMM}}(\theta)$ in Eq. (67).
**end while**

---

**Algorithm 2** Pushforward Sampling (details in Appendix F)

**Input:** model $\boldsymbol{f}^\theta$, $\{t_i\}_{i=0}^N$, $\mathcal{N}(0, \sigma_d^2 I)$, (optional) $w$
**Output:** $\mathbf{x}_{t_0}$
**Sample** $\mathbf{x}_N \sim \mathcal{N}(0, \sigma_d^2 I)$
**for** $i = N, \ldots, 1$ **do**
    $\mathbf{x}_{t_{i-1}} \leftarrow \boldsymbol{f}_{t_{i-1}, t_i}^\theta(\mathbf{x}_{t_i})$ or $\boldsymbol{f}_{t_{i-1}, t_i, w}^\theta(\mathbf{x}_{t_i})$.
**end for**

---

put. Similarly, we define our guided model as $\boldsymbol{f}_{s,t,w}^\theta(\mathbf{x}_t) = c_{\text{skip}}(s,t)\mathbf{x}_t + c_{\text{out}}(s,t)\boldsymbol{G}_\theta^w(\mathbf{x}_t, s, t, \mathbf{c})$ where $\boldsymbol{G}_\theta^w(\mathbf{x}_t, s, t, \mathbf{c})$ is as defined in Eq. (14) and we drop $c_{\text{in}}(\cdot)$ and $c_{\text{noise}}(\cdot)$ for notational simplicity. We justify this decision in Appendix E. Similar to diffusion models, $\mathbf{c}$ is randomly dropped with probability $p$ during training without special practices.

We present pushforward sampling in Algorithm 2 and detail both samplers in Appendix F.

## 5. Connection with Prior Works

Our work is closely connected with many prior works. Detailed analysis is found in Appendix G.

**Consistency Models.** Consistency models (CMs) (Song et al., 2023; Song & Dhariwal, 2023; Lu & Song, 2024) uses a network $\boldsymbol{g}_\theta(\mathbf{x}_t, t)$ that outputs clean data given noisy input $\mathbf{x}_t$. It requires point-wise consistency $\boldsymbol{g}_\theta(\mathbf{x}_t, t) = \boldsymbol{g}_\theta(\mathbf{x}_r, r)$ for any $r < t$ where $\mathbf{x}_r$ is obatined via an ODE solver from $\mathbf{x}_t$ using either pretrained model or groundtruth data. Discrete-time CM must satisfy $\boldsymbol{g}_\theta(\mathbf{x}_0, 0) = \mathbf{x}_0$ and trains via loss $\mathbb{E}_{\mathbf{x}_t, \mathbf{x}, t}[d(\boldsymbol{g}_\theta(\mathbf{x}_t, t), \boldsymbol{g}_{\theta^-}(\mathbf{x}_r, r))]$ where $d(\cdot, \cdot)$ is commonly chosen as $\mathcal{L}_2$ or LPIPS (Zhang et al., 2018).

We show in the following Lemma that CM objective with $\mathcal{L}_2$ distance is a single-particle estimate of IMM objective with energy kernel.

**Lemma 1.** *When $\mathbf{x}_t = \mathbf{x}'_t$, $\mathbf{x}_r = \mathbf{x}'_r$, $k(x,y) = -\|x-y\|^2$, and $s > 0$ is a small constant, Eq. (12) reduces to CM loss $\mathbb{E}_{\mathbf{x}_t, \mathbf{x}, t}\left[w(t)\|\boldsymbol{g}_\theta(\mathbf{x}_t, t) - \boldsymbol{g}_{\theta^-}(\mathbf{x}_r, r)\|^2\right]$ for some valid mapping $r(t) < t$.*

This single-particle estimate ignores the repulsion force

imposed by $k(\cdot, \cdot)$. Energy kernel also only matches the *first* moment, ignoring all higher moments. These decisions can be significant contributors to training instability and performance degradation of CMs.

Improved CMs (Song & Dhariwal, 2023) propose pseudo-huber loss as $d(\cdot, \cdot)$ which we justify in the Lemma below.

**Lemma 2.** *Negative pseudo-huber loss* $k_c(x, y) = c - \sqrt{\|x - y\|^2 + c^2}$ *for* $c > 0$ *is a conditionally positive definite kernel that matches all moments of* $x$ *and* $y$ *where weights on higher moments depend on* $c$.

From a moment-matching perspective, the improved performance is explained by the loss matching all moments of the distributions. In addition to pseudo-huber loss, many other kernels (Laplace, RBF, *etc.*) are all valid choices in the design space.

We also extend IMM loss to the differential limit by taking $r(s, t) \to t$, the result of which subsumes the continuous-time CM (Lu & Song, 2024) as a single-particle estimate (Appendix H). We leave experiments for this to future work.

**Diffusion GAN and Adversarial Consistency Distillation.** Diffusion GAN (Xiao et al., 2021) parameterizes the generative distribution as $p_{s|t}^\theta(\mathbf{x}_s|\mathbf{x}_t) = \iint q_{s|t}(\mathbf{x}_s|\mathbf{x}, \mathbf{x}_t)\delta(\mathbf{x} - \mathbf{G}_\theta(\mathbf{x}_t, \mathbf{z}, t))p(\mathbf{z})\mathrm{d}\mathbf{z}\mathrm{d}\mathbf{x}$ for $s$ as a fixed decrement from $t$ and $p(\mathbf{z})$ a noise distribution. It defines the interpolant $q_{s|t}(\mathbf{x}_s|\mathbf{x}, \mathbf{x}_t)$ as the DDPM posterior distribution, which is self-consistent (see Appendix G.2) and introduces randomness to the sampling process to match $q_t(\mathbf{x}|\mathbf{x}_t)$ instead of the marginal. Both Diffusion GAN and Adversarial Consistency Distillation (Sauer et al., 2025) use GAN objective, which shares similarity to MMD in that MMD is defined as an integral probability metric where the optimal discriminator is chosen in RKHS. This eliminates the need for explicit adversarial optimization of a neural-network discriminator.

**Generative Moment Matching Network.** GMMN (Li et al., 2015) directly applies MMD to train a generator $\mathbf{G}_\theta(\mathbf{z})$ where $\mathbf{z} \sim \mathcal{N}(0, I)$ to match the data distribution. It is a special case of IMM in that when $t = 1$ and $r(s, t) \equiv s = 0$ our loss reduces to naïve GMMN objective.

## 6. Related Works

**Diffusion, Flow Matching, and stochastic interpolants.** Diffusion models (Sohl-Dickstein et al., 2015; Song et al., 2020b; Ho et al., 2020; Kingma et al., 2021) and Flow Matching (Lipman et al., 2022; Liu et al., 2022) are widely used generative frameworks that learn a score or velocity field of a noising process from data into a simple prior. They have been scaled successfully for text-to-image (Rombach et al., 2022; Saharia et al., 2022; Podell et al., 2023; Chen et al., 2023; Esser et al., 2024) and text-to-video (Ho et al., 2022a; Blattmann et al., 2023; OpenAI, 2024) tasks.

Stochastic interpolants (Albergo et al., 2023; Albergo & Vanden-Eijnden, 2022) extend these ideas by explicitly defining a stochastic path between data and prior, then matching its velocity to facilitate distribution transfer. While IMM builds on top of the interpolant construction, it directly learns one-step mappings between any intermediate marginal distributions.

**Diffusion distillation.** To resolve diffusion models' sampling inefficiency, recent methods (Salimans & Ho, 2022; Meng et al., 2023; Yin et al., 2024; Zhou et al., 2024; Luo et al., 2024a; Heek et al., 2024) focus on distilling one-step or few-step models from pre-trained diffusion models. This two-stage approach currently has the best generation quality and efficiency. However, such methods usually require jointly optimizing two networks and training requires careful tuning in practice to avoid mode collapse (Yin et al., 2024). Another recent work (Salimans et al., 2024) explicitly matches the *first* moment of the data distribution available from pre-trained diffusion models. In contrast, our method implicitly matches *all* moments using MMD and can be trained from scratch with a single model.

**Few-step generative models from scratch.** Early one-step generative models primarily relied on GANs (Goodfellow et al., 2020; Karras et al., 2020; Brock, 2018) and MMD (Li et al., 2015; 2017) (or their combination) but scaling adversarial training remains challenging. Recent independent classes of few-step models, *e.g.* Consistency Models (CMs) (Song et al., 2023; Song & Dhariwal, 2023; Lu & Song, 2024), Consistency Trajectory Models (CTMs) (Kim et al., 2023; Heek et al., 2024) and Shortcut Models (SMs) (Frans et al., 2024) still face training instability and require specialized components (Lu & Song, 2024) (e.g., JVP for flash attention) or other special practices (e.g., high weight decay for SMs, combined LPIPS (Zhang et al., 2018) and GAN losses for CTMs, and special training schedules (Geng et al., 2024)) to remain stable. In contrast, our method trains stably with a single loss and achieves strong performance without special training practices.

## 7. Experiments

We evaluate IMM's empirical performance (Section 7.1), training stability (Section 7.2), sampling choices (Section 7.3), scaling behavior (Section 7.4) and ablate our practical decisions (Section 7.5).

### 7.1. Image Generation

We present FID (Heusel et al., 2017) results for unconditional CIFAR-10 and class-conditional ImageNet-256×256 in Table 1 and 2. For CIFAR-10, we separate baselines into diffusion and flow models, distillation models, and few-step models from scratch. IMM belongs to the last category

| Family | Method | FID ($\downarrow$) | Steps ($\downarrow$) |
|---|---|---|---|
| Diffusion & Flow | DDPM (Ho et al., 2020) | 3.17 | 1000 |
| | DDPM++ (Song et al., 2020b) | 3.16 | 1000 |
| | NCSN++ (Song et al., 2020b) | 2.38 | 1000 |
| | DPM-Solver (Lu et al., 2022) | 4.70 | 10 |
| | iDDPM (Nichol & Dhariwal, 2021) | 2.90 | 4000 |
| | EDM (Karras et al., 2022) | 2.05 | 35 |
| | Flow Matching (Lipman et al., 2022) | 6.35 | 142 |
| | Rectified Flow (Liu et al., 2022) | 2.58 | 127 |
| Few-Step via Distillation | PD (Salimans & Ho, 2022) | 4.51 | 2 |
| | 2-Rectified Flow (Salimans & Ho, 2022) | 4.85 | 1 |
| | DFNO (Zheng et al., 2023) | 3.78 | 1 |
| | KD (Luhman & Luhman, 2021) | 9.36 | 1 |
| | TRACT (Berthelot et al., 2023) | 3.32 | 2 |
| | Diff-Instruct (Luo et al., 2024a) | 5.57 | 1 |
| | PID (LPIPS) (Tee et al., 2024) | 3.92 | 1 |
| | DMD (Yin et al., 2024) | 3.77 | 1 |
| | CD (LPIPS) (Song et al., 2023) | 2.93 | 2 |
| | CTM (w/ GAN) (Kim et al., 2023) | **1.87** | 2 |
| | SiD (Zhou et al., 2024) | 1.92 | 1 |
| | SiM (Luo et al., 2024b) | 2.06 | 1 |
| | sCD (Lu & Song, 2024) | 2.52 | 2 |
| Few-Step from Scratch | iCT (Song & Dhariwal, 2023) | 2.83 | 1 |
| | | 2.46 | 2 |
| | ECT (Geng et al., 2024) | 3.60 | 1 |
| | | 2.11 | 2 |
| | sCT (Lu & Song, 2024) | 2.97 | 1 |
| | | 2.06 | 2 |
| | **IMM (ours)** | 3.20 | 1 |
| | | **1.98** | 2 |

*Table 1.* CIFAR-10 results trained without label conditions.

| Family | Method | FID($\downarrow$) | Steps ($\downarrow$) | #Params |
|---|---|---|---|---|
| GAN | BigGAN (Brock, 2018) | 6.95 | 1 | 112M |
| | GigaGAN (Kang et al., 2023) | 3.45 | 1 | 569M |
| | StyleGAN-XL (Karras et al., 2020) | 2.30 | 1 | 166M |
| Masked & AR | VQGAN (Esser et al., 2021) | 26.52 | 1024 | 227M |
| | MaskGIT (Chang et al., 2022) | 6.18 | 8 | 227M |
| | MAR (Li et al., 2024) | 1.98 | 100 | 400M |
| | VAR-$d20$ (Tian et al., 2024a) | 2.57 | 10 | 600M |
| | VAR-$d30$ (Tian et al., 2024a) | **1.92** | 10 | 2B |
| Diffusion & Flow | ADM (Dhariwal & Nichol, 2021) | 10.94 | 250 | 554M |
| | CDM (Ho et al., 2022b) | 4.88 | 8100 | - |
| | SimDiff (Hoogeboom et al., 2023) | 2.77 | 512 | 2B |
| | LDM-4-G (Rombach et al., 2022) | 3.60 | 250 | 400M |
| | U-DiT-L (Tian et al., 2024b) | 3.37 | 250 | 916M |
| | U-ViT-H (Bao et al., 2023) | 2.29 | 50 | 501M |
| | DiT-XL/2 ($w = 1.0$) (Peebles & Xie, 2023) | 9.62 | 250 | 675M |
| | DiT-XL/2 ($w = 1.25$) (Peebles & Xie, 2023) | 3.22 | 250 | 675M |
| | DiT-XL/2 ($w = 1.5$) (Peebles & Xie, 2023) | 2.27 | 250 | 675M |
| | SiT-XL/2 ($w = 1.0$) (Ma et al., 2024) | 9.35 | 250 | 675M |
| | SiT-XL/2 ($w = 1.5$) (Ma et al., 2024) | 2.15 | 250 | 675M |
| Few-Step from Scratch | iCT (Song et al., 2023) | 34.24 | 1 | 675M |
| | | 20.3 | 2 | 675M |
| | Shortcut (Frans et al., 2024) | 10.60 | 1 | 675M |
| | | 7.80 | 4 | 675M |
| | | 3.80 | 128 | 675M |
| | **IMM (ours)** (XL/2, $w = 1.25$) | 7.77 | 1 | 675M |
| | | 5.33 | 2 | 675M |
| | | 3.66 | 4 | 675M |
| | | 2.77 | 8 | 675M |
| | **IMM (ours)** (XL/2, $w = 1.5$) | 8.05 | 1 | 675M |
| | | 3.99 | 2 | 675M |
| | | 2.51 | 4 | 675M |
| | | **1.99** | 8 | 675M |

*Table 2.* Class-conditional ImageNet-256×256 results.

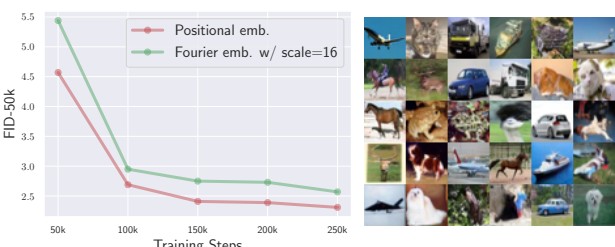

*Figure 4.* FID convergence for different embeddings (left). CIFAR-10 samples from Fourier embedding (scale = 16) (right).

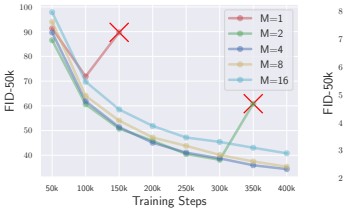

*Figure 5.* More particles indicate more stable training on ImageNet-256×256.

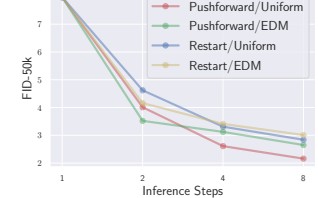

*Figure 6.* ImageNet-256×256 FID with different sampler types.

in which it achieves state-of-the-art performance of 1.98 using pushforward sampler. For ImageNet-256×256, we use the popular DiT (Peebles & Xie, 2023) architecture because of its scalability, and compare it with GANs, masked and autoregressive models, diffusion and flow models, and few-step models trained from scratch.

We observe decreasing FID with more steps and IMM achieves 1.99 FID with 8 steps (with $w = 1.5$), surpassing DiT and SiT (Ma et al., 2024) using the same architecture except for trivially injecting time $s$ (see Appendix I). Notably, we also achieve better 8-step FID than the 10-step VAR (Tian et al., 2024a) of comparable size. At 16 steps, IMM also achieves 1.90 FID outperforming VAR's 2B variant (see Appendix I.4). However, different from VAR, IMM grants flexibility of variable number of inference steps. Lastly, we similarly surpass Shortcut models' (Frans et al., 2024) best performance with only 8 steps[2]. We defer inference details to Section 7.3 and Appendix I.2.

---

[2]NFE is twice the number of steps.

## 7.2. IMM Training is Stable

We show that IMM is stable and achieves reasonable performance across a range of parameterization choices.

**Positional vs. Fourier embedding.** A known issue for CMs (Song et al., 2023) is its training instability when using Fourier embedding with scale 16, which forces reliance on positional embeddings for stability. We find that IMM does not face this problem (see Figure 4). For Fourier embedding we use the standard NCSN++ (Song et al., 2020b) architecture and set embedding scale to 16; for positional embeddings, we adopt DDPM++ (Song et al., 2020b). Both embedding types converge reliably, and we include samples from the Fourier embedding model in Figure 4.

**Particle number.** Particle number $M$ for estimating MMD is an important parameter for empirical success (Gretton et al., 2012; Li et al., 2015), where the estimate is more accurate with larger $M$. In our case, naïvely increasing $M$ can slow down convergence because we have a fixed batch size $B$ in which the samples are grouped into $B/M$ groups

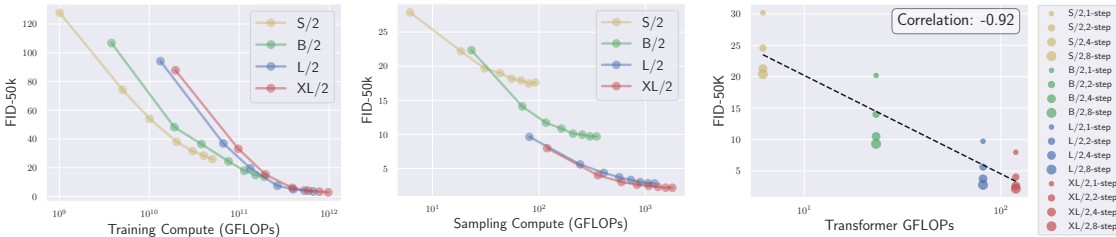

Figure 7. IMM scales with both training and inference compute, and exhibits strong correlation between model size and performance.

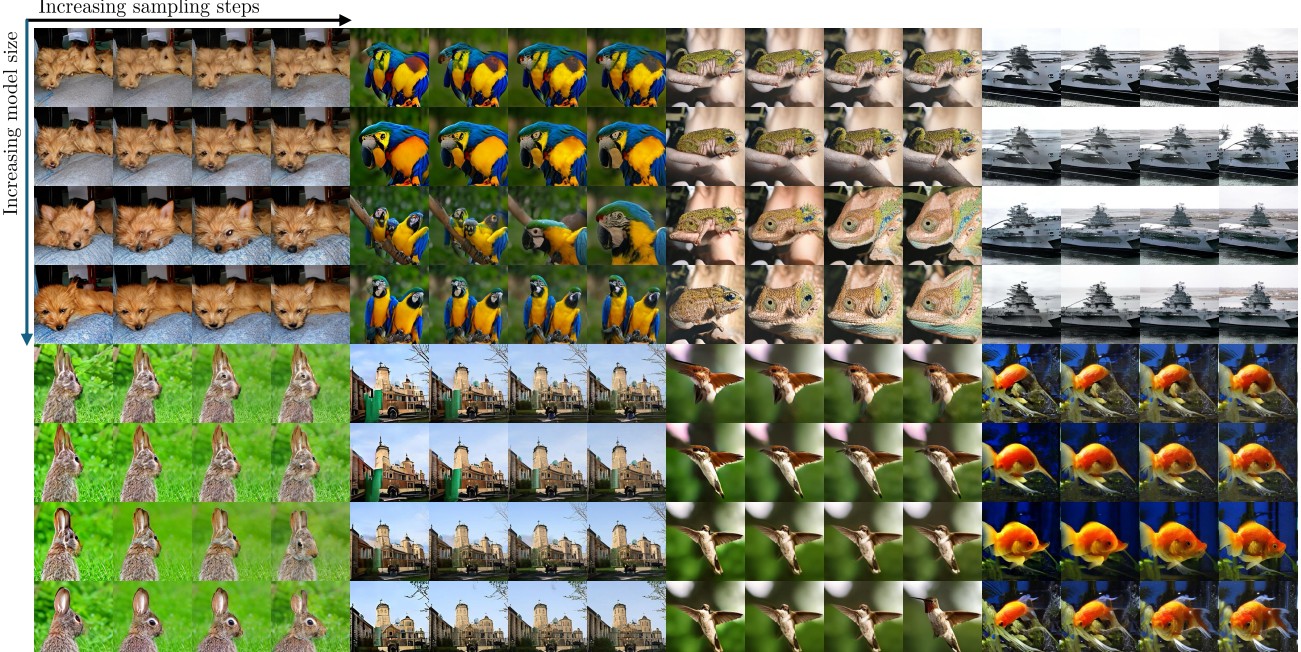

Figure 8. Sample visual quality increases with increase in both model size and sampling compute.

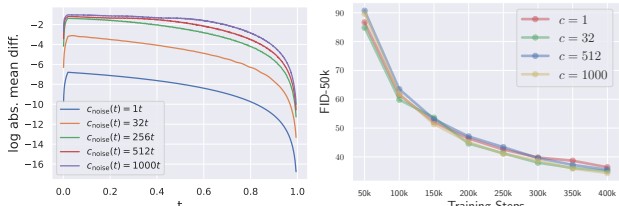

Figure 9. Log distance in embedding space for $c_{\text{noise}}(t) = ct$ (left). Similar ImageNet-256×256 convergence across different $c$ (right).

of $M$ where each group shares the same $t$. The larger $M$ means that fewer $t$'s are sampled. On the other hand, using extremely small numbers of particles, *e.g.* $M = 2$, leads to training instability and performance degradation, especially on a large scale with DiT architectures. We find that there exists a sweet spot where a few particles effectively help with training stability while further increasing $M$ slows down convergence (see Figure 5). We see that in ImageNet-256×256, training collapses when $M = 1$ (which is CM) and $M = 2$, and achieves lowest FID under the same computation budget with $M = 4$. We hypothesize $M < 4$ does not allow sufficient mixing between particles and larger $M$

means fewer $t$'s are sampled for each step, thus slowing convergence. A general rule of thumb is to use a *large enough* $M$ for stability, but not too large for slowed convergence.

**Noise embedding $c_{\text{noise}}(\cdot)$.** We plot in Figure 9 the log absolute mean difference of $t$ and $r(s, t)$ in the positional embedding space. Increasing $c$ increases distinguishability of nearby distributions. We also observe similar convergence on ImageNet-256×256 across different $c$, demonstrating the insensitivity of our framework w.r.t. noise function.

### 7.3. Sampling

We investigate different sampling settings for best performance. One-step sampling is performed by simple pushforward from $T$ to $\epsilon$ (concrete values in Appendix I.2). On CIFAR-10 we use 2 steps and set intermediate time $t_1$ such that $\eta_{t_1} = 1.4$, a choice we find to work well empirically. On ImageNet-256×256 we go beyond 2 steps and, for simplicity, investigate (1) uniform decrement in $t$ and (2) EDM (Karras et al., 2024) schedule (detailed in Appendix I.2). We plot FID of all sampler settings in Figure 6 with guidance weight $w = 1.5$. We find pushforward sam-

|  | id/cos | id/FM | sEDM/cos | sEDM/FM | eFM |
|---|---|---|---|---|---|
| CIFAR-10 | 3.77 | 3.45 | 2.39 | **2.10** | 2.53 |
| ImageNet-256×256 | 46.44 | 47.32 | 27.33 | 28.67 | **27.01** |

*Table 3.* FID results with different flow schedules and network parameterization.

plers with uniform schedule to work the best on ImageNet-256×256 and use this as our default setting for multi-step generation. Additionally, we concede that pushforward combined with restart samplers can achieve superior results. We leave such experiments to future works.

### 7.4. Scaling Behavior

Similar to diffusion models, IMM scale with training and inference compute as well as model size on ImageNet-256×256. We plot in Figure 7 FID vs. training and inference compute in GFLOPs and we find strong correlation between compute used and performance. We further visualize samples in Figure 8 with increasing model size, *i.e.* DiT-S, DiT-B, DiT-L, DiT-XL, and increasing inference steps, *i.e.* 1, 2, 4, 8 steps. The sample quality increases along both axes as larger transformers with more inference steps capture more complex distributions. This also explains that more compute can sometimes yield different visual content from the same initial noise as shown in the visual results.

### 7.5. Ablation Studies

All ablation studies are done with DDPM++ architecture for CIFAR-10 and DiT-B for ImageNet-256×256. FID comparisons use 2-step samplers by default.

**Flow schedules and parameterization.** We investigate all combinations of network parameterization and flow schedules: Simple-EDM + cosine (sEDM/cos), Simple-EDM + OT-FM (sEDM/FM), Euler-FM + OT-FM (eFM), Identity + cosine (id/cos), Identity + OT-FM (id/FM). Identity parameterization consistently fall behind other types of parameterization, which all show similar performance across datasets (see Table 3). We see that on smaller scale (CIFAR-10), sEDM/FM works the best but on larger scale (ImageNet-256×256), eFM works the best, indicating that OT-FM schedule and Euler paramaterization may be more scalable than other choices.

**Mapping function** $r(s, t)$**.** Our choices for ablation are (1) constant decrement in $\eta_t$, (2) constant decrement in $t$, (3) constant decrement in $\lambda_t = \log(\alpha_t^2/\sigma_t^2)$, (4) constant increment in $1/\eta_t$ (see Appendix C.6). For fair comparison, we choose the decrement gap so that the minimum $t - r(s, t)$ is $\approx 10^{-3}$ and use the same network parameterization. FID progression in Figure 11 show that (1) consistently outperforms other choices. We additionally ablate the mapping gap using $M = 4$ in (1). The constant decrement is in

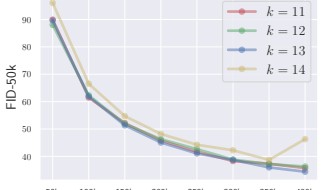

*Figure 10.* ImageNet-256×256 FID progression with different $t, r$ gap with $M = 4$.

|  | FID-50k |
|---|---|
| $w(s, t) = 1$ | 40.19 |
| + ELBO weight | 96.43 |
| + $\alpha_t$ | 33.44 |
| + $1/(\alpha_t^2 + \sigma_t^2)$ | **27.43** |

*Table 4.* Ablation of weighting function $w(s, t)$ on ImageNet-256×256.

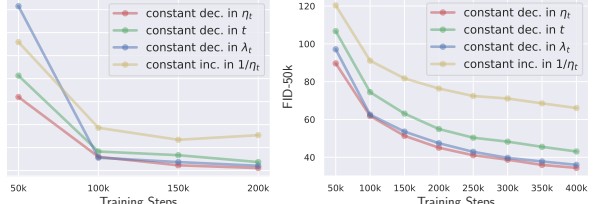

*Figure 11.* FID progression on different types of mapping function $r(s, t)$ for CIFAR-10 (left) and ImageNet-256×256 (right).

the form of $(\eta_{\max} - \eta_{\min})/2^k$ for an appropriately chosen $k$. We show in Figure 10 that the performance is relatively stable across $k \in \{11, 12, 13\}$ but experiences instability for $k = 14$. This suggests that, for a given particle number, there exists a largest $k$ for stable optimization.

**Weighting function.** In Table 4 we first ablate the weighting factors in three groups: (1) the VDM ELBO factors $\frac{1}{2}\sigma(b - \lambda_t)(-\frac{d}{dt}\lambda_t)$, (2) weighting $\alpha_t$ (*i.e.* when $a = 1$), and (3) weighting $1/(\alpha_t^2 + \sigma_t^2)$. We find it necessary to use $\alpha_t$ jointly with ELBO weighting because it converts $v$-pred network to a $\epsilon$-pred parameterization (see Appendix C.9), consistent with diffusion ELBO-objective. Factor $1/(\alpha_t^2 + \sigma_t^2)$ upweighting middle time-steps further boosts performance, a helpful practice also known for FM training (Esser et al., 2024). We leave additional study of the exponent $a$ to Appendix I.5 and find that $a = 2$ emphasizes optimizing the loss when $t$ is small while $a = 1$ more equally distributes weights to larger $t$. As a result, $a = 2$ achieves higher quality multi-step generation than $a = 1$.

## 8. Conclusion

We present Inductive Moment Matching, a framework that learns a few-step generative model from scratch. It trains by first leveraging self-consistent interpolants to interpolate between data and prior and then matching all moments of its own distribution interpolated to be closer to that of data. Our method guarantees convergence in distribution and generalizes many prior works. Our method also achieves state-of-the-art performance across benchmarks while achieving orders of magnitude faster inference. We hope it provides a new perspective on training few-step models from scratch and inspire a new generation of generative models.

# Impact Statement

This paper advances research in diffusion models and generative AI, which enable new creative possibilities and democratize content creation but also raise important considerations. Potential benefits include expanding artistic expression, assisting content creators, and generating synthetic data for research. However, we acknowledge challenges around potential misuse for deepfakes, copyright concerns, and impacts on creative industries. While our work aims to advance technical capabilities, we encourage ongoing discussion about responsible development and deployment of these technologies.

# Acknowledgement

We thank Wanqiao Xu, Bokui Shen, Connor Lin, and Samrath Sinha for additional technical discussions and helpful suggestions.

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

## A. Background: Properties of Stochastic Interpolants

We note some relevant properties of stochastic interpolants for our exposition.

**Boundary satisfaction.** For an interpolant distribution $q_t(\mathbf{x}_t|\mathbf{x}, \boldsymbol{\epsilon})$ defined in Albergo et al. (2023), and the marginal $q_t(\mathbf{x}_t)$ as defined in Eq. (2), we can check that $q_1(\mathbf{x}_1) = p(\mathbf{x}_1)$ and $q_0(\mathbf{x}_0) = q(\mathbf{x}_0)$ so that $\mathbf{x}_1 = \boldsymbol{\epsilon}$ and $\mathbf{x}_0 = \mathbf{x}$.

$$q_1(\mathbf{x}_1) = \iint q_1(\mathbf{x}_1|\mathbf{x}, \boldsymbol{\epsilon})q(\mathbf{x})p(\boldsymbol{\epsilon})\mathrm{d}\mathbf{x}\mathrm{d}\boldsymbol{\epsilon} \tag{15}$$

$$= \iint \delta(\mathbf{x}_1 - \boldsymbol{\epsilon})q(\mathbf{x})p(\boldsymbol{\epsilon})\mathrm{d}\mathbf{x}\mathrm{d}\boldsymbol{\epsilon} \tag{16}$$

$$= \iint q(\mathbf{x})p(\mathbf{x}_1)\mathrm{d}\mathbf{x} \tag{17}$$

$$= p(\mathbf{x}_1) \tag{18}$$

$$q_0(\mathbf{x}_0) = \iint q_0(\mathbf{x}_0|\mathbf{x}, \boldsymbol{\epsilon})q(\mathbf{x})p(\boldsymbol{\epsilon})\mathrm{d}\mathbf{x}\mathrm{d}\boldsymbol{\epsilon} \tag{19}$$

$$= \iint \delta(\mathbf{x}_0 - \mathbf{x})q(\mathbf{x})p(\boldsymbol{\epsilon})\mathrm{d}\mathbf{x}\mathrm{d}\boldsymbol{\epsilon} \tag{20}$$

$$= \iint q(\mathbf{x}_0)p(\boldsymbol{\epsilon})\mathrm{d}\mathbf{x} \tag{21}$$

$$= q(\mathbf{x}_0) \tag{22}$$

**Joint distribution.** The joint distribution of $\mathbf{x}$ and $\mathbf{x}_t$ is written as

$$q_t(\mathbf{x}, \mathbf{x}_t) = \int q_t(\mathbf{x}_t|\mathbf{x}, \boldsymbol{\epsilon})q(\mathbf{x})p(\boldsymbol{\epsilon})\mathrm{d}\boldsymbol{\epsilon} \tag{23}$$

**Indepedence of joint at $t = 1$.**

$$q_1(\mathbf{x}, \mathbf{x}_1) = \int q_1(\mathbf{x}_1|\mathbf{x}, \boldsymbol{\epsilon})q(\mathbf{x})p(\boldsymbol{\epsilon})\mathrm{d}\boldsymbol{\epsilon} \tag{24}$$

$$= \int \delta(\mathbf{x}_1 - \boldsymbol{\epsilon})q(\mathbf{x})p(\boldsymbol{\epsilon})\mathrm{d}\boldsymbol{\epsilon} \tag{25}$$

$$= q(\mathbf{x})p(\mathbf{x}_1) \tag{26}$$

$$= q(\mathbf{x})p(\boldsymbol{\epsilon}) \tag{27}$$

in which case $\mathbf{x}_1 = \boldsymbol{\epsilon}$.

## B. Theorems and Derivations

### B.1. Divergence Minimizer

**Lemma 3.** *Assuming marginal-preserving interpolant and metric $D(\cdot, \cdot)$, a minimizer $\theta^*$ of Eq. (7) exists, i.e. $p_{s|t}^{\theta^*}(\mathbf{x}|\mathbf{x}_t) = q_t(\mathbf{x}|\mathbf{x}_t)$, and the minimum is 0.*

*Proof.* We directly substitute $q_t(\mathbf{x}|\mathbf{x}_t)$ into the objective to check. First,

$$p_{s|t}^{\theta^*}(\mathbf{x}_s) = \iint q_{s|t}(\mathbf{x}_s|\mathbf{x}, \mathbf{x}_t)p_{s|t}^{\theta^*}(\mathbf{x}|\mathbf{x}_t)q_t(\mathbf{x}_t)\mathrm{d}\mathbf{x}_t\mathrm{d}\mathbf{x} \tag{28}$$

$$= \iint q_{s|t}(\mathbf{x}_s|\mathbf{x}, \mathbf{x}_t)q_t(\mathbf{x}|\mathbf{x}_t)q_t(\mathbf{x}_t)\mathrm{d}\mathbf{x}_t\mathrm{d}\mathbf{x} \tag{29}$$

$$\overset{(a)}{=} q_s(\mathbf{x}_s) \tag{30}$$

where $(a)$ is due to definition of marginal preservation. So the objective becomes

$$\mathbb{E}_{s,t}\left[w(s,t)D(q_s(\mathbf{x}_s), p_{s|t}^{\theta^*}(\mathbf{x}_s))\right] = \mathbb{E}_{s,t}\left[w(s,t)D(q_s(\mathbf{x}_s), q_s(\mathbf{x}_s))\right] = 0 \tag{31}$$

In general, the minimizer $q_t(\mathbf{x}|\mathbf{x}_t)$ exists. However, this does not show that the minimizer is unique. In fact, the minimizer is not unique in general because a deterministic minimizer can also exist under certain assumptions on the interpolant (see Appendix B.6). $\qquad\square$

**Failure Case without Marginal Preservation.** We additionally show that marginal-preservation property of the interpolant $q_{s|t}(\mathbf{x}_s|\mathbf{x}_t)$ is important for the naïve objective in Eq. (7) to attain 0 loss (Lemma 3). Consider the failure case below where the constructed interpolant is a generalized interpolant but not necessarily marginal-preserving. Then we show that there exists a $t$ such that $p_{s|t}^{\theta}(\mathbf{x}_s)$ can never reach $q_s(\mathbf{x}_s)$ regardless of $\theta$.

**Proposition 2** (Example Failure Case)**.** *Let* $q(\mathbf{x}) = \delta(\mathbf{x})$, $p(\boldsymbol{\epsilon}) = \delta(\boldsymbol{\epsilon} - 1)$, *and suppose an interpolant* $\mathbf{I}_{s|t}(\mathbf{x}, \mathbf{x}_t) = (1 - \frac{s}{t})\mathbf{x} + \frac{s}{t}\mathbf{x}_t$ *and* $\gamma_{s|t} = \frac{s}{t}\sqrt{1 - \frac{s}{t}}\mathbb{1}(t < 1)$, *then* $D(q_s(\mathbf{x}_s), p_{s|t}^{\theta}(\mathbf{x}_s)) > 0$ *for all* $0 < s < t < 1$ *regardless of the learned distribution* $p_{s|t}^{\theta}(\mathbf{x}|\mathbf{x}_t)$ *given any metric* $D(\cdot, \cdot)$.

*Proof.* This example first implies the learning target

$$q_s(\mathbf{x}_s) = \iint q_s(\mathbf{x}_s|\mathbf{x}, \boldsymbol{\epsilon})q(\mathbf{x})p(\boldsymbol{\epsilon})\mathrm{d}\mathbf{x}\mathrm{d}\boldsymbol{\epsilon} \tag{32}$$

$$= \iint \delta(\mathbf{x}_s - ((1-s)\mathbf{x} + s\boldsymbol{\epsilon}))\delta(\mathbf{x})\delta(\boldsymbol{\epsilon} - 1)\mathrm{d}\mathbf{x}\mathrm{d}\boldsymbol{\epsilon} \tag{33}$$

$$= \delta(\mathbf{x}_s - s) \tag{34}$$

is a delta distribution. However, we show that if we select any $t < 1$ and $0 < s < t$, $p_{s|t}^{\theta}(\mathbf{x}_s)$ can never be a delta distribution.

$$p_{s|t}^{\theta}(\mathbf{x}_s) = \iint q_{s|t}(\mathbf{x}_s|\mathbf{x}, \mathbf{x}_t)p_{s|t}^{\theta}(\mathbf{x}|\mathbf{x}_t)q_t(\mathbf{x}_t)\mathrm{d}\mathbf{x}_t\mathrm{d}\mathbf{x} \tag{35}$$

$$= \iint \mathcal{N}((1 - \frac{s}{t})\mathbf{x} + \frac{s}{t}\mathbf{x}_t, \frac{s^2}{t^2}(1 - \frac{s^2}{t^2})I)p_{s|t}^{\theta}(\mathbf{x}|\mathbf{x}_t)\delta(\mathbf{x}_t - t)\mathrm{d}\mathbf{x}_t\mathrm{d}\mathbf{x} \tag{36}$$

$$= \int \mathcal{N}((1 - \frac{s}{t})\mathbf{x} + s, \frac{s^2}{t^2}(1 - \frac{s^2}{t^2})I)p_{s|t}^{\theta}(\mathbf{x}|\mathbf{x}_t = t)\mathrm{d}\mathbf{x} \tag{37}$$

Now, we show the model distribution has non-zero variance under these choices of $t$ and $s$. Expectations are over $p_{s|t}^{\theta}(\mathbf{x}_s)$ or conditional interpolant $q_{s|t}(\mathbf{x}_s|\mathbf{x}, \mathbf{x}_t)$ for all equations below.

$$\mathrm{Var}(\mathbf{x}_s) = \mathbb{E}\left[\mathbf{x}_s^2\right] - \mathbb{E}\left[\mathbf{x}_s\right]^2 \tag{38}$$

$$= \iint \mathbb{E}\left[\mathbf{x}_s^2|\mathbf{x}, \mathbf{x}_t\right]p_{s|t}^{\theta}(\mathbf{x}|\mathbf{x}_t)q_t(\mathbf{x}_t)\mathrm{d}\mathbf{x}\mathrm{d}\mathbf{x}_t - \mathbb{E}\left[\mathbf{x}_s\right]^2 \tag{39}$$

$$= \iint \mathbb{E}\left[\mathbf{x}_s^2|\mathbf{x}, \mathbf{x}_t\right]p_{s|t}^{\theta}(\mathbf{x}|\mathbf{x}_t)q_t(\mathbf{x}_t)\mathrm{d}\mathbf{x}\mathrm{d}\mathbf{x}_t - \mathbb{E}\left[\mathbf{x}_s\right]^2 \tag{40}$$

$$= \iint \left[\mathrm{Var}(\mathbf{x}_s|\mathbf{x}, \mathbf{x}_t) + \mathbb{E}\left[\mathbf{x}_s|\mathbf{x}, \mathbf{x}_t\right]^2\right]p_{s|t}^{\theta}(\mathbf{x}|\mathbf{x}_t)q_t(\mathbf{x}_t)\mathrm{d}\mathbf{x}\mathrm{d}\mathbf{x}_t - 2\mathbb{E}\left[\mathbf{x}_s\right]^2 + \mathbb{E}\left[\mathbf{x}_s\right]^2 \tag{41}$$

$$= \iint \left[\mathrm{Var}(\mathbf{x}_s|\mathbf{x}, \mathbf{x}_t) + \mathbb{E}\left[\mathbf{x}_s|\mathbf{x}, \mathbf{x}_t\right]^2\right]p_{s|t}^{\theta}(\mathbf{x}|\mathbf{x}_t)q_t(\mathbf{x}_t)\mathrm{d}\mathbf{x}\mathrm{d}\mathbf{x}_t \tag{42}$$

$$\quad - \iint \mathbb{E}\left[\mathbf{x}_s\right]\mathbb{E}\left[\mathbf{x}_s|\mathbf{x}\mathbf{x}_t\right]p_{s|t}^{\theta}(\mathbf{x}|\mathbf{x}_t)q_t(\mathbf{x}_t)\mathrm{d}\mathbf{x}\mathrm{d}\mathbf{x}_t + \mathbb{E}\left[\mathbf{x}_s\right]^2$$

$$= \iint \underbrace{\left[\mathrm{Var}(\mathbf{x}_s|\mathbf{x}, \mathbf{x}_t) + \mathbb{E}\left[\mathbf{x}_s|\mathbf{x}, \mathbf{x}_t\right]^2 - \mathbb{E}\left[\mathbf{x}_s\right]\mathbb{E}\left[\mathbf{x}_s|\mathbf{x}, \mathbf{x}_t\right] + \mathbb{E}\left[\mathbf{x}_s\right]^2\right]}_{(a)}p_{s|t}^{\theta}(\mathbf{x}|\mathbf{x}_t)q_t(\mathbf{x}_t)\mathrm{d}\mathbf{x}\mathrm{d}\mathbf{x}_t \tag{43}$$

where $(a)$ can be simplified as

$$\text{Var}(\mathbf{x}_s|\mathbf{x}, \mathbf{x}_t) + \left(\mathbb{E}\left[\mathbf{x}_s|\mathbf{x}, \mathbf{x}_t\right]^2 - \mathbb{E}\left[\mathbf{x}_s\right]\right)^2 > 0 \tag{44}$$

because $\text{Var}(\mathbf{x}_s|\mathbf{x}, \mathbf{x}_t) > 0$ for all $0 < s < t < 1$ due to its non-zero Gaussian noise. Therefore, $\text{Var}(\mathbf{x}_s) > 0$, implying $p^\theta_{s|t}(\mathbf{x}_s)$ can never be a delta function regardless of model $p^\theta_{s|t}(\mathbf{x}|\mathbf{x}_t)$. A valid metric $D(\cdot, \cdot)$ over probability $p^\theta_{s|t}(\mathbf{x}_s)$ and $q_s(\mathbf{x}_s)$ implies

$$D(q_s(\mathbf{x}_s), p^\theta_{s|t}(\mathbf{x}_s)) = 0 \iff p^\theta_{s|t}(\mathbf{x}_s) = q_s(\mathbf{x}_s)$$

which means

$$p^\theta_{s|t}(\mathbf{x}_s) \neq q_s(\mathbf{x}_s) \implies D(q_s(\mathbf{x}_s), p^\theta_{s|t}(\mathbf{x}_s)) > 0$$

$\square$

## B.2. Boundary Satisfaction of Model Distribution

The operator output $p^\theta_{s|t}(\mathbf{x}_s)$ satisfies boundary condition.

**Lemma 4** (Boundary Condition). *For all $s \in [0, 1]$ and all $\theta$, the following boundary condition holds.*

$$q_s(\mathbf{x}_s) = p^\theta_{s|s}(\mathbf{x}_s) \tag{45}$$

*Proof.*

$$
\begin{aligned}
p^\theta_{s|s}(\mathbf{x}_s) &= \iint \underbrace{q_{s,s}(\mathbf{x}_s|\mathbf{x}, \bar{\mathbf{x}}_s)}_{\delta(\mathbf{x}_s - \bar{\mathbf{x}}_s)} p^\theta_{s|s}(\mathbf{x}|\bar{\mathbf{x}}_s) q_{s,1}(\bar{\mathbf{x}}_s) \mathrm{d}\bar{\mathbf{x}}_s \mathrm{d}\mathbf{x} \\
&= \int p^\theta_{s|s}(\mathbf{x}|\mathbf{x}_s) q_s(\mathbf{x}_s) \mathrm{d}\mathbf{x} \\
&= q_s(\mathbf{x}_s) \int p^\theta_{s|s}(\mathbf{x}|\mathbf{x}_s) \mathrm{d}\mathbf{x} \\
&= q_s(\mathbf{x}_s)
\end{aligned}
$$

$\square$

## B.3. Definition of Well-Conditioned $r(s, t)$

For simplicity, the mapping function $r(s, t)$ is well-conditioned if

$$r(s, t) = \max(s, t - \Delta(t)) \tag{46}$$

where $\Delta(t) \geq \epsilon > 0$ is a positive function such that $r(s, t)$ is increasing for $t \geq c_0(s)$ where $c_0(s)$ is the largest $t$ that is mapped to $s$. Formally, $c_0(s) = \sup\{t : r(s, t) = s\}$. For $t \geq c_0(s)$, the inverse w.r.t. $t$ exists, *i.e.* $r^{-1}(s, \cdot)$ and $r^{-1}(s, r(s, t)) = t$. All practical implementations follow this general form, and are detailed in Appendix C.6.

## B.4. Main Theorem

**Theorem 1.** *Assuming $r(s, t)$ is well-conditioned, the interpolant is marginal-preserving, and $\theta^*_n$ is a minimizer of Eq. (8) for each $n$ with infinite data and network capacity, for all $t \in [0, 1]$, $s \in [0, t]$,*

$$\lim_{n \to \infty} \text{MMD}^2(q_s(\mathbf{x}_s), p^{\theta^*_n}_{s|t}(\mathbf{x}_s)) = 0. \tag{9}$$

*Proof.* We prove by induction on sequence number $n$. First, $r(s, t)$ is well-conditioned by following the definition in Eq. (46). Furthermore, for notational convenience, we let $r^{-1}_n(s, \cdot) := r^{-1}(s, r^{-1}(s, r^{-1}(s, \dots)))$ be $n$ nested application of $r^{-1}(s, \cdot)$ on the second argument. Additionally, $r^{-1}_0(s, t) = t$.

**Base case:** $n = 1$. Given any $s \geq 0$, $r(s, u) = s$ for all $s < u \leq c_0(s)$, implying

$$\text{MMD}^2(p_{s|s}^{\theta_0}(\mathbf{x}_s), p_{s|u}^{\theta_1^*}(\mathbf{x}_s)) \overset{(a)}{=} \text{MMD}^2(q_s(\mathbf{x}_s), p_{s|u}^{\theta_1^*}(\mathbf{x}_s)) \overset{(b)}{=} 0 \tag{47}$$

for $u \leq c_0(s)$ where $(a)$ is implied by Lemma 4 and $(b)$ is implied by Lemma 3.

**Inductive assumption:** $n - 1$. We assume $p_{s|u}^{\theta_{n-1}^*}(\mathbf{x}_s) = q_s(\mathbf{x}_s)$ for all $s \leq u \leq r_{n-2}^{-1}(s, c_0(s))$.

We inspect the target distribution $p_{s|r(s,u)}^{\theta_{n-1}^*}(\mathbf{x}_s)$ in Eq. (8) if optimized on $s \leq u \leq r_{n-1}^{-1}(s, c_0(s))$. On this interval, we can apply $r(s, \cdot)$ to the inequality and get $s = r(s, s) \leq r(s, u) \leq r(s, r_{n-1}^{-1}(s, c_0(s))) = r_{n-2}^{-1}(s, c_0(s))$ since $r(s, \cdot)$ is increasing. And by inductive assumption $p_{s|r(s,u)}^{\theta_{n-1}^*}(\mathbf{x}_s) = q_s(\mathbf{x}_s)$ for $s \leq r(s, u) \leq r_{n-2}^{-1}(s, c_0(s))$, this implies minimizing

$$\mathbb{E}_{s,u}\left[w(s, u)\text{MMD}^2(p_{s|r(s,u)}^{\theta_{n-1}^*}(\mathbf{x}_s), p_{s|u}^{\theta_n}(\mathbf{x}_s))\right]$$

on $s \leq u \leq r_{n-1}^{-1}(s, c_0(s))$ is equivalent to minimizing

$$\mathbb{E}_{s,u}\left[w(s, u)\text{MMD}^2(q_s(\mathbf{x}_s), p_{s|u}^{\theta_n}(\mathbf{x}_s))\right] \tag{48}$$

for $s \leq u \leq r_{n-1}^{-1}(s, c_0(s))$. Lemma 3 implies that its minimum achieves $p_{s|u}^{\theta_n^*}(\mathbf{x}_s) = q_s(\mathbf{x}_s)$.

Lastly, taking $n \to \infty$ implies $\lim_{n\to\infty} r_n^{-1}(s, c_0(s)) = 1$ and thus the induction covers the entire $[s, 1]$ interval given each $s$. Therefore, $\lim_{n\to\infty} \text{MMD}(q_s(\mathbf{x}_s), p_{s|t}^{\theta_n^*}(\mathbf{x}_s)) = 0$ for all $0 \leq s \leq t \leq 1$. $\qquad\square$

## B.5. Self-Consistency Implies Marginal Preservation

Without assuming marginal preservation, it is important to define the marginal distribution of $\mathbf{x}_s$ under generalized interpolants $q_{s|t}(\mathbf{x}_s | \mathbf{x}, \mathbf{x}_t)$ as

$$q_{s|t}(\mathbf{x}_s) = \iint q_{s|t}(\mathbf{x}_s | \mathbf{x}, \mathbf{x}_t) q_t(\mathbf{x} | \mathbf{x}_t) q_t(\mathbf{x}_t) \mathrm{d}\mathbf{x}_t \mathrm{d}\mathbf{x} \tag{49}$$

and we show that with self-consistent interpolants, this distribution is invariant of $t$, *i.e.* $q_{s|t}(\mathbf{x}_s) = q_s(\mathbf{x}_s)$.

**Lemma 5.** *If the interpolant $q_{s|t}(\mathbf{x}_s | \mathbf{x}, \mathbf{x}_t)$ is self-consistent, the marginal distribution $q_{s|t}(\mathbf{x}_s)$ as defined in Eq. (49) satisfies $q_s(\mathbf{x}_s) = q_{s|t}(\mathbf{x}_s)$ for all $t \in [s, 1]$.*

*Proof.* For $t \in [s, 1]$,

$$q_{s|t}(\mathbf{x}_s) = \iint q_{s|t}(\mathbf{x}_s | \mathbf{x}, \mathbf{x}_t)) q_t(\mathbf{x} | \mathbf{x}_t) q_t(\mathbf{x}_t) \mathrm{d}\mathbf{x}_t \mathrm{d}\mathbf{x} \tag{50}$$

$$= \iint q_{s|t}(\mathbf{x}_s | \mathbf{x}, \mathbf{x}_t)) q_t(\mathbf{x}_t) \int \frac{q_t(\mathbf{x}_t | \mathbf{x}, \epsilon) q(\mathbf{x}) p(\epsilon)}{q_t(\mathbf{x}_t)} \mathrm{d}\epsilon \mathrm{d}\mathbf{x}_t \mathrm{d}\mathbf{x} \tag{51}$$

$$= \iint q_{s|t}(\mathbf{x}_s | \mathbf{x}, \mathbf{x}_t) q(\mathbf{x}) \int q_t(\mathbf{x}_t | \mathbf{x}, \epsilon) p(\epsilon) \mathrm{d}\epsilon \mathrm{d}\mathbf{x}_t \mathrm{d}\mathbf{x} \tag{52}$$

$$= \iint q(\mathbf{x}) p(\epsilon) \left( \int q_{s|t}(\mathbf{x}_s | \mathbf{x}, \mathbf{x}_t) q_t(\mathbf{x}_t | \mathbf{x}, \epsilon) \mathrm{d}\mathbf{x}_t \right) \mathrm{d}\epsilon \mathrm{d}\mathbf{x} \tag{53}$$

$$= \iint q(\mathbf{x}) p(\epsilon) \left( \int q_{s|t}(\mathbf{x}_s | \mathbf{x}, \mathbf{x}_t) q_{t,1}(\mathbf{x}_t | \mathbf{x}, \epsilon) \mathbf{x}_t \right) \mathrm{d}\epsilon \mathrm{d}\mathbf{x} \tag{54}$$

$$\overset{(a)}{=} \iint q_{s,1}(\mathbf{x}_s | \mathbf{x}, \epsilon) q(\mathbf{x}) p(\epsilon) \mathrm{d}\epsilon \mathrm{d}\mathbf{x} \tag{55}$$

$$\overset{(b)}{=} \iint q_s(\mathbf{x}_s | \mathbf{x}, \epsilon) q(\mathbf{x}) p(\epsilon) \mathrm{d}\epsilon \mathrm{d}\mathbf{x} \tag{56}$$

$$= q_s(\mathbf{x}_s) \tag{57}$$

where $(a)$ uses definition of self-consistent interpolants and $(b)$ uses definition of our generalized interpolant. $\qquad\square$

We show in Appendix C.1 that DDIM is an example self-consistent interpolant. Furthermore, DDPM posteior (Ho et al., 2020; Kingma et al., 2021) is also self-consistent (see Lemma 6).

### B.6. Existence of Deterministic Minimizer

We present the formal statement for the deterministic minimizer.

**Proposition 3.** *If for all $t \in [0,1]$, $s \in [0,t]$, $\gamma_{s|t} \equiv 0$, $\boldsymbol{I}_{s|t}(\mathbf{x}, \mathbf{x}_t)$ is invertible w.r.t. $\mathbf{x}$, and there exists $C_1 < \infty$ such that $\left\|\boldsymbol{I}_{t|1}(\mathbf{x}, \boldsymbol{\epsilon})\right\| < C_1 \|\mathbf{x} - \boldsymbol{\epsilon}\|$, then there exists a function $h_{s|t} : \mathbb{R}^D \to \mathbb{R}^D$ such that*

$$q_s(\mathbf{x}_s) = \iint q_{s|t}(\mathbf{x}_s|\mathbf{x}, \mathbf{x}_t)\delta(\mathbf{x} - h_{s|t}(\mathbf{x}_t))q_t(\mathbf{x}_t)\mathrm{d}\mathbf{x}\mathrm{d}\mathbf{x}_t. \tag{58}$$

*Proof.* Let $\boldsymbol{I}_{s|t}^{-1}(\cdot, \mathbf{x}_t)$ be the inverse of $\boldsymbol{I}_{s|t}$ w.r.t. $\mathbf{x}$ such that $\boldsymbol{I}_{s|t}^{-1}(\boldsymbol{I}_{s|t}(\mathbf{x}, \mathbf{x}_t), \mathbf{x}_t) = \mathbf{x}$ and $\boldsymbol{I}_{s|t}(\boldsymbol{I}_{s|t}^{-1}(\mathbf{y}, \mathbf{x}_t), \mathbf{x}_t) = \mathbf{y}$ for all $\mathbf{x}, \mathbf{x}_t, \mathbf{y} \in \mathbb{R}^D$. Since there exists $C_1 < \infty$ such that $\left\|\boldsymbol{I}_{t|1}(\mathbf{x}, \boldsymbol{\epsilon})\right\| < C_1 \|\mathbf{x} - \boldsymbol{\epsilon}\|$ for all $t \in [0,1]$, the PF-ODE of the original interpolant $\boldsymbol{I}_{t|1}(\mathbf{x}, \boldsymbol{\epsilon}) = \boldsymbol{I}_t(\mathbf{x}, \boldsymbol{\epsilon})$ exists for all $t \in [0,1]$ (Albergo et al., 2023). Then, for all $t \in [0,1]$, $s \in [0,t]$, we let

$$\hat{h}_{s|t}(\mathbf{x}_t) = \mathbf{x}_t + \int_t^s \mathbb{E}_{\mathbf{x}, \boldsymbol{\epsilon}}\left[\frac{\partial}{\partial u}\boldsymbol{I}_u(\mathbf{x}, \boldsymbol{\epsilon})\bigg|\mathbf{x}_u\right] \mathrm{d}u, \tag{59}$$

which pushes forward the measure $q_t(\mathbf{x}_t)$ to $q_s(\mathbf{x}_s)$. We define:

$$h_{s|t}(\mathbf{x}_t) = \boldsymbol{I}_{s|t}^{-1}(\hat{h}_{s|t}(\mathbf{x}_t), \mathbf{x}_t). \tag{60}$$

Then, since $\gamma_{s|t} \equiv 0$, $q_{s|t}(\mathbf{x}_s|\mathbf{x}, \mathbf{x}_t) = \delta(\mathbf{x}_s - \boldsymbol{I}_{s|t}(\mathbf{x}, \mathbf{x}_t))$ where $\mathbf{x} \sim \delta(\mathbf{x} - h_{s|t}(\mathbf{x}_t))$. Therefore,

$$\mathbf{x}_s = \boldsymbol{I}_{s|t}(h_{s|t}(\mathbf{x}_t), \mathbf{x}_t) = \boldsymbol{I}_{s|t}(\boldsymbol{I}_{s|t}^{-1}(\hat{h}_{s|t}(\mathbf{x}_t), \mathbf{x}_t), \mathbf{x}_t) = \hat{h}_{s|t}(\mathbf{x}_t) \tag{61}$$

whose marginal follows $q_s(\mathbf{x}_s)$ due to it being the result of PF-ODE trajectories starting from $q_t(\mathbf{x}_t)$. $\square$

Concretely, DDIM interpolant satisfies all of the deterministic assumption, the regularity condition, and the invertibility assumption because it is a linear function of $\mathbf{x}$ and $\mathbf{x}_t$. Therefore, any diffusion or FM schedule with DDIM interpolant will enjoy a deterministic minimizer $p_{s|t}^\theta(\mathbf{x}|\mathbf{x}_t)$.

## C. Analysis of Simplified Parameterization

### C.1. DDIM Interpolant

We check that DDIM interpolant is self-consistent. By definition, $q_{s|t}(\mathbf{x}_s|\mathbf{x}, \mathbf{x}_t) = \delta(\mathbf{x}_s - \mathrm{DDIM}(\mathbf{x}_t, \mathbf{x}, t, s))$. We check that for all $s \le r \le t$,

$$\int q_{s|r}(\mathbf{x}_s|\mathbf{x}, \mathbf{x}_r)q_{r|t}(\mathbf{x}_r|\mathbf{x}, \mathbf{x}_t)\mathrm{d}\mathbf{x}_r = \int \delta(\mathbf{x}_s - \mathrm{DDIM}(\mathbf{x}_r, \mathbf{x}, r, s))\delta(\mathbf{x}_r - \mathrm{DDIM}(\mathbf{x}_t, \mathbf{x}, t, r))\mathrm{d}\mathbf{x}_r$$
$$= \delta(\mathbf{x}_s - \mathrm{DDIM}(\mathrm{DDIM}(\mathbf{x}_t, \mathbf{x}, t, r), \mathbf{x}, r, s))$$

where

$$\mathrm{DDIM}(\mathrm{DDIM}(\mathbf{x}_t, \mathbf{x}, t, r), \mathbf{x}, r, s) = \alpha_s\mathbf{x} + (\sigma_s/\sigma_r)([\alpha_r\mathbf{x} + (\sigma_r/\sigma_t)(\mathbf{x}_t - \alpha_t\mathbf{x})] - \alpha_r\mathbf{x})$$
$$= \alpha_s\mathbf{x} + (\sigma_s/\sigma_r)(\sigma_r/\sigma_t)(\mathbf{x}_t - \alpha_t\mathbf{x})$$
$$= \alpha_s\mathbf{x} + (\sigma_s/\sigma_t)(\mathbf{x}_t - \alpha_t\mathbf{x})$$
$$= \mathrm{DDIM}(\mathbf{x}_t, \mathbf{x}, t, s)$$

Therefore, $\delta(\mathbf{x}_s - \mathrm{DDIM}(\mathrm{DDIM}(\mathbf{x}_t, \mathbf{x}, t, r), \mathbf{x}, r, s)) = \delta(\mathbf{x}_s - \mathrm{DDIM}(\mathbf{x}_t, \mathbf{x}, t, s))$. So DDIM is self-consistent.

It also implies a Gaussian forward process $q_t(\mathbf{x}_t|\mathbf{x}) = \mathcal{N}(\alpha_t\mathbf{x}, \sigma_t^2\sigma_d^2 I)$ as in diffusion models. By definition,

$$q_{t,1}(\mathbf{x}_t|\mathbf{x}) = q_t(\mathbf{x}_t|\mathbf{x}) = \int q_t(\mathbf{x}_t|\mathbf{x}, \boldsymbol{\epsilon})p(\boldsymbol{\epsilon})\mathrm{d}\boldsymbol{\epsilon}$$

so that $\mathbf{x}_t$ is a deterministic transform given $\mathbf{x}$ and $\boldsymbol{\epsilon}$, *i.e.*, $\mathbf{x}_t = \mathrm{DDIM}(\boldsymbol{\epsilon}, \mathbf{x}, t, 1) = \alpha_t\mathbf{x} + \sigma_t\boldsymbol{\epsilon}$, which implies $q_t(\mathbf{x}_t|\mathbf{x}) = \mathcal{N}(\alpha_t\mathbf{x}, \sigma_t^2\sigma_d^2 I)$.

## C.2. Reusing $\mathbf{x}_t$ for $\mathbf{x}_r$

We propose that instead of sampling $\mathbf{x}_r$ via forward flow $\alpha_r \mathbf{x} + \sigma_r \boldsymbol{\epsilon}$ we reuse $\mathbf{x}_t$ such that $\mathbf{x}_r = \mathrm{DDIM}(\mathbf{x}_t, \mathbf{x}, r, t)$ to reduce variance. In fact, for any self-consistent interpolant, one can reuse $\mathbf{x}_t$ via $\mathbf{x}_r \sim q_{r|t}(\mathbf{x}_r|\mathbf{x}, \mathbf{x}_t)$ and $\mathbf{x}_r$ will follow $q_r(\mathbf{x}_r)$ marginally. We check

$$q_r(\mathbf{x}_r) \stackrel{(a)}{=} q_{r|t}(\mathbf{x}_r) = \iint q_{r|t}(\mathbf{x}_r|\mathbf{x}, \mathbf{x}_t) q_t(\mathbf{x}|\mathbf{x}_t) q_t(\mathbf{x}_t) \mathrm{d}\mathbf{x}\mathrm{d}\mathbf{x}_t = \iint q_{r|t}(\mathbf{x}_r|\mathbf{x}, \mathbf{x}_t) \int q_t(\mathbf{x}_t|\mathbf{x}, \boldsymbol{\epsilon}) q(\mathbf{x}) p(\boldsymbol{\epsilon}) \mathrm{d}\boldsymbol{\epsilon}\mathrm{d}\mathbf{x}\mathrm{d}\mathbf{x}_t$$

where $(a)$ is due to Lemma 5. We can see that sampling $\mathbf{x}, \mathbf{x}_t$ first then $\mathbf{x}_r \sim q_{r|t}(\mathbf{x}_r|\mathbf{x}, \mathbf{x}_t)$ respects the marginal distribution $q_r(\mathbf{x}_r)$.

## C.3. Simplified Objective

We derive our simplified objective. Given MMD defined in Eq. (1), we write our objective as

$$\mathcal{L}_{\mathrm{IMM}}(\theta) = \mathbb{E}_{s,t}\left[ w(s,t) \left\| \mathbb{E}_{\mathbf{x}_t}[k(\boldsymbol{f}_{s,t}^\theta(\mathbf{x}_t), \cdot)] - \mathbb{E}_{\mathbf{x}_r}[k(\boldsymbol{f}_{s,r}^{\theta^-}(\mathbf{x}_r), \cdot)] \right\|_{\mathcal{H}}^2 \right] \tag{62}$$

$$\stackrel{(a)}{=} \mathbb{E}_{s,t}\left[ w(s,t) \left\| \mathbb{E}_{\mathbf{x}_t, \mathbf{x}_r}[k(\boldsymbol{f}_{s,t}^\theta(\mathbf{x}_t), \cdot) - k(\boldsymbol{f}_{s,r}^{\theta^-}(\mathbf{x}_r), \cdot)] \right\|_{\mathcal{H}}^2 \right] \tag{63}$$

$$= \mathbb{E}_{s,t}\left[ w(s,t) \left\langle \mathbb{E}_{\mathbf{x}_t, \mathbf{x}_r}[k(\boldsymbol{f}_{s,t}^\theta(\mathbf{x}_t), \cdot) - k(\boldsymbol{f}_{s,r}^{\theta^-}(\mathbf{x}_r), \cdot)], \mathbb{E}_{\mathbf{x}_t', \mathbf{x}_r'}[k(\boldsymbol{f}_{s,t}^\theta(\mathbf{x}_t'), \cdot) - k(\boldsymbol{f}_{s,r}^{\theta^-}(\mathbf{x}_r'), \cdot)] \right\rangle \right] \tag{64}$$

$$= \mathbb{E}_{s,t}\left[ w(s,t) \, \mathbb{E}_{\mathbf{x}_t, \mathbf{x}_r, \mathbf{x}_t', \mathbf{x}_r'}\left[ \langle k(\boldsymbol{f}_{s,t}^\theta(\mathbf{x}_t), \cdot), k(\boldsymbol{f}_{s,t}^\theta(\mathbf{x}_t'), \cdot) \rangle + \langle k(\boldsymbol{f}_{s,r}^{\theta^-}(\mathbf{x}_r), \cdot), k(\boldsymbol{f}_{s,r}^{\theta^-}(\mathbf{x}_r'), \cdot) \rangle \right. \right. \tag{65}$$
$$\left. \left. - \langle k(\boldsymbol{f}_{s,t}^\theta(\mathbf{x}_t), \cdot), k(\boldsymbol{f}_{s,r}^{\theta^-}(\mathbf{x}_r'), \cdot) \rangle - \langle k(\boldsymbol{f}_{s,t}^\theta(\mathbf{x}_t'), \cdot), k(\boldsymbol{f}_{s,r}^{\theta^-}(\mathbf{x}_r), \cdot) \rangle \right] \right]$$

$$= \mathbb{E}_{\mathbf{x}_t, \mathbf{x}_r, \mathbf{x}_t', \mathbf{x}_r', s, t}\left[ w(s,t) \left[ k(\boldsymbol{f}_{s,t}^\theta(\mathbf{x}_t), \boldsymbol{f}_{s,t}^\theta(\mathbf{x}_t')) + k(\boldsymbol{f}_{s,r}^{\theta^-}(\mathbf{x}_r), \boldsymbol{f}_{s,r}^{\theta^-}(\mathbf{x}_r')) \right. \right. \tag{66}$$
$$\left. \left. - k(\boldsymbol{f}_{s,t}^\theta(\mathbf{x}_t), \boldsymbol{f}_{s,r}^{\theta^-}(\mathbf{x}_r')) - k(\boldsymbol{f}_{s,t}^\theta(\mathbf{x}_t'), \boldsymbol{f}_{s,r}^{\theta^-}(\mathbf{x}_r)) \right] \right]$$

where $\langle \cdot, \cdot \rangle$ is in RKHS, $(a)$ is due to the correlation between $\mathbf{x}_r$ and $\mathbf{x}_t$ by re-using $\mathbf{x}_t$.

## C.4. Empirical Estimation

As proposed in Gretton et al. (2012), MMD is typically estimated with V-statistics by instantiating a matrix of size $M \times M$ such that a batch of $B$ $\mathbf{x}$ samples, $\{x^{(i)}\}_{i=1}^B$, is separated into groups of $M$ (assume $B$ is divisible by $M$) particles $\{x^{(i,j)}\}_{i=1,j=1}^{B/M, M}$ where each group share a $(s^i, r^i, t^i)$ sample. The Monte Carlo estimate becomes

$$\hat{\mathcal{L}}_{\mathrm{IMM}}(\theta) = \frac{1}{B/M} \sum_{i=1}^{B/M} w(s^i, t^i) \frac{1}{M^2} \sum_{j=1}^M \sum_{k=1}^M \left[ k(\boldsymbol{f}_{s^i,t^i}^\theta(x_{t^i}^{(i,j)}), \boldsymbol{f}_{s^i,t^i}^\theta(x_{t^i}^{(i,k)})) + k(\boldsymbol{f}_{s^i,r^i}^{\theta^-}(x_{r^i}^{(i,j)}), \boldsymbol{f}_{s^i,r^i}^{\theta^-}(x_{r^i}^{(i,k)})) \right. \tag{67}$$
$$\left. - 2k(\boldsymbol{f}_{s^i,t^i}^\theta(x_{t^i}^{(i,j)}), \boldsymbol{f}_{s^i,r^i}^{\theta^-}(x_{r^i}^{(i,k)})) \right]$$

**Computational efficiency.** First we note that regardless of $M$, we require only 2 model forward passes - one with and one without stop gradient, since the model takes in all $B$ instances together within the batch and produce outputs for the entire batch. For the calculation of our loss, although the need for $M$ particles may imply inefficient computation, the cost of this matrix computation is negligible in practice compared to the complexity of model forward pass. Suppose a forward pass for a single instance is $\mathcal{O}(K)$, then the total computation for computation loss for a batch of $B$ instances is $\mathcal{O}(BK) + \mathcal{O}(BM)$. Deep neural networks often has $K \gg M$, so $\mathcal{O}(BK)$ dominates the computation.

## C.5. Simplified Parameterization

We derive $\boldsymbol{f}_{s,t}^\theta(\mathbf{x}_t)$ for each parameterization, which now generally follows the form

$$\boldsymbol{f}_{s,t}^\theta(\mathbf{x}_t) = c_{\mathrm{skip}}(s,t)\mathbf{x}_t + c_{\mathrm{out}}(s,t)\boldsymbol{G}_\theta(c_{\mathrm{in}}(s,t)\mathbf{x}_t, c_{\mathrm{noise}}(s), c_{\mathrm{noise}}(t))$$

**Identity.** This is simply DDIM with $x$-prediction network.

$$\boldsymbol{f}_{s,t}^{\theta}(\mathbf{x}_t) = (\alpha_s - \frac{\sigma_s}{\sigma_t}\alpha_t)\boldsymbol{G}_\theta\left(\frac{\mathbf{x}_t}{\sigma_d\sqrt{\alpha_t^2 + \sigma_t^2}}, c_{\text{noise}}(s), c_{\text{noise}}(t)\right) + \frac{\sigma_s}{\sigma_t}\mathbf{x}_t$$

**Simple-EDM.**

$$\boldsymbol{f}_{s,t}^{\theta}(\mathbf{x}_t) = (\alpha_s - \frac{\sigma_s}{\sigma_t}\alpha_t)\left[\frac{\alpha_t}{\alpha_t^2 + \sigma_t^2}\mathbf{x}_t - \frac{\sigma_t}{\sqrt{\alpha_t^2 + \sigma_t^2}}\sigma_d\boldsymbol{G}_\theta\left(\frac{\mathbf{x}_t}{\sigma_d\sqrt{\alpha_t^2 + \sigma_t^2}}, c_{\text{noise}}(s), c_{\text{noise}}(t)\right)\right] + \frac{\sigma_s}{\sigma_t}\mathbf{x}_t$$

$$= \frac{\alpha_s\alpha_t + \sigma_s\sigma_t}{\alpha_t^2 + \sigma_t^2}\mathbf{x}_t - \frac{\alpha_s\sigma_t - \sigma_s\alpha_t}{\sqrt{\alpha_t^2 + \sigma_t^2}}\sigma_d\boldsymbol{G}_\theta\left(\frac{\mathbf{x}_t}{\sigma_d\sqrt{\alpha_t^2 + \sigma_t^2}}, c_{\text{noise}}(s), c_{\text{noise}}(t)\right)$$

When noise schedule is cosine, $\boldsymbol{f}_{s,t}^{\theta}(\mathbf{x}_t) = \cos(\frac{1}{2}\pi(t-s))\mathbf{x}_t - \sin(\frac{1}{2}\pi(t-s))\sigma_d\boldsymbol{G}_\theta\left(\frac{\mathbf{x}_t}{\sigma_d\sqrt{\alpha_t^2 + \sigma_t^2}}, c_{\text{noise}}(s), c_{\text{noise}}(t)\right)$.
And similar to Lu & Song (2024), we can show that predicting $\mathbf{x}_s = \text{DDIM}(\mathbf{x}_t, \mathbf{x}, s, t)$ with $\mathcal{L}_2$ loss is equivalent to $v$-prediction with cosine schedule.

$$w(s,t)\left\|\boldsymbol{f}_{s,t}^{\theta}(\mathbf{x}_t) - \left((\alpha_s - \frac{\sigma_s}{\sigma_t}\alpha_t)\mathbf{x} + \frac{\sigma_s}{\sigma_t}\mathbf{x}_t\right)\right\|^2$$

$$= w(s,t)\left\|\frac{\alpha_s\alpha_t + \sigma_s\sigma_t}{\alpha_t^2 + \sigma_t^2}\mathbf{x}_t - \frac{\alpha_s\sigma_t - \sigma_s\alpha_t}{\sqrt{\alpha_t^2 + \sigma_t^2}}\sigma_d\boldsymbol{G}_\theta\left(\frac{\mathbf{x}_t}{\sigma_d\sqrt{\alpha_t^2 + \sigma_t^2}}, c_{\text{noise}}(s), c_{\text{noise}}(t)\right) - \left((\alpha_s - \frac{\sigma_s}{\sigma_t}\alpha_t)\mathbf{x} + \frac{\sigma_s}{\sigma_t}\mathbf{x}_t\right)\right\|^2$$

$$= \tilde{w}(s,t)\left\|\sigma_d\boldsymbol{G}_\theta\left(\frac{\mathbf{x}_t}{\sigma_d\sqrt{\alpha_t^2 + \sigma_t^2}}, c_{\text{noise}}(s), c_{\text{noise}}(t)\right) - \underbrace{\left(-\frac{\sqrt{\alpha_t^2 + \sigma_t^2}}{\alpha_s\sigma_t - \sigma_s\alpha_t}\right)\left((\alpha_s - \frac{\sigma_s}{\sigma_t}\alpha_t)\mathbf{x} + \frac{\sigma_s}{\sigma_t}\mathbf{x}_t - \frac{\alpha_s\alpha_t + \sigma_s\sigma_t}{\alpha_t^2 + \sigma_t^2}\mathbf{x}_t\right)}_{\boldsymbol{G}_{\text{target}}}\right\|^2$$

where

$$\boldsymbol{G}_{\text{target}} = \left(-\frac{\sqrt{\alpha_t^2 + \sigma_t^2}}{\alpha_s\sigma_t - \sigma_s\alpha_t}\right)\left((\alpha_s - \frac{\sigma_s}{\sigma_t}\alpha_t)\mathbf{x} + \frac{\sigma_s}{\sigma_t}\mathbf{x}_t - \frac{\alpha_s\alpha_t + \sigma_s\sigma_t}{\alpha_t^2 + \sigma_t^2}\mathbf{x}_t\right)$$

$$= \left(-\frac{\sqrt{\alpha_t^2 + \sigma_t^2}}{\alpha_s\sigma_t - \sigma_s\alpha_t}\right)\left((\alpha_s - \frac{\sigma_s}{\sigma_t}\alpha_t)\mathbf{x} + \frac{\sigma_s\alpha_t^2 + \cancel{\sigma_s\sigma_t^2} - \alpha_s\alpha_t\sigma_t - \cancel{\sigma_s\sigma_t^2}}{\sigma_t(\alpha_t^2 + \sigma_t^2)}(\alpha_t\mathbf{x} + \sigma_t\boldsymbol{\epsilon})\right)$$

$$= \left(-\frac{\sqrt{\alpha_t^2 + \sigma_t^2}}{\alpha_s\sigma_t - \sigma_s\alpha_t}\right)\left(\frac{(\alpha_s\sigma_t - \sigma_s\alpha_t)(\alpha_t^2 + \sigma_t^2) + \sigma_s\alpha_t^3 - \alpha_s\alpha_t^2\sigma_t}{\sigma_t(\alpha_t^2 + \sigma_t^2)}\mathbf{x} + \frac{\sigma_s\alpha_t^2 - \alpha_s\alpha_t\sigma_t}{\alpha_t^2 + \sigma_t^2}\boldsymbol{\epsilon}\right)$$

$$= \left(-\frac{\sqrt{\alpha_t^2 + \sigma_t^2}}{\alpha_s\sigma_t - \sigma_s\alpha_t}\right)\left(\frac{(\alpha_s\sigma_t - \sigma_s\alpha_t)(\alpha_t^2 + \sigma_t^2) - (\alpha_s\sigma_t - \sigma_s\alpha_t)\alpha_t^2}{\alpha_t^2 + \sigma_t^2}\mathbf{x} - \frac{(\alpha_s\sigma_t - \sigma_s\alpha_t)\alpha_t}{\sigma_t(\alpha_t^2 + \sigma_t^2)}\boldsymbol{\epsilon}\right)$$

$$= \frac{\alpha_t\boldsymbol{\epsilon} - \sigma_t\mathbf{x}}{\sqrt{\alpha_t^2 + \sigma_t^2}}$$

This reduces to $v$-target if cosine schedule is used, and it deviates from $v$-target if FM schedule is used instead.

**Euler-FM.** We assume OT-FM schedule.

$$\boldsymbol{f}_{s,t}^{\theta}(\mathbf{x}_t) = ((1-s) - \frac{s}{t}(1-t))\left[\mathbf{x}_t - t\sigma_d\boldsymbol{G}_\theta\left(\frac{\mathbf{x}_t}{\sigma_d\sqrt{\alpha_t^2 + \sigma_t^2}}, c_{\mathrm{noise}}(s), c_{\mathrm{noise}}(t)\right)\right] + \frac{s}{t}\mathbf{x}_t$$

$$= ((1-\frac{s}{t})\left[\mathbf{x}_t - t\sigma_d\boldsymbol{G}_\theta\left(\frac{\mathbf{x}_t}{\sigma_d\sqrt{\alpha_t^2 + \sigma_t^2}}, c_{\mathrm{noise}}(s), c_{\mathrm{noise}}(t)\right)\right] + \frac{s}{t}\mathbf{x}_t$$

$$= \mathbf{x}_t - (t-s)\sigma_d\boldsymbol{G}_\theta\left(\frac{\mathbf{x}_t}{\sigma_d\sqrt{\alpha_t^2 + \sigma_t^2}}, c_{\mathrm{noise}}(s), c_{\mathrm{noise}}(t)\right)$$

This results in Euler ODE from $\mathbf{x}_t$ to $\mathbf{x}_s$. We also show that the network output reduces to $v$-prediction if matched with $\mathbf{x}_s = \mathrm{DDIM}(\mathbf{x}_t, \mathbf{x}, s, t)$. To see this,

$$w(s,t)\left\|\boldsymbol{f}_{s,t}^{\theta}(\mathbf{x}_t) - \left(((1-s) - \frac{s}{t}(1-t))\mathbf{x} + \frac{s}{t}\mathbf{x}_t\right)\right\|^2$$

$$= w(s,t)\left\|\boldsymbol{f}_{s,t}^{\theta}(\mathbf{x}_t) - \left((1-\frac{s}{t})\mathbf{x} + \frac{s}{t}\mathbf{x}_t\right)\right\|^2$$

$$= w(s,t)\left\|\mathbf{x}_t - (t-s)\sigma_d\boldsymbol{G}_\theta\left(\frac{\mathbf{x}_t}{\sigma_d\sqrt{\alpha_t^2 + \sigma_t^2}}, c_{\mathrm{noise}}(s), c_{\mathrm{noise}}(t)\right) - \left((1-\frac{s}{t})\mathbf{x} + \frac{s}{t}\mathbf{x}_t\right)\right\|^2$$

$$= \tilde{w}(s,t)\left\|\sigma_d\boldsymbol{G}_\theta\left(\frac{\mathbf{x}_t}{\sigma_d\sqrt{\alpha_t^2 + \sigma_t^2}}, c_{\mathrm{noise}}(s), c_{\mathrm{noise}}(t)\right) - \left(-\frac{1}{(t-s)}\right)\left((1-\frac{s}{t})\mathbf{x} + (\frac{s}{t} - 1)\mathbf{x}_t\right)\right\|^2$$

$$= \tilde{w}(s,t)\left\|\sigma_d\boldsymbol{G}_\theta\left(\frac{\mathbf{x}_t}{\sigma_d\sqrt{\alpha_t^2 + \sigma_t^2}}, c_{\mathrm{noise}}(s), c_{\mathrm{noise}}(t)\right) - \underbrace{\left(-\frac{1}{(t-s)}\right)\left((1-\frac{s}{t})\mathbf{x} + (\frac{s}{t} - 1)\mathbf{x}_t\right)}_{\boldsymbol{G}_{\mathrm{target}}}\right\|^2$$

where

$$\boldsymbol{G}_{\mathrm{target}} = \left(-\frac{1}{(t-s)}\right)\left((1-\frac{s}{t})\mathbf{x} + (\frac{s}{t} - 1)((1-t)\mathbf{x} + t\boldsymbol{\epsilon})\right)$$

$$= \left(-\frac{1}{(t-s)}\right)\left((1-\frac{s}{t})\mathbf{x} + (\frac{s}{t} - s - 1 + t)\mathbf{x} + (s-t)\boldsymbol{\epsilon}\right)$$

$$= \left(-\frac{1}{(t-s)}\right)((t-s)\mathbf{x} + (s-t)\boldsymbol{\epsilon})$$

$$= \boldsymbol{\epsilon} - \mathbf{x}$$

which is $v$-target under OT-FM schedule. This parameterization naturally allows zero-SNR sampling and satisfies boundary condition at $s = 0$, similar to Simple-EDM above. This is not true for Identity parametrization using $\boldsymbol{G}_\theta$ as it satisfies boundary condition only at $s > 0$.

### C.6. Mapping Function $r(s,t)$

We discuss below the concrete choices for $r(s,t)$. We use a constant decrement $\epsilon > 0$ in different spaces.

**Constant decrement in** $\eta(t) := \eta_t = \sigma_t/\alpha_t$. This is the choice that we find to work better than other choices in practice. First, let its inverse be $\eta^{-1}(\cdot)$,

$$r(s,t) = \max\left(s, \eta^{-1}(\eta(t) - \epsilon)\right)$$

We choose $\epsilon = (\eta_{\max} - \eta_{\min})/2^k$ for some $k$. We generally choose $\eta_{\max} \approx 160$ and $\eta_{\min} \approx 0$. We find $k = \{10, \ldots, 15\}$ works well enough depending on datasets.

**Constant decrement in $t$.**

$$r(s, t) = \max\left(s, t - \epsilon\right)$$

We choose $\epsilon = (T - \epsilon)/2^k$.

**Constant decrement in $\lambda(t) := \text{log-SNR}_t = 2\log(\alpha_t/\sigma_t)$.**

Let its inverse be $\lambda^{-1}(\cdot)$, then

$$r(s, t) = \max\left(s, \lambda^{-1}(\lambda(t) - \epsilon))\right)$$

We choose $\epsilon = (\lambda_{\max} - \lambda_{\min})/2^k$. This choice comes close to the first choice, but we refrain from this because $r(s, t)$ becomes close to $t$ both when $t \approx 0$ and $t \approx 1$ instead of just $t \approx 1$. This gives more chances for training instability than the first choice.

**Constant increment in $1/\eta(t)$.**

$$r(s, t) = \max\left(s, \eta^{-1}\left(\frac{1}{1/\eta(t) + \epsilon}\right)\right)$$

We choose $\epsilon = (1/\eta(t)_{\min} - 1/\eta(t)_{\max})/2^k$.

### C.7. Time Distribution $p(s, t)$

In all cases we choose $p(t) = \mathcal{U}(\epsilon, T)$ and $p(s|t) = \mathcal{U}(\epsilon, t)$ for some $\epsilon \geq 0$ and $T \leq 1$. The decision for time distribution is coupled with $r(s, t)$. We list the constraints on $p(s, t)$ for each $r(s, t)$ choice below.

**Constant decrement in $\eta(t)$.** We need to choose $T < 1$ because, for example, assuming OT-FM schedule, $\eta_t = t/(1 - t)$, one can observe that constant decrement in $\eta_t$ when $t \approx 1$ results in $r(s, t)$ that is too close to $t$ due to $\eta_t$'s exploding gradient around 1. We need to define $T < 1$ such that $r(s, T)$ is not too close to $T$ for $s$ reasonably far away. With $\eta_{\max} \approx 160$, we can choose $T = 0.994$ for OT-FM and $T = 0.996$ for VP-diffusion.

**Constant decrement in $t$.** No constraints needed. $T = 1$, $\epsilon = 0$.

**Constant decrement in $\lambda_t$.** One can similarly observe exploding gradient causing $r(s, t)$ to be too close to $t$ at both $t \approx 0$ and $t \approx 1$, so we can choose $\epsilon > 0$, *e.g.* 0.001, in addition to choosing $T = 0.994$ for OT-FM and $T = 0.996$ for VP-diffusion.

**Constant increment in $1/\eta_t$.** This experience exploding gradient for $t \approx 0$, so we require $\epsilon > 0$, *e.g.* 0.005. And $T = 1$.

### C.8. Kernel Function

For our Laplace kernel $k(x, y) = \exp(-\tilde{w}(s, t)\max(\|x - y\|_2, \epsilon)/D)$, we let $\epsilon > 0$ be a reasonably small constant, *e.g.* $10^{-8}$. Looking at its gradient w.r.t. $x$,

$$\nabla_x e^{-\tilde{w}(s,t)\max(\|x-y\|_2, \epsilon)/D} = \begin{cases} -\frac{\tilde{w}(s,t)}{D} e^{-\tilde{w}(s,t)\|x-y\|_2/D} \frac{x-y}{\|x-y\|_2}, & \text{if } \|x - y\|_2 > \epsilon \\ 0, & \text{otherwise} \end{cases}$$

one can notice that the gradient is self-normalized to be a unit vector, which is helpful in practice. In comparison, the gradient of RBF kernel of the form $k(x, y) = \exp\left(-\frac{1}{2}\tilde{w}(s, t)\|x - y\|^2/D\right)$,

$$\nabla_x e^{-\tilde{w}(s,t)\|x-y\|^2/D} = -\frac{\tilde{w}(s,t)}{D} e^{-\tilde{w}(s,t)\frac{1}{2}\|x-y\|^2/D}(x - y) \tag{68}$$

whose magnitude can vary a lot depending on how far $x$ is from $y$.

For $\tilde{w}(s, t)$, we find it helpful to write out the $\mathcal{L}_2$ loss between the arguments. For simplicity, we denote $\hat{G}_\theta(\mathbf{x}_t, s, t) =$

$$\boldsymbol{G}_\theta\left(\frac{\mathbf{x}_t}{\sigma_d\sqrt{\alpha_t^2+\sigma_t^2}}, c_{\text{noise}}(s), c_{\text{noise}}(t)\right)$$

$$\tilde{w}(s,t)\left\|\boldsymbol{f}_{s,t}^\theta(\mathbf{x}_t) - \boldsymbol{f}_{s,r}^{\theta^-}(\mathbf{x}_r')\right\|_2$$

$$= \tilde{w}(s,t)\left\|c_{\text{skip}}(s,t)\mathbf{x}_t + c_{\text{out}}(s,t)\hat{\boldsymbol{G}}_\theta(\mathbf{x}_t,s,t) - \left(c_{\text{skip}}(s,r)\mathbf{x}_r' + c_{\text{out}}(s,r)\hat{\boldsymbol{G}}(\mathbf{x}_r',s,r)\right)\right\|_2$$

$$= \tilde{w}(s,t)c_{\text{out}}(s,t)\left\|\hat{\boldsymbol{G}}_\theta(\mathbf{x}_t,s,t) - \frac{1}{c_{\text{out}}(s,t)}\left(c_{\text{skip}}(s,r)\mathbf{x}_r' + c_{\text{out}}(s,r)\hat{\boldsymbol{G}}(\mathbf{x}_r',s,r) - c_{\text{skip}}(s,t)\mathbf{x}_t\right)\right\|_2$$

We simply set $\tilde{w}(s,t) = 1/c_{\text{out}}(s,t)$ for the overall weighting to be 1. This allows invariance of magnitude of kernels w.r.t. $t$.

### C.9. Weighting Function $w(s,t)$

To review VDM (Kingma et al., 2021), the negative ELBO loss for diffusion model is

$$\mathcal{L}_{\text{ELBO}}(\theta) = \frac{1}{2}\mathbb{E}_{\mathbf{x},\boldsymbol{\epsilon},t}\left[\left(-\frac{\text{d}}{\text{d}t}\lambda_t\right)\|\boldsymbol{\epsilon}_\theta(\mathbf{x}_t,t) - \boldsymbol{\epsilon}\|^2\right] \tag{69}$$

where $\boldsymbol{\epsilon}_\theta$ is the noise-prediction network and $\lambda_t = \log\text{-SNR}_t$. The weighted-ELBO loss proposed in Kingma & Gao (2024) introduces an additional weighting function $w(t)$ monotonically increasing in $t$ (monotonically decreasing in $\log\text{-SNR}_t$) understood as a form of data augmentation. Specifically, they use sigmoid as the function such that the weighted ELBO is written as

$$\mathcal{L}_{w\text{-ELBO}}(\theta) = \frac{1}{2}\mathbb{E}_{\mathbf{x},\boldsymbol{\epsilon},t}\left[\sigma(b - \lambda_t)\left(-\frac{\text{d}}{\text{d}t}\lambda_t\right)\|\boldsymbol{\epsilon}_\theta(\mathbf{x}_t,t) - \boldsymbol{\epsilon}\|^2\right] \tag{70}$$

where $\sigma(\cdot)$ is sigmoid function.

The $\alpha_t$ is tailored towards the Simple-EDM and Euler-FM parameterization as we have shown in Appendix C.5 that the networks $\sigma_d\boldsymbol{G}_\theta$ amounts to $v$-prediction in cosine and OT-FM schedules. Notice that ELBO diffusion loss matches $\boldsymbol{\epsilon}$ instead of $\mathbf{v}$. Inspecting the gradient of Laplace kernel, we have (again, for simplicity we let $\hat{\boldsymbol{G}}_\theta(\mathbf{x}_t,s,t) = \boldsymbol{G}_\theta(\frac{\mathbf{x}_t}{\sigma_d\sqrt{\alpha_t^2+\sigma_t^2}}, c_{\text{noise}}(s), c_{\text{noise}}(t)))$

$$\frac{\partial}{\partial\theta}e^{-\tilde{w}(s,t)\left\|\boldsymbol{f}_{s,t}^\theta(\mathbf{x}_t) - \boldsymbol{f}_{s,r}^{\theta^-}(\mathbf{x}_r)\right\|_2/D}$$

$$= -\frac{\tilde{w}(s,t)}{D}e^{-\tilde{w}(s,t)\left\|\boldsymbol{f}_{s,t}^\theta(\mathbf{x}_t) - \boldsymbol{f}_{s,r}^{\theta^-}(\mathbf{x}_r)\right\|_2/D}\frac{\boldsymbol{f}_{s,t}^\theta(\mathbf{x}_t) - \boldsymbol{f}_{s,r}^{\theta^-}(\mathbf{x}_r)}{\left\|\boldsymbol{f}_{s,t}^\theta(\mathbf{x}_t) - \boldsymbol{f}_{s,r}^{\theta^-}(\mathbf{x}_r)\right\|_2}\frac{\partial}{\partial\theta}\boldsymbol{f}_{s,t}^\theta(\mathbf{x}_t)$$

$$= -\frac{\tilde{w}(s,t)}{D}e^{-\tilde{w}(s,t)\left\|\boldsymbol{f}_{s,t}^\theta(\mathbf{x}_t) - \boldsymbol{f}_{s,r}^{\theta^-}(\mathbf{x}_r)\right\|_2/D}\frac{\hat{\boldsymbol{G}}_\theta(\mathbf{x}_t,s,t) - \hat{\boldsymbol{G}}_{\text{target}}}{\left\|\hat{\boldsymbol{G}}_\theta(\mathbf{x}_t,s,t) - \hat{\boldsymbol{G}}_{\text{target}}\right\|_2}\frac{\partial}{\partial\theta}\boldsymbol{f}_{s,t}^\theta(\mathbf{x}_t)$$

for some constant $\hat{\boldsymbol{G}}_{\text{target}}$. We can see that gradient $\frac{\partial}{\partial\theta}\boldsymbol{f}_{s,t}^\theta(\mathbf{x}_t)$ is guided by vector $\hat{\boldsymbol{G}}_\theta(\mathbf{x}_t,s,t) - \hat{\boldsymbol{G}}_{\text{target}}$. Assuming $\hat{\boldsymbol{G}}_\theta(\mathbf{x}_t,s,t)$ is $v$-prediction, as is the case for Simple-EDM parameterization with cosine schedule and Euler-FM parameterization with OT-FM schedule, we can reparameterize $v$- to $\epsilon$-prediction with $\boldsymbol{\epsilon}_\theta$ as the new parameterization. We omit arguments to network for simplicity.

We show below that for both cases $\boldsymbol{\epsilon}_\theta - \boldsymbol{\epsilon}_{\text{target}} = \alpha_t(\hat{\boldsymbol{G}}_\theta - \hat{\boldsymbol{G}}_{\text{target}})$ for some constants $\boldsymbol{\epsilon}_{\text{target}}$ and $\hat{\boldsymbol{G}}_{\text{target}}$. For Simple-EDM, we know $x$-prediction from $v$-prediction parameterization (Salimans & Ho, 2022), $\mathbf{x}_\theta = \alpha_t\mathbf{x}_t - \sigma_t\hat{\boldsymbol{G}}_\theta$, and we also know

$x$-prediction from $\epsilon$-prediction, $\mathbf{x}_\theta = (\mathbf{x}_t - \sigma_t \epsilon_\theta)/\alpha_t$. We have

$$\frac{\mathbf{x}_t - \sigma_t \epsilon_\theta}{\alpha_t} = \alpha_t \mathbf{x}_t - \sigma_t \hat{\boldsymbol{G}}_\theta \tag{71}$$

$$\iff \mathbf{x}_t - \sigma_t \epsilon_\theta = \alpha_t^2 \mathbf{x}_t - \alpha_t \sigma_t \hat{\boldsymbol{G}}_\theta \tag{72}$$

$$\iff (1 - \alpha_t^2)\mathbf{x}_t + \alpha_t \sigma_t \hat{\boldsymbol{G}}_\theta = \sigma_t \epsilon_\theta \tag{73}$$

$$\iff \sigma_t^2 \mathbf{x}_t + \alpha_t \sigma_t \hat{\boldsymbol{G}}_\theta = \sigma_t \epsilon_\theta \tag{74}$$

$$\iff \epsilon_\theta = \sigma_t \mathbf{x}_t + \alpha_t \hat{\boldsymbol{G}}_\theta \tag{75}$$

$$\iff \epsilon_\theta - \epsilon_{\text{target}} = \sigma_t \mathbf{x}_t + \alpha_t \hat{\boldsymbol{G}}_\theta - \left(\sigma_t \mathbf{x}_t + \alpha_t \hat{\boldsymbol{G}}_{\text{target}}\right) \tag{76}$$

$$\iff \epsilon_\theta - \epsilon_{\text{target}} = \alpha_t(\hat{\boldsymbol{G}}_\theta - \hat{\boldsymbol{G}}_{\text{target}}) \tag{77}$$

For Euler-FM, we know $x$-prediction from $v$-prediction parameterization, $\mathbf{x}_\theta = \mathbf{x}_t - t\hat{\boldsymbol{G}}_\theta$ and we also know $x$-prediction from $\epsilon$-prediction, $\mathbf{x}_\theta = (\mathbf{x}_t - t\epsilon_\theta)/(1 - t)$. We have

$$\frac{\mathbf{x}_t - t\epsilon_\theta}{1 - t} = \mathbf{x}_t - t\hat{\boldsymbol{G}}_\theta \tag{78}$$

$$\iff \mathbf{x}_t - t\epsilon_\theta = (1 - t)\mathbf{x}_t - t(1 - t)\hat{\boldsymbol{G}}_\theta \tag{79}$$

$$\iff t\mathbf{x}_t + t(1 - t)\hat{\boldsymbol{G}}_\theta = t\epsilon_\theta \tag{80}$$

$$\iff \epsilon_\theta = \mathbf{x}_t + (1 - t)\hat{\boldsymbol{G}}_\theta \tag{81}$$

$$\iff \epsilon_\theta = \mathbf{x}_t + \alpha_t \hat{\boldsymbol{G}}_\theta \tag{82}$$

$$\iff \epsilon_\theta - \epsilon_{\text{target}} = \mathbf{x}_t + \alpha_t \hat{\boldsymbol{G}}_\theta - \left(\mathbf{x}_t + \alpha_t \hat{\boldsymbol{G}}_{\text{target}}\right) \tag{83}$$

$$\iff \epsilon_\theta - \epsilon_{\text{target}} = \alpha_t(\hat{\boldsymbol{G}}_\theta - \hat{\boldsymbol{G}}_{\text{target}}) \tag{84}$$

In both cases, $(\hat{\boldsymbol{G}}_\theta(\mathbf{x}_t, s, t) - \hat{\boldsymbol{G}}_{\text{target}})$ can be rewritten to $(\epsilon_\theta(\mathbf{x}_t, s, t) - \epsilon_{\text{target}})$ by multiplying a factor $\alpha_t$, and the guidance vector now matches that of the ELBO-diffusion loss. Therefore, we are motivated to incorporate $\alpha_t$ into $w(s, t)$ as proposed.

The exponent $a$ for $\alpha_t^a$ takes a value of either 1 or 2. We explain the reason for each decision here. When $a = 1$, we guide the gradient $\frac{\partial}{\partial \theta} \boldsymbol{f}_{s,t}^\theta(\mathbf{x}_t)$, with score difference $(\epsilon_\theta(\mathbf{x}_t, s, t) - \epsilon_{\text{target}})$. To motivate $a = 2$, we first note that the weighted gradient

$$\tilde{w}(s, t)\frac{\partial}{\partial \theta} \boldsymbol{f}_{s,t}^\theta(\mathbf{x}_t) = \frac{1}{|c_{\text{out}}(s, t)|}\frac{\partial}{\partial \theta}\boldsymbol{f}_{s,t}^\theta(\mathbf{x}_t) = \pm\frac{\partial}{\partial \theta}\hat{\boldsymbol{G}}_\theta(\mathbf{x}_t, s, t)$$

and as shown above that $\epsilon$-prediction is parameterized as

$$\epsilon_\theta(\mathbf{x}_t, s, t) = \mathbf{x}_t + \alpha_t \hat{\boldsymbol{G}}_\theta(\mathbf{x}_t, s, t).$$

Multiplying an additional $\alpha_t$ to $\frac{\partial}{\partial \theta}\hat{\boldsymbol{G}}_\theta(\mathbf{x}_t, s, t)$ therefore implicitly reparameterizes our model into an $\epsilon$-prediction model. The case of $a = 2$ therefore implicitly reparameterizes our model into an $\epsilon$-prediction model guided by the score difference $(\epsilon_\theta(\mathbf{x}_t, s, t) - \epsilon_{\text{target}})$. Empirically, $\alpha_t^2$ additionally downweights loss for larger $t$ compared to $\alpha_t$, allowing the model to train on smaller time-steps more effectively.

Lastly, the division of $\alpha_t^2 + \sigma_t^2$ is inspired by the increased weighting for middle time-steps (Esser et al., 2024) for Flow Matching training. This is purely an empirical decision.

## D. Training Algorithm

We present the training algorithm in Algorithm 3.

## E. Classifier-Free Guidance

We refer readers to Appendix C.5 for analysis of each parameterization. Most notably, the network $\boldsymbol{G}_\theta$ in both (1) Simple-EDM with cosine diffusion schedule and (2) Euler-FM with OT-FM schedule are equivalent to $v$-prediction parameterization

---

**Algorithm 3** IMM Training

---

**Input:** model $\boldsymbol{f}^\theta$, data distribution $q(\mathbf{x})$ and label distribution $q(\mathbf{c}|\mathbf{x})$ (if label is used), prior distribution $\mathcal{N}(0, \sigma_d^2 I)$, time distribution $p(t)$ and $p(s|t)$, DDIM interpolator $\text{DDIM}(\mathbf{x}_t, \mathbf{x}, s, t)$ and its flow coefficients $\alpha_t, \sigma_t$, mapping function $r(s,t)$, kernel function $k(\cdot, \cdot)$, weighting function $w(s,t)$, batch size $B$, particle number $M$, label dropout probability $p$
**Output:** learned model $\boldsymbol{f}^\theta$
Initialize $n \leftarrow 0, \theta^0 \leftarrow \theta$
**while** model not converged **do**

    Sample a batch of data, label, and prior, and split into $B/M$ groups, $\{(x^{(i,j)}, c^{(i,j)}, \epsilon^{(i,j)})\}_{i=1,j=1}^{B/M,M}$

    For each group, sample $\{(s^i, t^i)\}_{i=1}^{B/M}$ and $r^i = r(s^i, t^i)$ for each $i$. This results in a tuple $\{(s^i, r^i, t^i)\}_{i=1}^{B/M}$

    $x_{t^i}^{(i,j)} \leftarrow \text{DDIM}(\epsilon^{(i,j)}, x^{(i,j)}, t^i, 1) = \alpha_{t^i} x^{(i,j)} + \sigma_{t^i} \epsilon^{(i,j)}, \forall(i,j)$

    $x_{r^i}^{(i,j)} \leftarrow \text{DDIM}(x_{t^i}^{(i,j)}, x^{(i,j)}, r^i, t^i), \forall(i,j)$

    (Optional) Randomly drop each label $c^{(i,j)}$ to be null token $\varnothing$ with probability $p$

    $\theta_{n+1} \leftarrow$ optimizer step by minimizing $\hat{\mathcal{L}}_{\text{IMM}}(\theta_n)$ using model $\boldsymbol{f}^{\theta_n}$ (see Eq. (67)) (optionally inputting $c^{(i,j)}$ into network)

**end while**

---

in diffusion (Salimans & Ho, 2022) and FM (Lipman et al., 2022). When conditioned on label **c** during sampling, it is customary to use classifier-free guidance to reweight this $v$-prediction network via

$$\boldsymbol{G}_\theta^w(c_{\text{in}}(t)\mathbf{x}_t, c_{\text{noise}}(s), c_{\text{noise}}(t), \mathbf{c}) \tag{85}$$
$$= w\boldsymbol{G}_\theta(c_{\text{in}}(t)\mathbf{x}_t, c_{\text{noise}}(s), c_{\text{noise}}(t), \mathbf{c}) + (1-w)\boldsymbol{G}_\theta(c_{\text{in}}(t)\mathbf{x}_t, c_{\text{noise}}(s), c_{\text{noise}}(t), \varnothing)$$

with guidance weight $w$ so that the classifier-free guided $\boldsymbol{f}_{s,t,w}^\theta(\mathbf{x}_t)$ is

$$\boldsymbol{f}_{s,t,w}^\theta(\mathbf{x}_t) = c_{\text{skip}}(s,t)\mathbf{x}_t + c_{\text{out}}(s,t)\boldsymbol{G}_\theta^w(c_{\text{in}}(t)\mathbf{x}_t, c_{\text{noise}}(s), c_{\text{noise}}(t), \mathbf{c}) \tag{86}$$

## F. Sampling Algorithms

**Pushforward sampling.** See Algorithm 4. We assume a series of $N$ time steps $\{t_i\}_{i=0}^N$ with $T = t_N > t_{N-1} > \cdots > t_2 > t_1 > t_0 = \epsilon$ for the maximum time $T$ and minimum time $\epsilon$. Denote $\sigma_d$ as data standard deviation.

**Restart sampling.** See Algorithm 5. Different from pushforward sampling, $N$ time steps $\{t_i\}_{i=0}^N$ do not need to be strictly decreasing for all time steps, *e.g.* $T = t_N \geq t_{N-1} \geq \cdots \geq t_2 \geq t_1 \geq t_0 = \epsilon$ (assuming $T > \epsilon$). Different from pushforward sampling, restart sampling first denoise a clean sample before resampling a noise to be added to this clean sample. Then a clean sample is predicted again. The process is iterated for $N$ steps.

## G. Connection with Prior Works

### G.1. Consistency Models

Consistency models explicitly match PF-ODE trajectories using a network $\boldsymbol{g}_\theta(\mathbf{x}_t, t)$ that directly outputs a sample given any $\mathbf{x}_t \sim q_t(\mathbf{x}_t)$. The network explicitly uses EDM parameterization to satisfy boundary condition $\boldsymbol{g}_\theta(\mathbf{x}_0, 0) = \mathbf{x}_0$ and trains via loss $\mathbb{E}_{\mathbf{x}_t, \mathbf{x}, t}\left[\|\boldsymbol{g}_\theta(\mathbf{x}_t, t) - \boldsymbol{g}_{\theta^-}(\mathbf{x}_r, r)\|^2\right]$ where $\mathbf{x}_r$ is a deterministic function of $\mathbf{x}_t$ from an ODE solver.

We show that CM loss is a special case of our simplified IMM objective.

**Lemma 1.** *When $\mathbf{x}_t = \mathbf{x}_t'$, $\mathbf{x}_r = \mathbf{x}_r'$, $k(x,y) = -\|x-y\|^2$, and $s > 0$ is a small constant, Eq. (12) reduces to CM loss $\mathbb{E}_{\mathbf{x}_t, \mathbf{x}, t}\left[w(t)\|\boldsymbol{g}_\theta(\mathbf{x}_t, t) - \boldsymbol{g}_{\theta^-}(\mathbf{x}_r, r)\|^2\right]$ for some valid mapping $r(t) < t$.*

*Proof.* Since $\mathbf{x}_t = \mathbf{x}_t'$, $\mathbf{x}_r = \mathbf{x}_r'$, we have $\boldsymbol{f}_{s,t}^\theta(\mathbf{x}_t) = \boldsymbol{f}_{s,t}^\theta(\mathbf{x}_t')$ and $\boldsymbol{f}_{s,r}^\theta(\mathbf{x}_r) = \boldsymbol{f}_{s,r}^\theta(\mathbf{x}_r')$. So $k(\boldsymbol{f}_{s,t}^\theta(\mathbf{x}_t), \boldsymbol{f}_{s,t}^\theta(\mathbf{x}_t')) = k(\boldsymbol{f}_{s,r}^\theta(\mathbf{x}_r), \boldsymbol{f}_{s,r}^\theta(\mathbf{x}_r')) = 0$ by definition. Since $k(x,y) = -\|x-y\|^2$, it is easy to see Eq. (12) reduces to

$$\mathbb{E}_{\mathbf{x}_t, t}\left[w(s,t)\|\boldsymbol{f}_{s,t}^\theta(\mathbf{x}_t) - \boldsymbol{f}_{s,r}^\theta(\mathbf{x}_r)\|^2\right] \tag{87}$$

---

**Algorithm 4** Pushforward Sampling

---

**Input:** model $\boldsymbol{f}^\theta$, time steps $\{t_i\}_{i=0}^N$, prior distribution $\mathcal{N}(0, \sigma_d^2 I)$, (optional) guidance weight $w$
**Output:** $\mathbf{x}_{t_0}$
**Sample** $\mathbf{x}_N \sim \mathcal{N}(0, \sigma_d^2 I)$
**for** $i = N, \ldots, 1$ **do**
    (Optional) $w \leftarrow 1$ if $N = 1$        // can optionally discard unconditional branch for $N = 1$
    $\mathbf{x}_{t_{i-1}} \leftarrow \boldsymbol{f}^\theta_{t_{i-1}, t_i}(\mathbf{x}_{t_i})$ or $\boldsymbol{f}^\theta_{t_{i-1}, t_i, w}(\mathbf{x}_{t_i})$
**end for**

---

**Algorithm 5** Restart Sampling

---

**Input:** model $\boldsymbol{f}^\theta$, time steps $\{t_i\}_{i=0}^N$, prior distribution $\mathcal{N}(0, \sigma_d^2 I)$, DDIM interpolant coefficients $\alpha_t$ and $\sigma_t$, (optional) guidance weight $w$
**Output:** $\mathbf{x}_{t_0}$
**Sample** $\mathbf{x}_N \sim \mathcal{N}(0, \sigma_d^2 I)$
**for** $i = N, \ldots, 1$ **do**
    (Optional) $w \leftarrow 1$ if $N = 1$        // can optionally discard unconditional branch for $N = 1$
    $\tilde{\mathbf{x}} \leftarrow \boldsymbol{f}^\theta_{t_0, t_i}(\mathbf{x}_{t_i})$ or $\boldsymbol{f}^\theta_{t_0, t_i, w}(\mathbf{x}_{t_i})$
    **if** $i \neq 1$ **then**
        $\tilde{\boldsymbol{\epsilon}} \sim \mathcal{N}(0, \sigma_d^2 I)$
        $\mathbf{x}_{t_{i-1}} \leftarrow \alpha_{t_{i-1}} \tilde{\mathbf{x}} + \sigma_{t_{i-1}} \tilde{\boldsymbol{\epsilon}}$                    // or more generally $\mathbf{x}_{t_{i-1}} \sim q_t(\mathbf{x}_{t_{i-1}} | \tilde{\mathbf{x}}, \tilde{\boldsymbol{\epsilon}})$
    **else**
        $\mathbf{x}_{t_0} = \tilde{\mathbf{x}}$
    **end if**
**end for**

---

where $w(s, t)$ is a weighting function. If $s$ is a small positive constant, we further have $\boldsymbol{f}^\theta_{s,t}(\mathbf{x}_t) \approx \boldsymbol{g}_\theta(\mathbf{x}_t, t)$ where we drop $s$ as input. If $\boldsymbol{g}_\theta(\mathbf{x}_t, t)$ itself satisfies boundary condition at $s = 0$, we can directly take $s = 0$ in which case $\boldsymbol{f}^\theta_{0,t}(\mathbf{x}_t) = \boldsymbol{g}_\theta(\mathbf{x}_t, t)$. And under these assumptions, our loss becomes

$$\mathbb{E}_{\mathbf{x}_t, \mathbf{x}, t} \left[ w(t) \| \boldsymbol{g}_\theta(\mathbf{x}_t, t) - \boldsymbol{g}_{\theta^-}(\mathbf{x}_r, r) \|^2 \right], \tag{88}$$

which is simply a CM loss using $\ell_2$ distance. $\qquad\square$

However, one can notice that from a moment-matching perspective, this loss significantly deviates from a proper divergence between distributions, and is problematic in two aspects. First, it assumes single-particle estimate, which now ignores the entropy repulsion term in MMD that arises only during multi-particle estimation. This can contribute to mode collapse and training instability of CM. Second, the choice of energy kernel is not a proper positive definite kernel required by MMD. At best, it only matches the *first* moment (its Taylor expansion cannot cover all moments as in RBF kernels), which is insufficient for matching two complex distributions! We should use kernels that match higher moments in practice. In fact, we show in the following Lemma that the pseudo-huber loss proposed in Song & Dhariwal (2023) matches higher moments as a kernel.

**Lemma 2.** *Negative pseudo-huber loss $k_c(x, y) = c - \sqrt{\|x - y\|^2 + c^2}$ for $c > 0$ is a conditionally positive definite kernel that matches all moments of $x$ and $y$ where weights on higher moments depend on $c$.*

*Proof.* We first check that negative pseudo-huber loss $c - \sqrt{\|x - y\|^2 + c^2}$ is a *conditionally positive definite kernel* (Auffray & Barbillon, 2009). By definition, $k(x, y)$ is conditionally positive definite if for $x_1, \cdots, x_n \in \mathbb{R}^D$ and $c_1, \cdots, c_n \in \mathbb{R}^D$ with $\sum_{i=1}^n c_i = 0$

$$\sum_{i=1}^n \sum_{j=1}^n c_i c_j k(x_i, x_j) \geq 0 \tag{89}$$

We know that negative $L_2$ distance $-\|x - y\|$ is conditionally positive definite. We prove this below for completion. Due to triangle inequality, $-\|x - y\| \geq -\|x\| - \|y\|$. Then

$$\sum_{i=1}^{n} \sum_{j=1}^{n} c_i c_j \left(-\|x_i - x_j\|\right) \geq \sum_{i=1}^{n} \sum_{j=1}^{n} c_i c_j \left(-\|x_i\| - \|x_j\|\right) \tag{90}$$

$$= -\sum_{i=1}^{n} c_i \left(\sum_{j=1}^{n} c_j \|x_j\|\right) - \sum_{j=1}^{n} c_j \left(\sum_{i=1}^{n} c_i \|x_i\|\right) \tag{91}$$

$$\overset{(a)}{=} 0 \tag{92}$$

where $(a)$ is due to $\sum_{i=1}^{n} c_i = 0$. Now since $c - \sqrt{\|z\|^2 + c^2} \geq -\|z\|$ for all $c > 0$, we have

$$\sum_{i=1}^{n} \sum_{j=1}^{n} c_i c_j \left(c - \sqrt{\|x_i - x_j\|^2 + c^2}\right) \geq \sum_{i=1}^{n} \sum_{j=1}^{n} c_i c_j \left(-\|x_i - x_j\|\right) \geq 0 \tag{93}$$

So negative pseudo-huber loss is a valid conditionally positive definite kernel.

Next, we analyze pseudo-huber loss's effect on higher-order moments by directly Taylor expanding $\sqrt{\|z\|^2 + c^2} - c$ at $z = 0$

$$\sqrt{\|z\|^2 + c^2} - c = \frac{1}{2c}\|z\|^2 - \frac{1}{8c^3}\|z\|^4 + \frac{1}{16c^5}\|z\|^6 - \frac{5}{128c^7}\|z\|^8 + \mathcal{O}(\|z\|^9) \tag{94}$$

$$= \frac{1}{2c}\|x - y\|^2 - \frac{1}{8c^3}\|x - y\|^4 + \frac{1}{16c^5}\|x - y\|^6 - \frac{5}{128c^7}\|x - y\|^8 + \mathcal{O}(\|x - y\|^9) \tag{95}$$

$$\tag{96}$$

where we substitute $z = x - y$. Each higher order $\|x - y\|^k$ for $k > 2$ expands to a polynomial containing up to $k$-th moments, *i.e.*, $\{x, x^2, \ldots x^k\}$, $\{y, y^2, \ldots y^k\}$, thus the implicit feature map contains all higher moments where $c$ contributes to the weightings in front of each term. $\square$

Furthermore, we extend our finite difference (between $r(s, t)$ and $t$) IMM objective to the differential limit by taking $r(s, t) \to t$ in Appendix H. This results in a new objective that similarly subsumes continuous-time CM (Song et al., 2023; Lu & Song, 2024) as a single-particle special case.

### G.2. Diffusion GAN and Adversarial Consistency Distillation

Diffusion GAN (Xiao et al., 2021) parameterizes its generative distribution as

$$p_{s|t}^{\theta}(\mathbf{x}_s|\mathbf{x}_t) = \int q_{s|t}(\mathbf{x}_s|\boldsymbol{G}_\theta(\mathbf{x}_t, \mathbf{z}), \mathbf{x}_t)p(\mathbf{z})\mathrm{d}\mathbf{z}$$

where $\boldsymbol{G}_\theta$ is a neural network, $p(\mathbf{z})$ is standard Gaussian distribution, and $q_{s|t}(\mathbf{x}_s|\mathbf{x}, \mathbf{x}_t)$ is the DDPM posterior

$$q_{s|t}(\mathbf{x}_s|\mathbf{x}, \mathbf{x}_t) = \mathcal{N}(\mu_{s,t}, \sigma_{s,t}^2 I) \tag{97}$$

$$\mu_Q = \frac{\alpha_t \sigma_s^2}{\alpha_s \sigma_t^2}\mathbf{x}_t + \alpha_s(1 - \frac{\alpha_t^2}{\alpha_s^2}\frac{\sigma_s^2}{\sigma_t^2})\mathbf{x}$$

$$\sigma_Q^2 = \sigma_s^2(1 - \frac{\alpha_t^2}{\alpha_s^2}\frac{\sigma_s^2}{\sigma_t^2})$$

Note that DDPM posterior is a *stochastic* interpolant, and more importantly, it is self-consistent, which we show in the Lemma below.

**Lemma 6.** *For all $0 \leq s < t \leq 1$, DDPM posterior distribution from $t$ to $s$ as defined in Eq. (97) is a self-consistent Gaussian interpolant between $\mathbf{x}$ and $\mathbf{x}_t$.*

*Proof.* Let $\mathbf{x}_r \sim q_{r|t}(\mathbf{x}_r|\mathbf{x}, \mathbf{x}_t)$ and $\mathbf{x}_s \sim q_{s|r}(\mathbf{x}_s|\mathbf{x}, \mathbf{x}_r)$, we show that $\mathbf{x}_s$ follows $q_{s|t}(\mathbf{x}_s|\mathbf{x}, \mathbf{x}_t)$.

$$\mathbf{x}_r = \frac{\alpha_t \sigma_r^2}{\alpha_r \sigma_t^2} \mathbf{x}_t + \alpha_r (1 - \frac{\alpha_t^2}{\alpha_r^2} \frac{\sigma_r^2}{\sigma_t^2}) \mathbf{x} + \sigma_r \sqrt{1 - \frac{\alpha_t^2}{\alpha_r^2} \frac{\sigma_r^2}{\sigma_t^2}} \boldsymbol{\epsilon}_1 \tag{98}$$

$$\mathbf{x}_s = \frac{\alpha_r \sigma_s^2}{\alpha_s \sigma_r^2} \mathbf{x}_r + \alpha_s (1 - \frac{\alpha_r^2}{\alpha_s^2} \frac{\sigma_s^2}{\sigma_t^2}) \mathbf{x} + \sigma_s \sqrt{1 - \frac{\alpha_r^2}{\alpha_s^2} \frac{\sigma_s^2}{\sigma_t^2}} \boldsymbol{\epsilon}_2 \tag{99}$$

where $\boldsymbol{\epsilon}_1, \boldsymbol{\epsilon}_2 \sim \mathcal{N}(0, I)$ are i.i.d. Gaussian noise. Directly expanding

$$\mathbf{x}_s = \frac{\alpha_r \sigma_s^2}{\alpha_s \sigma_r^2} \left[ \frac{\alpha_t \sigma_r^2}{\alpha_r \sigma_t^2} \mathbf{x}_t + \alpha_r (1 - \frac{\alpha_t^2}{\alpha_r^2} \frac{\sigma_r^2}{\sigma_t^2}) \mathbf{x} + \sigma_r \sqrt{1 - \frac{\alpha_t^2}{\alpha_r^2} \frac{\sigma_r^2}{\sigma_t^2}} \boldsymbol{\epsilon}_1 \right] + \alpha_s (1 - \frac{\alpha_r^2}{\alpha_s^2} \frac{\sigma_s^2}{\sigma_t^2}) \mathbf{x} + \sigma_s \sqrt{1 - \frac{\alpha_r^2}{\alpha_s^2} \frac{\sigma_s^2}{\sigma_t^2}} \boldsymbol{\epsilon}_2 \tag{100}$$

$$= \frac{\alpha_t \sigma_s^2}{\alpha_s \sigma_t^2} \mathbf{x}_t + \left[ \frac{\alpha_r^2 \sigma_s^2}{\alpha_s \sigma_r^2} (1 - \frac{\alpha_t^2}{\alpha_r^2} \frac{\sigma_r^2}{\sigma_t^2}) + \alpha_s (1 - \frac{\alpha_r^2}{\alpha_s^2} \frac{\sigma_r^2}{\sigma_t^2}) \right] \mathbf{x} + \frac{\alpha_r \sigma_s^2}{\alpha_s \sigma_r^2} \left[ \sigma_r \sqrt{1 - \frac{\alpha_t^2}{\alpha_r^2} \frac{\sigma_r^2}{\sigma_t^2}} \boldsymbol{\epsilon}_1 \right] + \sigma_s \sqrt{1 - \frac{\alpha_r^2}{\alpha_s^2} \frac{\sigma_s^2}{\sigma_t^2}} \boldsymbol{\epsilon}_2 \tag{101}$$

$$= \frac{\alpha_t \sigma_s^2}{\alpha_s \sigma_t^2} \mathbf{x}_t + \alpha_s (1 - \frac{\alpha_t^2}{\alpha_s^2} \frac{\sigma_s^2}{\sigma_t^2}) \mathbf{x} + \frac{\alpha_r \sigma_s^2}{\alpha_s \sigma_r} \sqrt{1 - \frac{\alpha_t^2}{\alpha_r^2} \frac{\sigma_r^2}{\sigma_t^2}} \boldsymbol{\epsilon}_1 + \sigma_s \sqrt{1 - \frac{\alpha_r^2}{\alpha_s^2} \frac{\sigma_s^2}{\sigma_t^2}} \boldsymbol{\epsilon}_2 \tag{102}$$

$$\overset{(a)}{=} \frac{\alpha_t \sigma_s^2}{\alpha_s \sigma_t^2} \mathbf{x}_t + \alpha_s (1 - \frac{\alpha_t^2}{\alpha_s^2} \frac{\sigma_s^2}{\sigma_t^2}) \mathbf{x} + \sigma_s \sqrt{1 - \frac{\alpha_t^2}{\alpha_s^2} \frac{\sigma_s^2}{\sigma_t^2}} \boldsymbol{\epsilon}_3 \tag{103}$$

where $(a)$ is due to the fact that sum of two independent Gaussian variables with variance $a^2$ and $b^2$ is also Gaussian with variance $a^2 + b^2$, and $\boldsymbol{\epsilon}_3 \sim \mathcal{N}(0, I)$ is another independent Gaussian noise. We show the calculation of the variance:

$$\frac{\alpha_r^2}{\alpha_s^2} \frac{\sigma_s^4}{\sigma_r^2} (1 - \frac{\alpha_t^2}{\alpha_r^2} \frac{\sigma_r^2}{\sigma_t^2}) + \sigma_s^2 (1 - \frac{\alpha_r^2}{\alpha_s^2} \frac{\sigma_s^2}{\sigma_t^2}) = \frac{\alpha_r^2}{\alpha_s^2} \frac{\sigma_s^{\cancel{4}}}{\cancel{\sigma_r^2}} - \frac{\alpha_t^2 \sigma_s^4}{\sigma_t^2 \alpha_s^2} + \sigma_s^2 - \frac{\alpha_r^2}{\alpha_s^2} \frac{\sigma_s^{\cancel{4}}}{\sigma_t^2}$$

$$= \sigma_s^2 (1 - \frac{\alpha_t^2}{\alpha_s^2} \frac{\sigma_s^2}{\sigma_t^2})$$

This shows $\mathbf{x}_s$ follows $q_{s|t}(\mathbf{x}_s|\mathbf{x}, \mathbf{x}_t)$ and completes the proof. □

This shows another possible design of the interpolant that can be used, and diffusion GAN's formulation generally complies with our design of the generative distribution, except that it learns this conditional distribution of $\mathbf{x}$ given $\mathbf{x}_t$ directly while we learn a marginal distribution. When they directly learn the conditional distribution by matching $p_{s|t}^\theta(\mathbf{x}|\mathbf{x}_t)$ with $q_t(\mathbf{x}|\mathbf{x}_t)$, the model is forced to learn $q_t(\mathbf{x}|\mathbf{x}_t)$ and there only exists one minimizer. However, in our case, the model can learn multiple different solutions because we match the marginals instead.

**GAN loss and MMD loss.** We also want to draw attention to similarity between GAN loss used in Xiao et al. (2021); Sauer et al. (2025) and MMD loss. MMD is an integral probability metric over a set of functions $\mathcal{F}$ in the following form

$$D_\mathcal{F}(P, Q) = \sup_{f \in \mathcal{F}} \left| \mathbb{E}_{\mathbf{x} \sim P(\mathbf{x})} f(\mathbf{x}) - \mathbb{E}_{\mathbf{y} \sim Q(\mathbf{y})} f(\mathbf{y}) \right|$$

where a supremum is taken on this set of functions. This naturally gives rise to an adversarial optimization algorithm if $\mathcal{F}$ is defined as the set of neural networks. However, MMD bypasses this by selecting $\mathcal{F}$ as the RKHS where the optimal $f$ can be analytically found. This eliminates the adversarial objective and gives a stable minimization objective in practice. However, this is not to say that RKHS is the best function set. With the right optimizers and training scheme, the adversarial objective may achieve better empirical performance, but this also makes the algorithm difficult to scale to large datasets.

### G.3. Generative Moment Matching Network

It is trivial to check that GMMN is a special parameterization. We fix $t = 1$, and due to boundary condition, $r(s, t) \equiv s = 0$ implies training target $p_{s|r(s,t)}^{\theta^-}(\mathbf{x}_s) = q_s(\mathbf{x}_s)$ is the data distribution. Additionally, $p_{s|t}^\theta(\mathbf{x}_s)$ is a simple pushforward of prior $p(\boldsymbol{\epsilon})$ through network $\mathbf{g}_\theta(\boldsymbol{\epsilon})$ where drop dependency on $t$ and $s$ since they are constant.

# H. Differential Inductive Moment Matching

Similar to the continuous-time CMs presented in (Lu & Song, 2024), our MMD objective can be taken to the differential limit. Consider the simplifed loss and parameterization in Eq. (12), we use the RBF kernel as our kernel of choice for simplicity.

**Theorem 2** (Differential Inductive Moment Matching). *Let $\boldsymbol{f}_{s,t}^{\theta}(\mathbf{x}_t)$ be a twice continuously differentiable function with bounded first and second derivatives, let $k(\cdot,\cdot)$ be RBF kernel with unit bandwidth, $\mathbf{x}, \mathbf{x}' \sim q(\mathbf{x})$, $\mathbf{x}_t \sim q_t(\mathbf{x}_t|\mathbf{x})$, $\mathbf{x}_t' \sim q_s(\mathbf{x}_t'|\mathbf{x}')$, $\mathbf{x}_r = \text{DDIM}(\mathbf{x}_t, \mathbf{x}, t, r)$ and $\mathbf{x}_r' = \text{DDIM}(\mathbf{x}_t', \mathbf{x}', t, r)$, the following objective*

$$\mathcal{L}_{\textit{IMM-}\infty}(\theta, t) = \lim_{r \to t} \frac{1}{(t-r)^2} \mathbb{E}_{\mathbf{x}_t, \mathbf{x}_t', \mathbf{x}_r, \mathbf{x}_r'} \left[ k\left(\boldsymbol{f}_{s,t}^{\theta}(\mathbf{x}_t), \boldsymbol{f}_{s,t}^{\theta}(\mathbf{x}_t')\right) + k\left(\boldsymbol{f}_{s,r}^{\theta}(\mathbf{x}_r), \boldsymbol{f}_{s,r}^{\theta}(\mathbf{x}_r')\right) \right. \tag{104}$$
$$\left. - k\left(\boldsymbol{f}_{s,t}^{\theta}(\mathbf{x}_t), \boldsymbol{f}_{s,r}^{\theta}(\mathbf{x}_r')\right) - k\left(\boldsymbol{f}_{s,t}^{\theta}(\mathbf{x}_t'), \boldsymbol{f}_{s,r}^{\theta}(\mathbf{x}_r)\right) \right]$$

*can be analytically derived as*

$$\mathbb{E}_{\mathbf{x}_t, \mathbf{x}_t'} \left[ e^{-\frac{1}{2}\|\boldsymbol{f}_{s,t}^{\theta}(\mathbf{x}_t') - \boldsymbol{f}_{s,t}^{\theta}(\mathbf{x}_t)\|^2} \left( \frac{\mathrm{d}\boldsymbol{f}_{s,t}^{\theta}(\mathbf{x}_t)}{\mathrm{d}t}^{\top} \frac{\mathrm{d}\boldsymbol{f}_{s,t}^{\theta}(\mathbf{x}_t')}{\mathrm{d}t} \right. \right. \tag{105}$$
$$\left. \left. - \frac{\mathrm{d}\boldsymbol{f}_{s,t}^{\theta}(\mathbf{x}_t)}{\mathrm{d}t}^{\top} \left(\boldsymbol{f}_{s,t}^{\theta}(\mathbf{x}_t) - \boldsymbol{f}_{s,t}^{\theta}(\mathbf{x}_t')\right) \left(\boldsymbol{f}_{s,t}^{\theta}(\mathbf{x}_t) - \boldsymbol{f}_{s,t}^{\theta}(\mathbf{x}_t')\right)^{\top} \frac{\mathrm{d}\boldsymbol{f}_{s,t}^{\theta}(\mathbf{x}_t')}{\mathrm{d}t} \right) \right]$$

*Proof.* Firstly, the limit can be exchanged with the expectation due to dominated convergence theorem where the integrand consists of kernel functions which can be assumed to be upper bounded by 1, *e.g.* RBF kernels are upper bounded by 1, and thus integrable. It then suffices to check the limit of the integrand. Before that, let us review the first and second-order Taylor expansion of $e^{-\|x-y\|^2}$. We let $a, b \in \mathbb{R}^D$ be constants to be expanded around. We note down the Taylor expansion to second-order below for notational convenience.

- $e^{-\|x-b\|^2/2}$ at $x = a$:

$$e^{-\|a-b\|^2/2} - e^{-\|a-b\|^2/2}(a-b)^{\top}(x-a) + \frac{1}{2}(x-a)^{\top}e^{-\|a-b\|^2/2}\left((a-b)(a-b)^{\top} - I\right)(x-a)$$

- $e^{-\|y-a\|^2/2}$ at $y = b$:

$$e^{-\|b-a\|^2/2} - e^{-\|b-a\|^2/2}(b-a)^{\top}(y-b) + \frac{1}{2}(y-b)^{\top}e^{-\|b-a\|^2/2}\left((b-a)(b-a)^{\top} - I\right)(y-b)$$

- $e^{-\|x-y\|^2/2}$ at $x = a, y = b$:

$$e^{-\|a-b\|^2/2} - e^{-\|a-b\|^2/2}(a-b)^{\top}(x-a) - e^{-\|b-a\|^2/2}(b-a)^{\top}(y-b)$$
$$+ \frac{1}{2}(y-b)^{\top}e^{-\|b-a\|^2/2}\left((b-a)(b-a)^{\top} - I\right)(y-b)$$
$$+ \frac{1}{2}(y-b)^{\top}e^{-\|b-a\|^2/2}\left((b-a)(b-a)^{\top} - I\right)(y-b)$$
$$+ (x-a)^{\top}e^{-\|b-a\|^2/2}\left(I - (a-b)(a-b)^{\top}\right)(y-b)$$

Putting it together, the above results imply

$$e^{-\|x-y\|^2/2} + e^{-\|a-b\|^2/2} - e^{-\|x-b\|^2/2} - e^{-\|y-a\|^2/2}$$
$$\approx (x-a)^{\top}e^{-\|b-a\|^2/2}\left(I - (a-b)(a-b)^{\top}\right)(y-b)$$

since it is easy to check that the remaining terms cancel.

Substituting $x = \boldsymbol{f}_{s,t}^{\theta}(\mathbf{x}_t)$, $a = \boldsymbol{f}_{s,r}^{\theta}(\mathbf{x}_r)$, $y = \boldsymbol{f}_{s,t}^{\theta}(\mathbf{x}_t')$, $b = \boldsymbol{f}_{s,r}^{\theta}(\mathbf{x}_r')$, we furthermore have

$$\lim_{r \to t} \frac{1}{t-r}(x - a) = \lim_{r \to t} \frac{1}{t-r} \left[ \boldsymbol{f}_{s,t}^{\theta}(\mathbf{x}_t) - \boldsymbol{f}_{s,r}^{\theta}(\mathbf{x}_r) \right] \tag{106}$$

$$= \lim_{r \to t} \frac{1}{t-r} \left[ (t-r)\frac{\mathrm{d}\boldsymbol{f}_{s,t}^{\theta}(\mathbf{x}_t)}{\mathrm{d}t} + \mathcal{O}(|t-r|^2) \right] \tag{107}$$

$$= \frac{\mathrm{d}\boldsymbol{f}_{s,t}^{\theta}(\mathbf{x}_t)}{\mathrm{d}t} \tag{108}$$

Similarly,

$$\lim_{r \to t} \frac{1}{t-r}(y - b) = \lim_{r \to t} \frac{1}{t-r} \left[ \boldsymbol{f}_{s,t}^{\theta}(\mathbf{x}_t') - \boldsymbol{f}_{s,r}^{\theta}(\mathbf{x}_r') \right] = \frac{\mathrm{d}\boldsymbol{f}_{s,t}^{\theta}(\mathbf{x}_t')}{\mathrm{d}t} \tag{109}$$

Therefore, $\mathcal{L}_{\text{IMM-}\infty}(\theta, t)$ can be derived as

$$\mathbb{E}_{\mathbf{x}_t, \mathbf{x}_t'} \left[ e^{-\frac{1}{2} \left\| \boldsymbol{f}_{s,t}^{\theta}(\mathbf{x}_t') - \boldsymbol{f}_{s,t}^{\theta}(\mathbf{x}_t) \right\|^2} \frac{\mathrm{d}\boldsymbol{f}_{s,t}^{\theta}(\mathbf{x}_t)}{\mathrm{d}t}^{\top} \right. \tag{110}$$

$$\left. \left( I - \left( \boldsymbol{f}_{s,t}^{\theta}(\mathbf{x}_t) - \boldsymbol{f}_{s,t}^{\theta}(\mathbf{x}_t') \right) \left( \boldsymbol{f}_{s,t}^{\theta}(\mathbf{x}_t) - \boldsymbol{f}_{s,t}^{\theta}(\mathbf{x}_t') \right)^{\top} \right) \frac{\mathrm{d}\boldsymbol{f}_{s,t}^{\theta}(\mathbf{x}_t')}{\mathrm{d}t} \right] \tag{111}$$

$$= \mathbb{E}_{\mathbf{x}_t, \mathbf{x}_t'} \left[ e^{-\frac{1}{2} \left\| \boldsymbol{f}_{s,t}^{\theta}(\mathbf{x}_t') - \boldsymbol{f}_{s,t}^{\theta}(\mathbf{x}_t) \right\|^2} \left( \frac{\mathrm{d}\boldsymbol{f}_{s,t}^{\theta}(\mathbf{x}_t)}{\mathrm{d}t}^{\top} \frac{\mathrm{d}\boldsymbol{f}_{s,t}^{\theta}(\mathbf{x}_t')}{\mathrm{d}t} \right. \right. \tag{112}$$

$$\left. \left. - \frac{\mathrm{d}\boldsymbol{f}_{s,t}^{\theta}(\mathbf{x}_t)}{\mathrm{d}t}^{\top} \left( \boldsymbol{f}_{s,t}^{\theta}(\mathbf{x}_t) - \boldsymbol{f}_{s,t}^{\theta}(\mathbf{x}_t') \right) \left( \boldsymbol{f}_{s,t}^{\theta}(\mathbf{x}_t) - \boldsymbol{f}_{s,t}^{\theta}(\mathbf{x}_t') \right)^{\top} \frac{\mathrm{d}\boldsymbol{f}_{s,t}^{\theta}(\mathbf{x}_t')}{\mathrm{d}t} \right) \right]$$

$$\square$$

### H.1. Pseudo-Objective

Due to the stop-gradient operation, we can similarly find a *pseudo-objective* whose gradient matches the gradient of $\mathcal{L}_{\text{IMM-}\infty}(\theta, t)$ in the limit of $r \to t$.

**Theorem 3.** *Let $\boldsymbol{f}_{s,t}^{\theta}(\mathbf{x}_t)$ be a twice continuously differentiable function with bounded first and second derivatives, $k(\cdot, \cdot)$ be RBF kernel with unit bandwidth, $\mathbf{x}, \mathbf{x}' \sim q(\mathbf{x})$, $\mathbf{x}_t \sim q_t(\mathbf{x}_t|\mathbf{x})$, $\mathbf{x}_t' \sim q_t(\mathbf{x}_t'|\mathbf{x}')$, $\mathbf{x}_r = \text{DDIM}(\mathbf{x}_t, \mathbf{x}, t, r)$ and $\mathbf{x}_r' = \text{DDIM}(\mathbf{x}_t', \mathbf{x}', t, r)$, the gradient of the following pseudo-objective*

$$\nabla_{\theta} \mathcal{L}_{IMM-\infty}^{-}(\theta, t) = \lim_{r \to t} \frac{1}{(t-r)} \nabla_{\theta} \mathbb{E}_{\mathbf{x}_t, \mathbf{x}_t', \mathbf{x}_r, \mathbf{x}_r'} \left[ k\left( \boldsymbol{f}_{s,t}^{\theta}(\mathbf{x}_t), \boldsymbol{f}_{s,t}^{\theta}(\mathbf{x}_t') \right) + k\left( \boldsymbol{f}_{s,r}^{\theta^-}(\mathbf{x}_r), \boldsymbol{f}_{s,r}^{\theta^-}(\mathbf{x}_r') \right) \right. \tag{113}$$

$$\left. - k\left( \boldsymbol{f}_{s,t}^{\theta}(\mathbf{x}_t), \boldsymbol{f}_{s,r}^{\theta^-}(\mathbf{x}_r') \right) - k\left( \boldsymbol{f}_{s,t}^{\theta}(\mathbf{x}_t'), \boldsymbol{f}_{s,r}^{\theta^-}(\mathbf{x}_r) \right) \right]$$

*can be used to optimize $\theta$ and can be analytically derived as*

$$\mathbb{E}_{\mathbf{x}_t, \mathbf{x}_t'} \left[ e^{-\frac{1}{2} \left\| \boldsymbol{f}_{s,t}^{\theta^-}(\mathbf{x}_t') - \boldsymbol{f}_{s,t}^{\theta^-}(\mathbf{x}_t) \right\|^2} \left( \right. \right. \tag{114}$$

$$\left( \frac{\mathrm{d}\boldsymbol{f}_{s,t}^{\theta^-}(\mathbf{x}_t')}{\mathrm{d}t}^{\top} - \left[ \boldsymbol{f}_{s,t}^{\theta^-}(\mathbf{x}_t) - \boldsymbol{f}_{s,t}^{\theta^-}(\mathbf{x}_t') \right]^{\top} \frac{\mathrm{d}\boldsymbol{f}_{s,t}^{\theta^-}(\mathbf{x}_t')}{\mathrm{d}t} \left[ \boldsymbol{f}_{s,t}^{\theta^-}(\mathbf{x}_t) - \boldsymbol{f}_{s,t}^{\theta^-}(\mathbf{x}_t') \right]^{\top} \right) \nabla_{\theta} \boldsymbol{f}_{s,t}^{\theta}(\mathbf{x}_t) +$$

$$\left. \left( \frac{\mathrm{d}\boldsymbol{f}_{s,t}^{\theta^-}(\mathbf{x}_t)}{\mathrm{d}t}^{\top} - \left[ \boldsymbol{f}_{s,t}^{\theta^-}(\mathbf{x}_t) - \boldsymbol{f}_{s,t}^{\theta^-}(\mathbf{x}_t') \right]^{\top} \frac{\mathrm{d}\boldsymbol{f}_{s,t}^{\theta^-}(\mathbf{x}_t)}{\mathrm{d}t} \left[ \boldsymbol{f}_{s,t}^{\theta^-}(\mathbf{x}_t) - \boldsymbol{f}_{s,t}^{\theta^-}(\mathbf{x}_t') \right]^{\top} \right) \nabla_{\theta} \boldsymbol{f}_{s,t}^{\theta}(\mathbf{x}_t') \right) \right]$$

*Proof.* Similar to the derivation of $\mathcal{L}_{\text{IMM-}\infty}(\theta, t)$, let $x = \boldsymbol{f}^\theta_{s,t}(\mathbf{x}_t)$, $a = \boldsymbol{f}^{\theta^-}_{s,r}(\mathbf{x}_r)$, $y = \boldsymbol{f}^\theta_{s,t}(\mathbf{x}'_t)$, $b = \boldsymbol{f}^{\theta^-}_{s,r}(\mathbf{x}'_r)$, we have

$$\lim_{r \to t} \frac{1}{(t-r)} \nabla_\theta (x-a)^\top e^{-\|b-a\|^2/2} \left( I - (a-b)(a-b)^\top \right)(y-b)$$

$$= \lim_{r \to t} \frac{1}{(t-r)} e^{-\|b-a\|^2/2} \left[ \left( (y-b)^\top - \left( (a-b)^\top (y-b) \right)(a-b)^\top \right) \nabla_\theta x + \right.$$

$$\left. \left( (x-a)^\top - \left( (a-b)^\top (x-a) \right)(a-b)^\top \right) \nabla_\theta y \right]$$

where

$$\lim_{r \to t} \frac{1}{(t-r)}(x-a) = \frac{\mathrm{d}\boldsymbol{f}^{\theta^-}_{s,t}(\mathbf{x}_t)}{\mathrm{d}t}, \qquad \lim_{r \to t} \frac{1}{(t-r)}(y-b) = \frac{\mathrm{d}\boldsymbol{f}^{\theta^-}_{s,t}(\mathbf{x}'_t)}{\mathrm{d}t}$$

Note that $\frac{\mathrm{d}\boldsymbol{f}^{\theta^-}_{s,t}(\mathbf{x}_t)}{\mathrm{d}t}$ is now parameterized by $\theta^-$ instead of $\theta$ because the gradient is already taken w.r.t. $\boldsymbol{f}^\theta_{s,t}(\mathbf{x}_t)$ outside of the brackets, so $(x-a)$ and $(y-b)$ merely require *evaluation* at current $\theta$ with no gradient information, which $\theta^-$ satisfies. The objective can be derived as

$$\nabla_\theta \mathcal{L}^-_{\text{IMM-}\infty}(\theta, t)$$

$$= \mathbb{E}_{\mathbf{x}_t, \mathbf{x}'_t} \left[ e^{-\frac{1}{2} \left\| \boldsymbol{f}^{\theta^-}_{s,t}(\mathbf{x}'_t) - \boldsymbol{f}^{\theta^-}_{s,t}(\mathbf{x}_t) \right\|^2} \left( \right.\right.$$

$$\left( \frac{\mathrm{d}\boldsymbol{f}^{\theta^-}_{s,t}(\mathbf{x}'_t)}{\mathrm{d}t}^\top - \left[ \boldsymbol{f}^{\theta^-}_{s,t}(\mathbf{x}_t) - \boldsymbol{f}^{\theta^-}_{s,t}(\mathbf{x}'_t) \right]^\top \frac{\mathrm{d}\boldsymbol{f}^{\theta^-}_{s,t}(\mathbf{x}'_t)}{\mathrm{d}t} \left[ \boldsymbol{f}^{\theta^-}_{s,t}(\mathbf{x}_t) - \boldsymbol{f}^{\theta^-}_{s,t}(\mathbf{x}'_t) \right]^\top \right) \nabla_\theta \boldsymbol{f}^\theta_{s,t}(\mathbf{x}_t) +$$

$$\left. \left( \frac{\mathrm{d}\boldsymbol{f}^{\theta^-}_{s,t}(\mathbf{x}_t)}{\mathrm{d}t}^\top - \left[ \boldsymbol{f}^{\theta^-}_{s,t}(\mathbf{x}_t) - \boldsymbol{f}^{\theta^-}_{s,t}(\mathbf{x}'_t) \right]^\top \frac{\mathrm{d}\boldsymbol{f}^{\theta^-}_{s,t}(\mathbf{x}_t)}{\mathrm{d}t} \left[ \boldsymbol{f}^{\theta^-}_{s,t}(\mathbf{x}_t) - \boldsymbol{f}^{\theta^-}_{s,t}(\mathbf{x}'_t) \right]^\top \right) \nabla_\theta \boldsymbol{f}^\theta_{s,t}(\mathbf{x}'_t) \right) \right]$$

$\square$

## H.2. Connection with Continuous-Time CMs

Observing Eq. (105) and Eq. (110), we can see that when $\mathbf{x}_t = \mathbf{x}'_t$ and $\mathbf{x}_r = \mathbf{x}'_r$, $s$ being a small positive constant, then $\boldsymbol{f}^\theta_{s,t}(\mathbf{x}'_t) = \boldsymbol{f}^\theta_{s,t}(\mathbf{x}_t)$, and $\exp\left(-\frac{1}{2}\|\boldsymbol{f}^\theta_{s,t}(\mathbf{x}'_t) - \boldsymbol{f}^\theta_{s,t}(\mathbf{x}_t)\|^2\right) = 1$, and $\boldsymbol{f}^\theta_{s,t}(\mathbf{x}'_t) \approx \boldsymbol{g}_\theta(\mathbf{x}_t, t)$ where since $s$ is fixed we discard the dependency on $s$ as input. Then, Eq. (105) reduces to

$$\mathbb{E}_{\mathbf{x}_t} \left[ \left\| \frac{\mathrm{d}\boldsymbol{f}^\theta_{s,t}(\mathbf{x}_t)}{\mathrm{d}t} \right\|^2 \right] \tag{115}$$

which is the same as differential consistency loss (Song et al., 2023; Geng et al., 2024). And Eq. (110) reduces to

$$\mathbb{E}_{\mathbf{x}_t} \left[ \frac{\mathrm{d}\boldsymbol{f}^{\theta^-}_{s,t}(\mathbf{x}_t)}{\mathrm{d}t}^\top \nabla_\theta \boldsymbol{f}^\theta_{s,t}(\mathbf{x}_t) \right] \tag{116}$$

which is the pseudo-objective for continuous-time CMs (Song et al., 2023; Lu & Song, 2024) (minus a weighting function of choice).

# I. Experiment Settings

## I.1. Training & Parameterization Settings

We summarize our best runs in Table 5. Specifically, for ImageNet-256×256, we adopt a latent space paradigm for computational efficiency. For its autoencoder, we follow EDM2 (Karras et al., 2024) and pre-encode all images from

| | CIFAR-10 | ImageNet-256×256 | | | | |
|---|---|---|---|---|---|---|
| **Parameterization Setting** | | | | | | |
| Architecture | DDPM++ | DiT-S | DiT-B | DiT-L | DiT-XL | DiT-XL |
| GFlops | 21.28 | 6.06 | 23.01 | 80.71 | 118.64 | 118.64 |
| Params (M) | 55 | 33 | 130 | 458 | 675 | 675 |
| $c_{\text{noise}}(t)$ | $1000t$ | $1000t$ | $1000t$ | $1000t$ | $1000t$ | $1000t$ |
| $2^{\text{nd}}$ time conditioning | $s$ | $s$ | $s$ | $s$ | $s$ | $t-s$ |
| Flow Trajectory | OT-FM | OT-FM | OT-FM | OT-FM | OT-FM | OT-FM |
| $g_\theta(\mathbf{x}_t, s, t)$ | Simple-EDM | Euler-FM | Euler-FM | Euler-FM | Euler-FM | Euler-FM |
| $\sigma_d$ | 0.5 | 0.5 | 0.5 | 0.5 | 0.5 | 0.5 |
| Training iter | 400K | 1.2M | 1.2M | 1.2M | 1.2M | 1.2M |
| **Training Setting** | | | | | | |
| Dropout | 0.2 | 0 | 0 | 0 | 0 | 0 |
| Optimizer | RAdam | AdamW | AdamW | AdamW | AdamW | AdamW |
| Optimizer $\epsilon$ | $10^{-8}$ | $10^{-8}$ | $10^{-8}$ | $10^{-8}$ | $10^{-8}$ | $10^{-8}$ |
| $\beta_1$ | 0.9 | 0.9 | 0.9 | 0.9 | 0.9 | 0.9 |
| $\beta_2$ | 0.999 | 0.999 | 0.999 | 0.999 | 0.999 | 0.999 |
| Learning Rate | 0.0001 | 0.0001 | 0.0001 | 0.0001 | 0.0001 | 0.0001 |
| Weight Decay | 0 | 0 | 0 | 0 | 0 | 0 |
| Batch Size | 4096 | 4096 | 4096 | 4096 | 4096 | 4096 |
| $M$ | 4 | 4 | 4 | 4 | 4 | 4 |
| Kernel | Laplace | Laplace | Laplace | Laplace | Laplace | Laplace |
| $r(s,t)$ | $\max(s, \eta^{-1}(\eta_t - \frac{(\eta_{\max}-\eta_{\min})}{2^{15}}))$ | $\max(s, \eta^{-1}(\eta_t - \frac{(\eta_{\max}-\eta_{\min})}{2^{12}}))$ | | | | |
| Minimum $t, r$ gap | - | - | - | - | - | $10^{-4}$ |
| $p(t)$ | $\mathcal{U}(0.006, 0.994)$ | $\mathcal{U}(0, 0.994)$ | $\mathcal{U}(0, 0.994)$ | $\mathcal{U}(0, 0.994)$ | $\mathcal{U}(0, 0.994)$ | $\mathcal{U}(0, 0.994)$ |
| $a$ | 1 | 1 | 1 | 1 | 1 | 2 |
| $b$ | 5 | 4 | 4 | 4 | 4 | 4 |
| EMA Rate | 0.9999 | 0.9999 | 0.9999 | 0.9999 | 0.9999 | 0.9999 |
| Label Dropout | - | 0.1 | 0.1 | 0.1 | 0.1 | 0.1 |
| $x$-flip | True | False | False | False | False | False |
| EDM augment (Karras et al., 2022) | True | False | False | False | False | False |
| **Inference Setting** | | | | | | |
| Sampler Type | Pushforward | Pushforward | Pushforward | Pushforward | Pushforward | Pushforward |
| Number of Steps | 2 | 8 | 8 | 8 | 8 | 8 |
| Schedule Type | $t_1 = 1.4$ | Uniform | Uniform | Uniform | Uniform | Uniform |
| FID-50K ($w = 0$) | 1.98 | - | - | - | - | - |
| FID-50K ($w = 1.0$, *i.e.* no guidance) | - | 42.28 | 26.02 | 9.33 | 7.25 | 6.69 |
| FID-50K ($w = 1.5$) | - | 20.36 | 9.69 | 2.80 | 2.13 | 1.99 |

*Table 5.* Experimental settings for different architectures and datasets.

ImageNet into latents without flipping, and calculate the channel-wise mean and std for normalization. We use Stable Diffusion VAE[3] and rescale the latents by channel mean $[0.86488, -0.27787343, 0.21616915, 0.3738409]$ and channel std $[4.85503674, 5.31922414, 3.93725398, 3.9870003]$. After this normalization transformation, we further multiply the latents by $0.5$ so that the latents roughly have std $0.5$. For DiT architecture of different sizes, we use the same hyperparameters for all experiments.

**Choices for $T$ and $\epsilon$.** By default assuming we are using mapping function $r(s,t)$ by constant decrement in $\eta_t$, we keep $\eta_{\max} \approx 160$. This implies that for time distribution of the form $\mathcal{U}(\epsilon, T)$, we set $T = 0.996$ for cosine diffusion and $T = 0.994$ for OT-FM. For $\epsilon$, we set it differently for pixel-space and latent-space model. For pixel-space on CIFAR-10, we follow Nichol & Dhariwal (2021) and set $\epsilon$ to a small positive constant because pixel quantization makes smaller noise imperceptible. We find $0.006$ to work well. For latent-space on ImageNet-256×256, we have no such intuition as in pixel-space. We simply set $\epsilon = 0$ in this case.

Exceptions occur when we ablate other choices of $r(s,t)$, *e.g.* constant decrement in $\lambda_t$ in which case we set $\epsilon = 0.001$ to prevent $r(s,t)$ for being too close to $t$ when $t$ is small.

**Injecting time $s$.** The design for additionally injecting $s$ can be categorized into 2 types – injecting $s$ directly and injecting stride size $(t - s)$. In both cases, architectural designs exactly follow the time injection of $t$. We simply extract positional time embedding of $s$ (or $t - s$) fed through 2-layer MLP (same as for $t$) before adding this new embedding to the embedding of $t$ after MLP. The summed embedding is then fed through all the Transformer blocks as in standard DiT architecture.

---

[3] https://huggingface.co/stabilityai/sd-vae-ft-mse

| | 10-step | 16-step | 32-step |
|---|---|---|---|
| FID w/ $w = 1.5$ | 1.98 | 1.90 | **1.89** |

*Table 6.* FID of ImageNet-256×256 beyond 8 steps presented in the main text. At 16 steps, IMM already outperforms VAR with 2B parameters (1.92 FID) and its performance saturates beyond that, with 32 steps performing marginally better.

| | 1-step | 2-step | 4-step | 8-step |
|---|---|---|---|---|
| $a = 1$ | 7.97 | 4.01 | 2.61 | 2.13 |
| $a = 2$ | 8.28 | 4.08 | 2.60 | **2.01** |

*Table 7.* Ablation of exponent $a$ in the weighting function on ImageNet-256×256. We see that $a = 2$ excels at multi-step generation while lagging slightly behind in 1-step generation.

| | 1-step | 2-step | 4-step | 8-step |
|---|---|---|---|---|
| TF32 w/ $a = 1$ | 7.97 | 4.01 | 2.61 | 2.13 |
| FP16 w/ $a = 1$ | 8.73 | 4.54 | 3.03 | 2.38 |
| FP16 w/ $a = 2$ | 8.05 | 3.99 | 2.51 | **1.99** |

*Table 8.* Ablation of lower precision training on ImageNet-256×256. For lower precision training, we employ both minimum gap $\Delta = 10^{-4}$ and $(t - r)$ conditioning.

**Improved CT baseline.** For ImageNet-256×256, we implement iCT baseline by using our improved parameterization with Simple-EDM and OT-FM schedule. We use the proposed pseudo-huber loss for training but find training often collapses using the same $r(s, t)$ schedule as ours. We carefully tune the gap to achieve reasonable performance without collapse and present our results in Table 2.

## I.2. Inference Settings

**Inference schedules.** For all one-step inference, we directly start from $\epsilon \sim \mathcal{N}(0, \sigma_d^2 I)$ at time $T$ to time $\epsilon$ through pushforward sampling. For all 2-step methods, we set the intermediate timestep $t_1$ such that $\eta_{t_1} = 1.4$; this choice is purely empirical which we find to work well. For $N \geq 4$ steps we explore two types of time schedules: (1) uniform decrement in $t$ with $\eta_0 < \eta_1 \cdots < \eta_N$ where

$$t_i = T + \frac{N - i}{N}(\epsilon - T) \tag{117}$$

and (2) EDM (Karras et al., 2022) time schedule. EDM schedule specifies $\eta_0 < \eta_1 \cdots < \eta_N$ where

$$\eta_i = \left( \eta_{\max}^{\frac{1}{\rho}} + \frac{N - i}{N}(\eta_{\min}^{\frac{1}{\rho}} - \eta_{\max}^{\frac{1}{\rho}}) \right)^{\rho} \quad \text{and} \quad \rho = 7 \tag{118}$$

We slightly modify the schedule so that $\eta_0 = \eta_{\min}$ is the endpoint instead of $\eta_1 = \eta_{\min}$ and $\eta_0 = 0$ as originally proposed, since our $\eta_0$ can be set to 0 without numerical issue.

We also specify the time schedule type used for our best runs in Table 5 and their results.

## I.3. Scaling Settings

**Model GFLOPs.** We reuse numbers from DiT (Peebles & Xie, 2023) for each model architecture.

**Training compute.** Following Peebles & Xie (2023), we use the formula model GFLOPs · batch size · training steps · 4 for training compute where, different from DiT, we have constant 4 because for each iteration we have 2 forward pass and 1 backward pass, which is estimated as twice the forward compute.

**Inference compute.** We calculate inference compute via model GFLOPs · number of steps.

## I.4. Scaling Beyond 8 Steps

We investigate when the performance saturates for ImageNet-256×256 in Table 6. We see continued improvement beyond 8 steps and at 16 steps our method already outperforms VAR with 2B parameters (1.92 FID).

## I.5. Ablation on exponent $a$

We compare the performance between $a = 1$ and $a = 2$ on full DiT-XL architecture in Table 7, which shows how $a$ affects results of different sampling steps. We observe that $a = 2$ causes slightly higher 1-step sampling FID but outperforms $a = 1$ in the multi-step regime.

### I.6. Caveats for Lower-Precision Training

For all experiments, we follow the original works (Song et al., 2020b; Peebles & Xie, 2023) and use the default TF32 precision for training and evaluation. When switching to lower precision such as FP16, we find that our mapping function, *i.e.* constant decrement in $\eta_t$, can cause indistinguishable time embedding after some MLP layers when $t$ is large. To mitigate this issue, we simply impose a minimum gap $\Delta$ between any $t$ and $r$, for example, $\Delta = 10^{-4}$. Our resulting mapping function becomes

$$r(s,t) = \max\left(s, \min\left(t - \Delta, \eta^{-1}(\eta(t) - \epsilon)\right)\right)$$

Optionally, we can also increase distinguishability between nearby time-steps inside the network by injecting $(r - s)$ instead of $s$ as our second time condition. We use this as default for FP16 training. With these simple changes, we observe minimal impact on generation performance.

Lastly, if training from scratch with lower precision, we recommend FP16 instead of BF16 because of higher precision that is needed to distinguish between nearby $t$ and $r$.

We show results in Table 8. For FP16, $a = 1$ causes slight performance degradation because of the small gap issue at large $t$. This is effectively resolved by $a = 2$ which downweights losses at large $t$ by focusing on smaller $t$ instead. At lower precision, while not necessary, $a = 2$ is an effective solution to achieve good performance that matches or even surpasses that of TF32.

## J. Additional Visualization

We present additional visualization results in the following page.

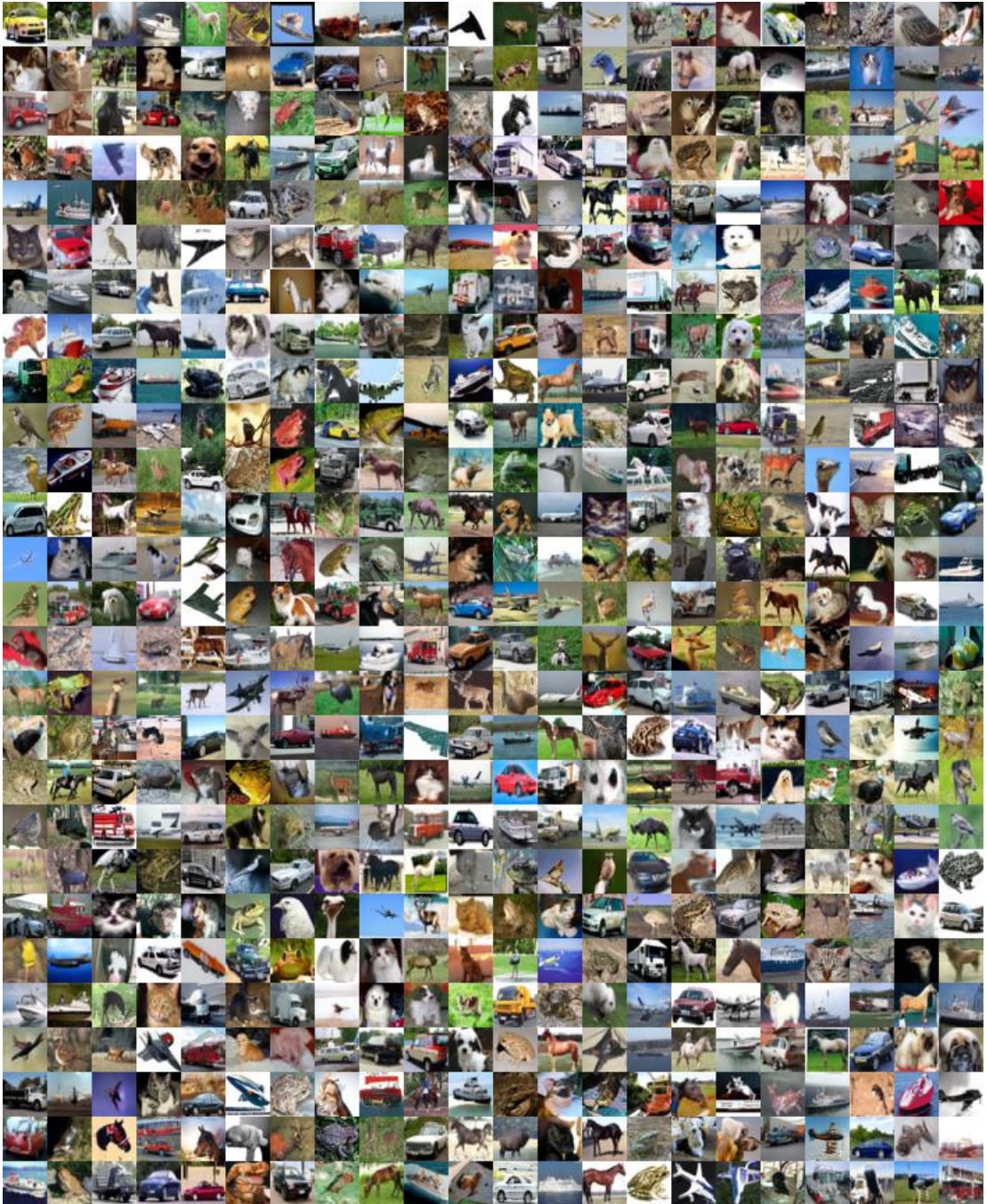

*Figure 12.* Uncurated samples on CIFAR-10, unconditional, 2 steps.

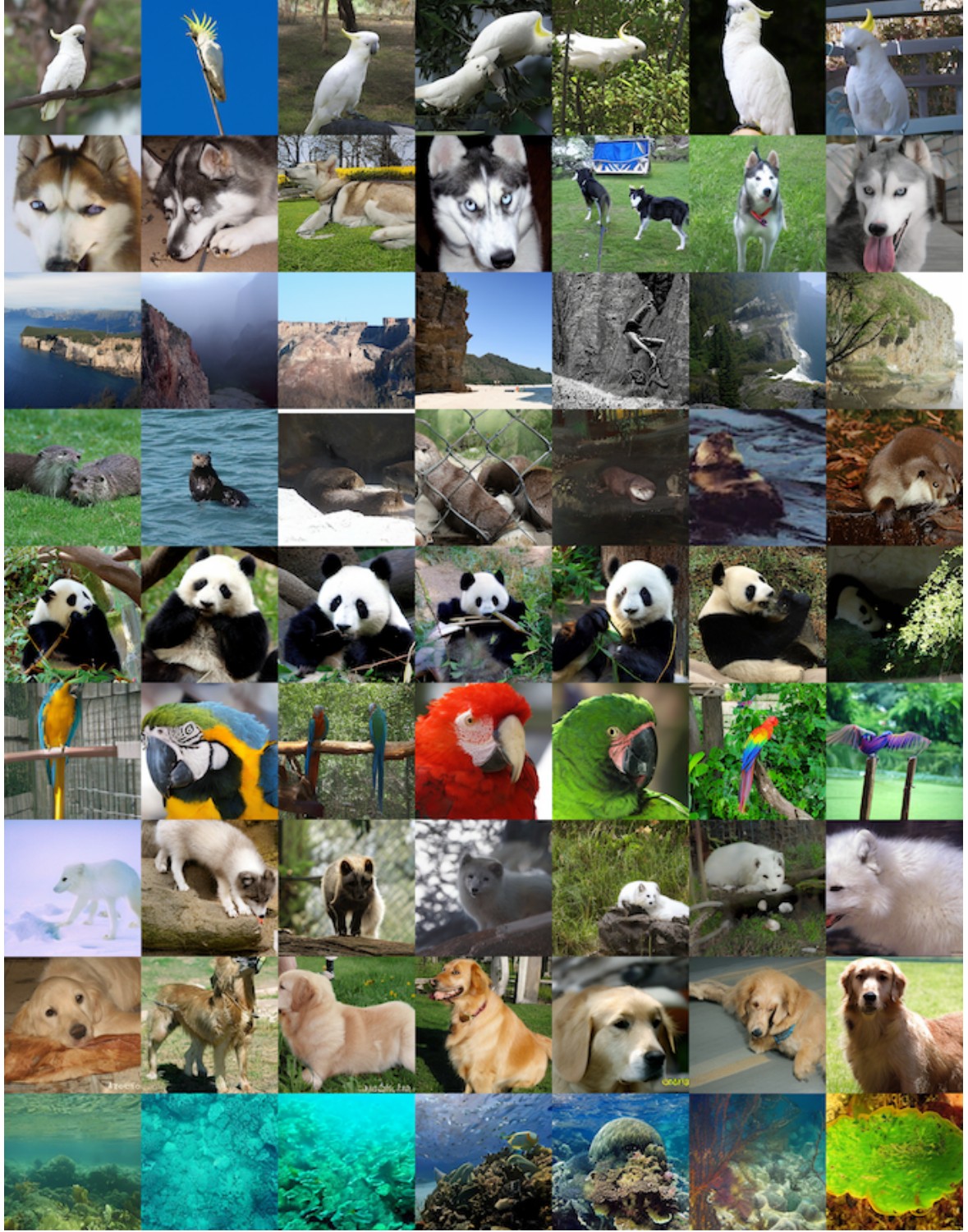

*Figure 13.* Uncurated samples on ImageNet-256×256 using DiT-XL/2 architecture. Guidance $w = 1.5$, 8 steps.

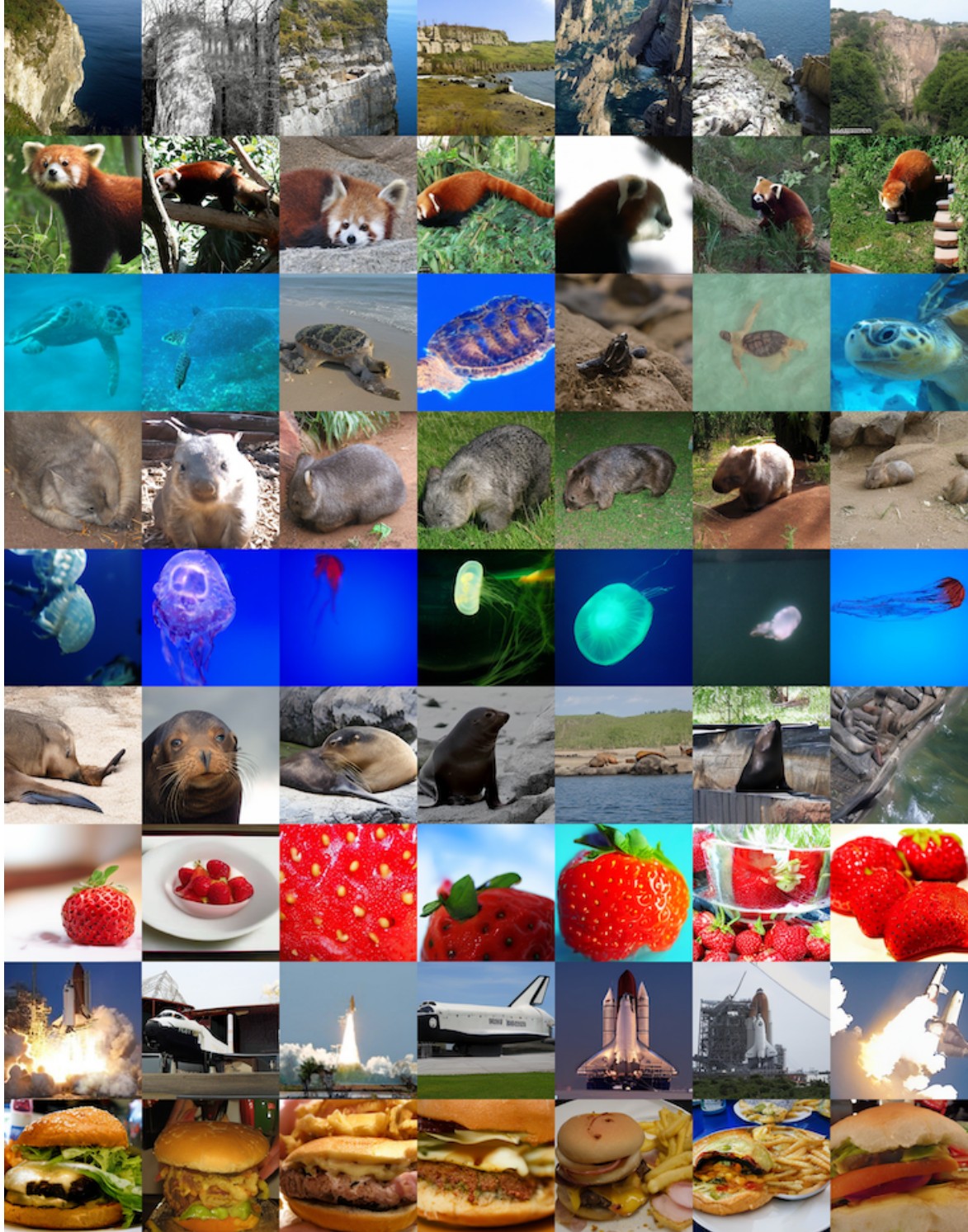

*Figure 14.* Uncurated samples on ImageNet-256×256 using DiT-XL/2 architecture. Guidance $w = 1.5$, 8 steps.

