# OpenReview forum: "Inductive Moment Matching"
_ICML.cc/2025/Conference — ICML 2025 oral_

### Official Review · Reviewer_m3Qo · 2025-03-12

**Overall Recommendation:** 4

**Summary:**

# Update

I gave the score of 4. My complaints are minor, and the authors have addressed them in the rebuttal. I'm comformtable with the paper being published in the form close what it is right now. So, I decided to not change my evaluation.

# Old Summary

This paper presents moment matching self distillation (MMSD), an algorithm for training a generative model from scratch that is based on the stochastic interpolant framework.

Given a data distribution $q(x)$ and a noise distribution $p(\epsilon)$, the stochastic framework constructs a stochastic process $\\{ q_t(x_t) : 0 \leq t \leq 1\\}$ such that $q_0(x_0) = q(x_0)$ and $q_1(x_1) = p(x_1)$. The generative model trained by MMSD is of the form $f^{\theta}\(x_t, s, t)$ where $\theta$ denotes the model parameter. The specification is that, for any $0 \leq s \leq t \leq 1$, if $x_t \sim q_t$, then $f^{\theta}(x_t, s, t) \sim q_s$. A well-trained model can thus be used to generate a data sample from a noise sample in 1 evaluation of $f^\theta$ or any other number of function evaluations.

The proposed algorithm is as follows. In each training iteration:
1. We sample two times $s$ and $t$ such that $0 \leq s < t \leq 1$. We also compute an intermediate time $r = r(s,t)$ such that $s < r < t$.
2. We then sample $N$ pairs of data items and noise vectors $(x_1, \epsilon_1)$, $(x_2, \epsilon_2)$, $\dotsc$, $(x_M, \epsilon_M)$ where $x_i \sim q$ and $p_i \sim p$.
3. For each $1 \leq i \leq N$, we compute $x_{t,i} = \alpha_t x_i + \sigma_t \epsilon_i$ and $x_{r,i} = \alpha_r x_i + \sigma_r \epsilon_i$ where $\alpha_t$ and $\sigma_t$ are the noise schedule functions used in the formulation of $q_t$.
4. Compute two sets of samples $\\{ y_{s,t,i}: y_{s,t,i} = f^{\theta}(x_{t,i}, s, t) \\}$ and $\\{ y_{s,r,i} : y_{s,r,i} = f^{\theta^-}(x_{r,i}, s, t) \\}$ where $\theta^-$ is the gradient-stop version of $\theta$.
5. Compute the loss $\mathcal{L}$ as the distance between the two sets of samples $\\{ y_{s,t,i} \\}$ and $\\{ y_{s,r,i} \\}$ using the minimum mean discrepancy (MMD).
6. Update $\theta$ with the gradient of $\mathcal{L}$.

The algorithm is strikingly simialr to a previous algorithm, consistency training (CT) [1]. The main difference MMSD uses MMD to compute the loss, but CT simply computes the loss as $\sum_i \mathcal{d}(y_{s,t,i}, y_{s,r,i})$ where $d$ is a distance metric. Another difference is that, in CT, the time $s$ is always zero, but it is variable in MMSD.

The paper provides theoretical basis for the MMSD algorithm and shows that CT is one of its special case. It demonstrates that MMSD is effective by competitive FID scores acheived by MMSD-trained models on the CIFAR-10 and ImageNet 256x256 datasets. It also claims that MMSD training in more stable than some other algorithms, and that MMSD does not require hypertuning.

*Citation*
* [1] Song et al. "Consistency Models." ICML 2023.

**Claims And Evidence:**

The paper makes several claims.

1. MMSD is theoretically sound.
2. MMSD generalizes CT.
3. MMSD results in competitive and fast generative models.
4. MMSD training is stable.
5. The abstract implies that MMSD does not require extensive tuning.

I do not have particular problems with (1), (2), and (3). The FID scores on CIFAR-10 and ImageNet 256x256 are good given that the models were trained from scratch.

However, (4) and (5) are not so evident. MMSD introduces a quite number of hyperparameters: the kernel function to use for MMD, the number of groups M inside a minibatch, how to compute $r(s,t)$, and the weighting function $w(s,t)$. It also inherits conditioning parmaeters from the EDM and EDM2 papers. These hyperparameters have to be picked carefully in order for the model to achieve good scores. (Some hyperparameters such as the weighting function $w(s,t)$, are derived from previous works, and I believe this is a form of extensive tuning.)  Moreover, the paper indicates that hyperparameters have impact on training stability. For example, picking $M \leq 2$ or picking $r$ to be too close to $t$ lead to training to become unstable.

**Essential References Not Discussed:**

The authors cite Tee et al.'s physics-informed distillation (PID) paper but did not discuss its approach. PID tries to make the network under training so that its velocity field conforms to ODE that defines the sampling trajectory of a diffusion model. This approach is quite different from the approach taken by the paper under review or other papers, and so a discussion can be illuminating. There are other several papers that take this approach that the authors might consider citing.

*Citation*
* [1] Boffi et al. Flow map matching. 2024.
* [2]  Yang et al. Consistency Flow Matching:Defining Straight Flows with Velocity Consistency. 2024.

**Experimental Designs Or Analyses:**

The experiments are simple: training models with MMSD on benchmark datasets using well-known architectures and compare the resulting FID scores to previous works. I find no particular problems.

**Methods And Evaluation Criteria:**

The paper uses CIFAR-10 and ImageNet 256x256, which are standard datasets for benchmarking generative models. It also uses the standard metric, the FID. There are no issues with these choices. The experiments are quite straightfoward as well: training models with MMSD and compare their FID scores to prior works. The paper includes scores upto NFE = 2 for CIFAR-10 and NFE = 8 for ImageNet 256x256 (along with guidance scale used), which makes comparison easy.

**Other Comments Or Suggestions:**

In Section 7.2, "we haver a fixed batch size $B$ in which the samples are grouped into $B/M$ group of $M$ where each group shares the same $t$. Shouldn't this be "where each groups shares the same $s$ and $t$?

How are $s$ and $t$ sampled during training? Am I missing something?

**Other Strengths And Weaknesses:**

N/A

**Questions For Authors:**

I would like to know how much wall-clock time training with MMSD is takes with different group sizes M but the same batch size B. Including this information in the appendix would be quite helpful for other practioners.

**Relation To Broader Scientific Literature:**

Training a fast generative model in a stable and reproducible manner from scratch is an important problem. This paper contributes a novel and relatively simple algorithm that works well on widely used benchmarks, and I think this is an important contribution.

**Theoretical Claims:**

I skimmed through most proofs, but did not check all the calculations. The proofs were generally easy to follow and seem to be sound.

---

> ### Author Rebuttal · Authors · 2025-03-31
>
> We thank the reviewer for the insights and suggestions. We would like to first refer the reviewer to the **Overall Update** section in our rebuttal for reviewer SxrL for important updates to the paper. And we would like to address the concerns below.
>
> > Claims on (4) training stability and (5) not requiring extensive tuning.
>
> We stress our abstract only claims that our method, similar to diffusion models, remains stable in the sense that common parameter choices can lead to a stable training process and decent quality, without the model becoming degenerate. We do not claim that they all obtain “good scores”, which may have been misinterpreted by the reviewer. In fact, even diffusion models require specific designs in the architecture to reach optimal scores – for example, the DiT paper found that AdaLN gives best performance with timestep conditioning, and in the EDM paper it is shown that variance preserving is much less sensitive to hyperparameter choices than variance exploding (see config A and config B in the paper).  For best performance in IMM, similar to diffusion and Flow Matching, we should expect some degree of tuning and different hyperparam choices. The stability is substantiated in Figure 4,5,6, and 7 which show that, as claimed in abstract, our method converges across different time embedding, time conditioning functions, $M$, $r(s,t)$, and $w(s,t)$ params. We introduced variability in these choices precisely to show our method’s stability across them.
>
> For $M$, we emphasize that our convergence stability is not sensitive to $M$ as long as $M\ge 4$, which is a reasonable range for practical purposes. No deep learning methods can train stably for *all* parameter choices (e.g. even for diffusion too large of lr or incorrect weighting for some $t$ can lead to degeneracy). We can only safely guarantee the stability exists for a large range of choices, which is sufficient for practical success. Similarly for $r$, as long as $r$ and $t$ has a reasonable finite gap (i.e. $k\le 12$ in our case), we can achieve good performance.
>
>
> We also argue $w(s,t)$ did not require extensive tuning and is simply carried over from VDM and Flow Matching. As motivated in Appendix C.9, since no weighting can be derived from MMD itself, we resort to the closest diffusion counterpart as in VDM [1], from which the terms $\frac{1}{2}\sigma(b-\lambda_t) (-\frac{d}{dt}\lambda_t)$ are carried over. The term $\alpha_t$ is also theoretically motivated to follow VDM gradients. These terms may look complex at first to suggest extensive tuning but they come as a group and do not require ablation of each individual subterms. The only additional  $\alpha_t^2 + \sigma_t^2$ term is simply motivated by [2] for Flow Matching schedule that upweights middle timesteps. The same effect can be achieved by sampling more middle timesteps instead.
>
>
>
> We also use EDM conditioning as a simple unifying notation for different parameterization choices (e.g. DDIM and Euler sampler can both be unified under this one notation, see Appendix C.5). We do not reuse the complex conditioning introduced in EDM. In fact, we find the most simple Euler parameterization $f_{s,t}(x_t) = x_t + (s-t)G_\theta(x_t,s,t)$ to work the best for ImageNet-256x256 (see Appendix C.5 and Table 4) and do not suggest deviating from this standard choice.
>
>
>
>
> > Discussion of Tee et al.'s physics-informed distillation (PID)
>
> PID is related as a distillation technique that explicitly matches the network’s own velocity field with that of diffusion. Different from other distillation methods, PID inputs noise $z$ and outputs $x_t$ whose derivative is matched with pretrained diffusion velocity field. The skip connection shares similarity with CM but requires $c_\text{out}(T)=1$ instead of $c_\text{out}(T)=0$, and different from distribution-matching distillation, it does not need to jointly train two networks.
>
> In addition to PID, we will discuss the two additional works in our revision.
>
> > Shouldn't this be "where each groups shares the same $t$ and $s$?
>
> Yes. This group should share the same $t$ and $s$.
>
>
> > Wall-clock time training with different group sizes M but the same batch size B.
>
> With $B=4096$ with DiT-XL architecture, we experimented across $M=2,4,8,16$ and find that all choices have per-step wallclock time of 0.53-0.55 seconds (one step here means a full optimization step accounting for both forward and backward passes). This is consistent with our analysis in Appendix C.4 that forward/backward pass is the computational bottleneck and time for computing the $M\times M$ matrix is negligible.
>
> \
> &nbsp;
>
>
> [1] Kingma, Diederik, and Ruiqi Gao. "Understanding diffusion objectives as the elbo with simple data augmentation." Advances in Neural Information Processing Systems 36 (2023): 65484-65516.
>
> [2] Esser, Patrick, et al. "Scaling rectified flow transformers for high-resolution image synthesis." Forty-first international conference on machine learning. 2024.

---

### Official Review · Reviewer_oBFV · 2025-03-17

**Overall Recommendation:** 4

**Summary:**

The paper introduces Moment Matching Self-Distillation (MMSD), a novel framework for training few-step generative models from scratch. MMSD offers a single-stage training procedure that avoids the need for pre-training or optimizing two networks. It leverages self-consistent interpolants to match the moments of its distribution to that of the data, ensuring distribution-level convergence.

## update after rebuttal

I thank the authors for their detailed response, which clarified my concerns about the use of classifier-free guidance in distillation models and the relationship between their method and Consistency Training. Given the authors' clarifications and commitments to improving notation and readability, I am happy to maintain my Accept recommendation.

**Claims And Evidence:**

The claims made in the submission are supported by clear and convincing evidence. The authors demonstrate the effectiveness of MMSD through extensive experiments on CIFAR10 and ImageNet 256×256.

**Essential References Not Discussed:**

After reviewing the paper, I did not find any particularly critical or essential references that were missing from the discussion.

**Experimental Designs Or Analyses:**

The experimental setups are well-structured and comprehensive. The authors evaluate MMSD on standard image benchmarks, including CIFAR-10, and ImageNet 256×256, which are widely used and respected in the field of generative modeling.

**Methods And Evaluation Criteria:**

The proposed methods and evaluation criteria are well-suited for the problem and application at hand. The use of benchmark datasets such as CIFAR-10 and ImageNet 256×256 is appropriate, as these are widely recognized and challenging benchmarks for evaluating generative models. The evaluation metric, Fréchet Inception Distance (FID), is a standard and reliable measure for assessing the quality and diversity of generated images.

**Other Comments Or Suggestions:**

I do not have any additional comments or suggestions for the paper.

**Other Strengths And Weaknesses:**

Strength:

1. A key strength of this paper is its novel and efficient approach to training few-step generative models from scratch.
2. By leveraging moment matching and self-consistent interpolants, MMSD guarantees distribution-level convergence and maintains stability across various settings, making it both robust and practical.
3. The theoretical formulations are rigorous.

Weakness:

1. The rationality of classifier-free guidance for distillation models is questionable, particularly for one-step models. The experiment also demonstrates that cfg = 1.5 yields inferior results compared to cfg = 1.25 when step = 1.
2. The writing and organization of this paper need further refinement to improve readability. Some proofs, such as the proof of Theorem 1, which I believe uses induction to extend from an infinitesimal step to a long-range step, are somewhat obscured by overly complex notation.

**Questions For Authors:**

I don't think consistency models can be entirely considered a special case of this paper's framework. While consistency models can be understood from a distribution matching perspective, they are theoretically guaranteed to be trajectory-based distillation. Based on my understanding, this paper's method doesn't theoretically guarantee that the learned deterministic mapping is consistent with the PF-ODE mapping, since the framework is at the distribution level. And re-using $x_t$ for $x_r$ is aligned with consistency training only if r is very close to t. I'm not sure if this understanding is correct.

**Relation To Broader Scientific Literature:**

The key contributions of the paper are related to diffusion/flow matching and distribution matching training/distillation, such as MMD-GAN, consistency model, and adversarial training.

**Theoretical Claims:**

I have gone through the theoretical proofs in the paper to a reasonable extent, and while I did not perform a 100% detailed verification, the proofs appear to be correct and well-justified.

---

> ### Author Rebuttal · Authors · 2025-03-31
>
> We thank the reviewer for the insights and suggestions. We would like to first refer the reviewer to the **Overall Update** section  in our rebuttal for reviewer SxrL for important updates to the paper. And we would like to address the concerns below.
>
>
> > The rationality of classifier-free guidance for distillation models is questionable, particularly for one-step models. The experiment also demonstrates that cfg = 1.5 yields inferior results compared to cfg = 1.25 when step = 1.
>
> It is true that theoretically CFG in the diffusion/Flow Matching context does not work well in 1-step context because we are no longer modeling the velocity field at any $t$. In addition, the transition kernel defined by this 1-step CFG should be subject to an acceptance rate as in Metropolis adjusted MCMC (see [3] for details) without which the generated distribution can deviate from data distribution. In practice, since we cannot evaluate the likelihood, we always accept the samples. In general, however, we empirically find that adding some CFG values indeed helps with quality. We find that without CFG, 1-step sampler yields 12.21 FID on ImageNet-256x256, which is significantly higher than the results using CFG (7.97 and 7.12). We hypothesize that with 1 step, the linear combination of conditional and unconditional branches can still effectively correct visual details that are inaccurately modeled using either branch. Our conclusion is that CFG is helpful for 1-step sampler, but CFG values that achieve superior results in multi-step regime may not necessarily transfer to 1 step because 1-step CFG is not as well theoretically motivated as its multi-step counterparts. We call for more studies on this phenomenon in future works.
>
> > The writing and organization of this paper need further refinement to improve readability. Some proofs, such as the proof of Theorem 1, which I believe uses induction to extend from an infinitesimal step to a long-range step, are somewhat obscured by overly complex notation.
>
> We will streamline our notation in proofs in our revision.
>
>
> > Questions regarding CT as a special case.
>
> We stress an important point: minimizer of Consistency Distillation (CD) loss is the PF-ODE of the pretrained diffusion but minimizer of Consistency Training (CT) loss is NOT the PF-ODE. Minimizer of CT coincides with PF-ODE only when data distribution is delta distribution, as assumed by many proofs in the original CM paper [1] (Appendix B.1 Remark 4) and iCT [2] (Sec 3.2).
>
> To see CT does not learn PF-ODE, recall CM loss $$\mathbb{E}\_{x_t,x,t}[w(t) || g_\theta(x_t,t) - g_{\theta^-}(x_r, r)  ||^2]$$, where $x_r$ is ODE solution from $x_t$. Assume $w(t)=1$, the minimizer of this loss is $$ g_{\theta^*}(x_t,t) = \mathbb{E}\_{x|x_t}[g_{\theta^-}(x_r, r)  ] $$. For CD, $x_r$ is ODE solution using pretrained score so $x_r$ does not depend on $x$, i.e. the conditional expectation can be dropped and $$ \mathbb{E}\_{x|x_t}[g_{\theta^-}(x_r, r)  ] = g_{\theta^-}(x_r, r)$$. However, for CT, $x_r$ depends on $x$, and the conditional expectation is irreducible. If we assume $g_{\theta^-}(x_r, r) $ is a PF-ODE result, this new minimizer at $t$ deviates from any single PF-ODE due to the expectation. The original CM paper [1] and iCT [2] have a delta-distribution assumption that similarly allow elimination of the conditional expectation, which downplays the aforementioned problem in the general case.
>
> We therefore call for understanding its loss at a distribution level, and find that it is a special case from this moment-matching perspective.
>
> > Our method does not guarantee learning of PF-ODE.
>
> We do not guarantee our method learns PF-ODE. However, our method can converge to a different and equally valid solution whose *distribution* matches the data distribution. Consider a toy 2D Gaussian distribution $\mathcal{N}(0,I)$ as data, and the same Gaussian $\mathcal{N}(0,I)$ as prior. The PF-ODE is an identity function mapping any point $x$ to itself. However, consider another function $$f_{s,t}(x_t) = \text{rotate}(x_t, 2\pi*(t-s))$$ where $\text{rotate}(x, \phi)$ is a rotation operation around the origin by angle $\phi$. Since Gaussian is rotation-invariant, distribution of $f_{s,t}(x_t)$ stays  $\mathcal{N}(0,I)$. This is another valid solution under distribution matching objective which our method can also possibly learn.
>
>
> \
> &nbsp;
>
>
> [1] Song, Yang, et al. "Consistency models." (2023).
>
> [2] Song, Yang, and Prafulla Dhariwal. "Improved techniques for training consistency models." arXiv preprint arXiv:2310.14189 (2023).
>
> [3] Du, Yilun, et al. "Reduce, reuse, recycle: Compositional generation with energy-based diffusion models and mcmc." International conference on machine learning. PMLR, 2023.

---

### Official Review · Reviewer_SxrL · 2025-03-17

**Overall Recommendation:** 4

**Summary:**

This paper introduces a novel generative model enabling the few-step generation of high-quality photorealistic images. The approach builds on prior work in consistency trajectory models [1] and flow map matching [2], where a generator learns to move a noisy image from one timestep to another. However, unlike [1,2], which focus on pointwise matching along the Probability Flow ODE (PF-ODE), the proposed method instead matches the marginal distribution.

The training follows a bootstrapping approach similar to consistency models, while distribution matching is achieved via Maximum Mean Discrepancy (MMD), eliminating the need for auxiliary networks required in prior methods like DMD [3] and MMD [4].

Beyond theoretical contributions, the paper introduces practical improvements for stable training and demonstrates strong performance on ImageNet with models trained from scratch.


[1] Kim, Dongjun, et al. "Consistency trajectory models: Learning probability flow ode trajectory of diffusion." arXiv preprint arXiv:2310.02279 (2023).

[2] Boffi, Nicholas M., Michael S. Albergo, and Eric Vanden-Eijnden. "Flow Map Matching." arXiv preprint arXiv:2406.07507 (2024).

[3] Yin, Tianwei, et al. "One-step diffusion with distribution matching distillation." Proceedings of the IEEE/CVF conference on computer vision and pattern recognition. 2024.

[4] Salimans, Tim, et al. "Multistep distillation of diffusion models via moment matching." Advances in Neural Information Processing Systems 37 (2024): 36046-36070.

**Claims And Evidence:**

Yes, they are well-supported by theoretical proofs and strong empirical results.

**Essential References Not Discussed:**

The discussion on related works are adequate.

**Experimental Designs Or Analyses:**

Yes, the design is sound.

**Methods And Evaluation Criteria:**

This paper is well-motivated, and the proposed method is intuitively sound. It follows a recent trend in multi-step generation, where the generator is trained to produce samples resembling those encountered at intermediate noisy distributions rather than directly targeting the clean distribution. This approach aligns with methods such as Piecewise Rectified Flow [1], the consistency trajectory model [2], and MMD [3], among others. It is exciting to see a novel method emerge along the same lines but driven by a different distribution matching objective.

Furthermore, the use of Maximum Mean Discrepancy (MMD) as a training signal is well-justified. It eliminates the need for auxiliary network training, simplifying the approach while enabling stable training and strong performance.

Beyond these methodological innovations, the overall results are solid, with rigorous benchmarking, systematic comparisons, and convincing ablation studies.

[1] Yan, Hanshu, et al. "Perflow: Piecewise rectified flow as universal plug-and-play accelerator." arXiv preprint arXiv:2405.07510 (2024).

[2] Kim, Dongjun, et al. "Consistency trajectory models: Learning probability flow ode trajectory of diffusion." arXiv preprint arXiv:2310.02279 (2023).

[3] Salimans, Tim, et al. "Multistep distillation of diffusion models via moment matching." Advances in Neural Information Processing Systems 37 (2024): 36046-36070.

**Other Comments Or Suggestions:**

n/a

**Other Strengths And Weaknesses:**

I would like to see additional results on T2I or T2V applications. These experiments should be relatively easy to do if initialized from a pretrained diffusion model and could greatly enhance the case for broader adoption.

**Questions For Authors:**

Q1: Would increasing the number of steps beyond 8 consistently improve performance? Is this improvement dependent on the training strategy, such as the choice of r(s,t)?

Q2: In MMD [1], a DDIM-style sampler is also used during inference. Could this be applied to the proposed method? Currently, it appears that only the consistency sampler is utilized.

[1] Salimans, Tim, et al. "Multistep distillation of diffusion models via moment matching." Advances in Neural Information Processing Systems 37 (2024): 36046-36070.

**Relation To Broader Scientific Literature:**

It has broader implications for the generative modeling community, enabling end-to-end training of models that support both few-step and many-step inference. By eliminating the need for an additional distillation step during deployment, this approach streamlines the training-to-inference pipeline.

**Theoretical Claims:**

I didn't check the full proofs.

---

> ### Author Rebuttal · Authors · 2025-03-31
>
> # Overall Update:
> \
> We thank all reviewers for their insights and helpful suggestions. We would like to announce several important updates.
>
> - To distinguish our method from other distillation-based post-training techniques, we change our title and model name from **“Moment Matching Self-Distillation”** to **“Inductive Moment Matching”**.
> - Better experimental results that outperforms the ones reported in submission.
>
> |     | CIFAR-10 FID |
> | -------- | :-------: |
> | 1-step  |  3.20    |
> | 2-step |  **1.98**     |
>
>
> |     | ImageNet-256x256 FID |
> | -------- | :-------: |
> | 1-step (w=1.25)  | 7.77   |
> | 2-step (w=1.25)   |  5.33     |
> | 4-step (w=1.25)   |  3.66     |
> | 8-step (w=1.25)   |  2.77     |
> | -------- | ------- |
> | 1-step (w=1.5)  | 8.05   |
> | 2-step (w=1.5)   |  3.99     |
> | 4-step (w=1.5)   |  2.51     |
> | 8-step (w=1.5)   |  **1.99**     |
>
> - Scaling beyond 8 steps for ImageNet-256x256
>
> |     | ImageNet-256x256 FID |
> | -------- | :-------: |
> | 10-step (w=1.5)  | 1.98   |
> | 16-step (w=1.5)   |  **1.90**     |
> | 32-step (w=1.5)   |   **1.89**  |
>
> The results continue to improve beyond 8 steps. We see that at 16 steps it achieves 1.90 FID and already outperforms the 2B parameter VAR baseline (1.92 FID). We see saturation beyond 16 steps, with marginal improvement at 32 steps (1.89 FID).
>
> \
> &nbsp;
> ------------------------------------------
> ## For reviewer SxrL:
>
> Thank you for your comments and suggestions. We would like to address your concerns below.
>
> > I would like to see additional results on T2I or T2V applications. These experiments should be relatively easy to do if initialized from a pretrained diffusion model and could greatly enhance the case for broader adoption.
>
> We test our algorithm on a text-to-image model trained on Datacomp [*] at 512x512 resolution. Samples using 8 steps can be found at [this link](https://drive.google.com/file/d/1e_2d0S1g4TStIkNtWE8UqGOO-BZZc9r0/view?usp=drive_link). While this is a preliminary result, we can see that our algorithm can easily transfer to T2I settings.
>
>
>
> > Q1: Would increasing the number of steps beyond 8 consistently improve performance? Is this improvement dependent on the training strategy, such as the choice of r(s,t)?
>
> Yes the result continue to improve beyond 8 steps. See results above in “Overall Update” that 16 steps attain 1.90 FID and performance tends to saturate beyond that, with 32 steps attaining 1.89 FID. Notably, this outperforms the VAR variant with 2B parameters. We additionally show a comparison between 8 step generation and 16 step generation in [this link](https://drive.google.com/file/d/1T8h-k7IW4b4srOMZbBppuwFT-LRk1CLP/view?usp=drive_link). We notice that extending to 16 steps only give minor shifts in visual content in most cases. This demonstrates that 8 steps already give near optimal solutions -- our method scales *efficiently* with sampling compute. With low probability, there can be major shift in content although their low-frequency component look similar (i.e. they look alike from afar) (see row 2 col 3). Additionally, the relative improvement in FID should be similar across different $r(s,t)$, although their convergence rate can significantly differ. This is evident during training (see Figure 6) where constant decrement in $t$ noticeably lags behind constant decrement in $\eta_t$.
>
>
>
>
>
>
> > Q2: In MMD [1], a DDIM-style sampler is also used during inference. Could this be applied to the proposed method? Currently, it appears that only the consistency sampler is utilized.
>
> We actually investigate both DDIM sampler (i.e. pushforward sampler) and consistency sampler (i.e. restart sampler) (see Sec 4.3). Pushforward sampler is equivalent to DDIM (while additionally injecting $s$ as conditioning) because $f_{s,t}^\theta(x_t)$ is defined as a DDIM step from $t$ to $s$ with $g_\theta(x_t, s, t)$ as the $x$-prediction network (see line 193-194). In fact, we also show that DDIM sampler is better than consistency sampler in Sec 7.3 and Figure 8.
>
> \
> &nbsp;
>
> [*] Gadre, Samir Yitzhak, et al. "Datacomp: In search of the next generation of multimodal datasets." Advances in Neural Information Processing Systems 36 (2023): 27092-27112.

---

### Decision · Program_Chairs · 2025-05-01

**Decision:**

Accept (oral)

**Comment:**

The paper introduces Moment Matching Self-Distillation, renamed Inductive Moment Matching (IMM) during the rebuttal, as a new approach for pretraining unconditional or label-conditional deep generative models from scratch. IMM has a built-in capability to generate images in one or a few steps, while continuing to improve as the number of inference steps increases. The method generalizes the consistency model as a special case and illustrates how classical distribution matching techniques such as Maximum Mean Discrepancy (MMD) can be reimagined for pretraining high-performing generative models.

While the reviewers are all in favor of accepting the paper, they have identified several instances of potential overclaiming that should be moderated to more fairly represent prior work. For example, the manuscript criticizes existing methods as being unstable and requiring extensive tuning, while simultaneously adopting their well-studied architectures and hyperparameters. In particular, the abstract makes a broad assertion: “distilling them into few-step models often leads to instability and extensive tuning.” Such statements should be revised or substantiated with concrete evidence from the paper. Otherwise, there is a risk of misattributing isolated shortcomings of one method (e.g., Method A, a member of the distillation family) as general flaws of the entire methodological family that Method A belongs to. This kind of unsupported generalization could propagate misinterpretations in future literature, and hence should be avoided.

Upon reviewing the paper, the AC also notes that the claims would be better supported by including comparisons with consistency models and score distillation methods on ImageNet 512×512. Such comparisons would help position the performance of IMM more clearly.

The AC fully acknowledges the significance of the paper in demonstrating how to pretrain generative models from scratch with the built-in ability to generate high-quality samples in a few steps, and recognizes the strong performance achieved under this setting. However, it is important to clarify that state-of-the-art results for one or few-step generations for unconditional/label-conditional generations are still currently held by methods using diffusion pretraining combined with score distillation.

Another important point that requires clarification is the number of function evaluations (NFEs). For IMM, if the AC's understanding is correct, the NFEs should be calculated as **twice** the number of inference steps, due to the use of classifier-free guidance (CFG) during generation. In contrast, many distillation-based methods do **not** apply CFG directly to the generator, but rather to the teacher and/or an auxiliary network. Therefore, reporting only the number of inference steps—without accounting for the doubled computational cost from CFG—can lead to misleading comparisons across methods. If the AC’s assessment is accurate, the paper should explicitly report NFEs to support fair and transparent performance evaluations.

Finally, the AC would like to highlight that Reviewer m3Qo has provided an excellent summary of the actual implementation of the proposed algorithm—one that the AC finds clearer and more accessible than the current version in the paper. The authors are encouraged to incorporate these details to improve clarity and make the implementation more understandable for readers.